# scSplit: Bringing Severity Cognizance to Image Decomposition in Fluorescence Microscopy

**Ashesh Ashesh**
Human Technopole
ashesh.ashesh@fht.org

Florian Jug
Human Technopole
florian.jug@fht.org

## Abstract

Fluorescence microscopy, while being a key driver for progress in the life sciences, is also subject to technical limitations. To overcome them, computational multiplexing techniques have recently been proposed, which allow multiple cellular structures to be captured in a single image and later be unmixed. Existing image decomposition methods are trained on a set of superimposed input images and the respective unmixed target images. It is critical to note that the relative strength (mixing ratio) of the superimposed images for a given input is a priori unknown. However, existing methods are trained on a fixed intensity ratio of superimposed inputs, making them not cognizant of the range of relative intensities that can occur in fluorescence microscopy. In this work, we propose a novel method called scSplit that is cognizant of the severity of the above-mentioned mixing ratio. Our idea is based on InDI, a popular iterative method for image restoration, and an ideal starting point to embrace the unknown mixing ratio in any given input. We introduce ($i$) a suitably trained regressor network that predicts the degradation level (mixing ratio) of a given input image and ($ii$) a degradation-specific normalization module, enabling degradation-aware inference across all mixing ratios. We show that this method solves two relevant tasks in fluorescence microscopy, namely image splitting and bleedthrough removal, and empirically demonstrate the applicability of scSplit on 5 public datasets. The source code with pre-trained models is hosted at https://github.com/juglab/scSplit/.

## 1 Introduction

Fluorescence microscopy is a widely utilized imaging technique in the life sciences, enabling researchers to visualize specific cellular and subcellular structures with high specificity. It employs distinct fluorescent markers to target different components, which are subsequently captured in separate image channels. The global fluorescence microscopy market, valued at 9.83$ billion in 2023, is projected to expand significantly in the coming years, reflecting its critical role in advancing biological research [1].

Still, there are practical limitations on the maximum number of structures that can be imaged in one sample. To mitigate this, the idea of imaging multiple structures into a single image channel has recently been gaining popularity [2, 3]. In such approaches, the image produced by the microscope is a superposition of multiple structures, and a deep-learning-based setup is then used to perform the image decomposition task, thereby yielding the constituent structures present in the superimposed input as separate images.

While these approaches have been beneficial, they have not explicitly addressed a particular aspect of this problem. The relative intensity of the superimposed structures in the input can vary significantly depending on sample properties, labeling densities, and microscope configuration. For instance, in a superimposed image of nuclei and mitochondria, nuclei may be dominant in their intensities,

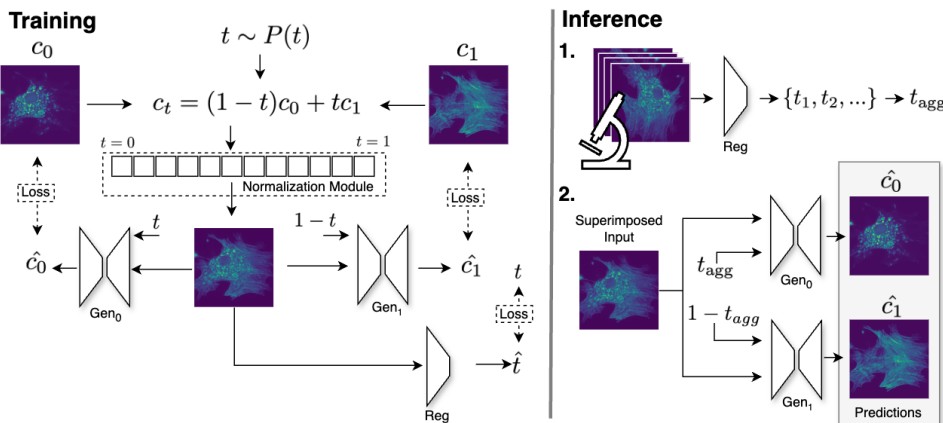

**Figure 1: Schematic overview of the scSplit framework for handling image superposition at varying severity levels. (Left)** Training pipeline: The input to the system is a superimposed image, generated as a weighted average of two images using a mixing ratio $t \in [0, 1]$. The superimposed image is passed through a normalization module, which performs ratio-specific normalization to ensure zero mean and unit standard deviation. The normalized image is then processed by two generative networks, $Gen_0$ and $Gen_1$, to estimate the individual structures. A regressor network, $Reg$, is trained to predict the mixing ratio $t$ from a normalized superimposed image. **(Right)** Inference pipeline: During inference, the mixing ratio $t$ is estimated for a set of superimposed input images using $Reg$, and the estimates are aggregated to obtain $t_{agg}$. The normalized superimposed images, along with $t_{agg}$, are fed into the generative networks $Gen_i$ to recover the individual structures. Thanks to the mixing-ratio specific normalization during training, the normalization during inference is simple and is performed using the mean and standard deviation computed from the set of test input patches.

with the mitochondria showing as relatively faint structures. Existing methods, which are not cognizant of such variations in superposition severity, exhibit performance degradation when applied to images with superposition characteristics different from those encountered during training. We highlight the significance of this issue of severity cognizance by noting that a related problem, known as Bleedthrough, exists in fluorescence microscopy. When imaging a biological structure into a dedicated channel, other structures can become visible due to insufficiently precise optical filtering. In such cases, we say that this other structure "bleeds through" into the currently imaged channel. See Sup. Fig. 1 for a pictorial description of the task at hand. Note, only if the challenge of relative intensity variation is effectively addressed will a single network ever be able to effectively solve both the image unmixing task and the bleedthrough removal task.

To address this, we propose scSplit, a method that incorporates the desired cognizance about the superposition severity directly into the inductive bias of the method itself. We leverage the inductive bias embedded in the training methodology of InDI [4], a popular image restoration method. For a given superimposed input containing structures A and B, scSplit first explicitly predicts a mixing ratio $t \in [0, 1]$, which quantifies the severity of the superposition, with $t = 0$ meaning that only A is visible in the input, while $t = 1$ conversely meaning that only structure B can be seen. scSplit then uses the estimated mixing ratio along with the superimposed input to predict estimates of A and B. To ensure that the network remains in-distribution for superimposed images with varying superposition severities, we introduce a Severity Cognizant Input Normalization (SCIN) module. This module not only addresses the normalization requirements but also simplifies the inference process, as we show in Section 3. Additionally, leveraging domain-specific knowledge from fluorescence microscopy, we incorporate an aggregation module that enhances the accuracy of the mixing ratio estimation during inference. By integrating these advancements into scSplit, we introduce a method designed to simultaneously address two critical tasks in fluorescence microscopy—image unmixing and bleedthrough removal—by being cognizant of the severity of the superposition.

## 2 Related Work

In the field of fluorescence microscopy, image decomposition techniques have recently gained significant attention for addressing the image unmixing problem. Seo et al. [5] introduced a linear unmixing approach to separate $k$ structures, which, however, necessitates $k$ input channels, each representing

a distinct superposition of the $k$ structures. More recently, deep learning-based frameworks have emerged [3, 2, 6], capable of predicting individual structures from a single image channel. Ashesh et al. [3] proposed a GPU-efficient meta-architecture, $\mu$Split, which leverages contextual information from surrounding regions of the input patch. HVAE [7–9] and U-Net [10] were used as the underlying architecture for $\mu$Split. Three variants of $\mu$Split were developed, each optimizing a trade-off between GPU utilization and performance. Further advancements were made with denoi$\mathbb{S}$plit [11], which combines unsupervised denoising with supervised image unmixing. More recently Micro$\mathbb{S}$plit [6] combined the GPU efficiency of $\mu$Split with unsupervised denoising, sampling, and calibration of denoi$\mathbb{S}$plit. It also provided several image unmixing datasets containing real microscopy images of different structure types. However, existing single-channel input methods [3, 11] typically assume the input to be an average of the two structures, thereby overlooking the variability in superposition intensity present in real-world microscopy images. While the Micro$\mathbb{S}$plit analysis successfully quantified the effects of superposition variability, it did not extend to proposing a resolution. This highlights the need for more robust approaches to handle the complexities of real imaging data.

Next, we situate the image unmixing task within the broader context of Computer Vision. Image unmixing can be viewed as a specialized form of image translation, where the objective is to map an image from a source data distribution to multiple images, each belonging to a specific target data distribution. Over the past decade, the field of image translation has witnessed significant advancements, with a wide array of methodologies being proposed. These include architectures such as U-Net [10], generative adversarial networks (GANs) [12–14], and iterative inference models like diffusion models [15, 16] and flow matching techniques [17–19], among others. These approaches have demonstrated remarkable capabilities in addressing various challenges in image-to-image transformation tasks, providing a rich foundation for advancing image unmixing techniques.

Iterative models offer the advantage of providing access to intermediate predictions during the inference process. In many such methods, these intermediate predictions—after accounting for noise—closely resemble a superposition of the source and target data distributions. Consequently, when the degradation process itself involves superposition, as is the case in our task, iterative models emerge as a natural choice for modeling the degradation. Literature suggests that the superposition of structures in fluorescence microscopy can be approximated as a linear superposition [20–22]. This insight led us to adopt InDI [4], a well-established iterative image restoration method that explicitly models degradation as a linear mixing process. In InDI, the idea is to take the weighted average between the clean target and the degraded input using a scalar mixing ratio to generate a 'less' degraded input. The generated input and the mixing ratio are then fed to a network as inputs, and the network is trained to predict the clean target. The inductive bias of this training framework aligns precisely with the linear superposition observed in fluorescence microscopy, making InDI a suitable foundation for our proposed approach.

Finally, we observe that within the broader domain of image translation, the task of image unmixing shares similarities with tasks such as reflection removal, dehazing, and deraining [23–26]. However, these tasks differ fundamentally from fluorescence microscopy unmixing in aspects like superposition linearity and ground truth availability. See Sup. Sec. J for more details.

## 3 Our Method

Here, we begin by establishing the necessary formal notation. Then, we address the limitations of existing normalization schemes when performing inference from intermediate timesteps, and present our improved normalization approach. Finally, we outline the training process, including the loss formulations for the generative networks ($Gen_0$ and $Gen_1$) and the regressor network ($Reg$), as illustrated in Figure 1.

### 3.1 Problem Definition

Let us denote a set of $k$ image pairs by $C = \{(c_0^1, c_1^1), (c_0^2, c_1^2), ..., (c_0^k, c_1^k)\}$. We denote by $C_0 = \{c_0^1, c_0^2, ...\}$ and $C_1 = \{c_1^1, c_1^2, ...\}$ the two sets of images from the two distributions of images we intend to learn to unmix. For brevity and readability, we will omit the superscript unless needed. For a pair of images ($c_0 \in C_0, c_1 \in C_1$) and a *mixing ratio* $t \in [0, 1]$, an input to be unmixed is defined by the pixel-wise linear combination

$$c_t = (1 - t)c_0 + tc_1. \tag{1}$$

With this notation at hand, we define the task of *Image Decomposition* as the computational unmixing of a given superimposed image $c_t$ into estimates $\hat{c}_0$ and $\hat{c}_1$. In this context, we introduced the term superposition severity to describe the dominance of one channel in the superimposed input, which is precisely quantified by the mixing ratio $t$.

The assessment of the quality of any solution for the image decomposition task is evaluated by computing the similarity between $\hat{c}_0$ and $\hat{c}_1$ to the true images $c_0$ and $c_1$, respectively.

### 3.2  Severity Cognizant Input Normalization (SCIN)

We begin by observing that the normalization procedure for the input patch has not received any special attention in the existing image unmixing works [3, 2, 6], and the standard practice of mean and standard deviation based normalization is performed, where the mean and standard deviation computation is done over the entire training data. With natural images, a common normalization strategy is to divide by 255. Such a data-independent normalization is not suitable for Fluorescence microscopy data, which is typically stored in the `uint16` data type, since intensity distributions vary significantly depending upon the imaging conditions. For instance, it is common to have the maximum pixel intensity for a noisy acquisition to be less than 200, whereas the pixel intensities can easily be larger than 20000 for less noisy acquisitions.

In InDI [4], where the input also has the same formulation as Eq. 1, $c_0$ and $c_1$ are separately normalized according to statistics derived from $C_0$ and $C_1$, respectively. It is worth noting that similar normalization schemes are commonly employed in iterative models in general that operate with two data distributions and have the objective of translating from one distribution to another.

To understand why a more involved normalization module is required, we next provide the expression for the expected mean $\mathbb{E}[\mu(t)]$ and the variance $\mathbb{E}[\sigma^2(t)]$ of a random superimposed patch $c_t$ given a mixing ratio $t$, with expectation computed over the set of patches $c_t$ for a given $t$. Please refer to the Sup. Sec. A for the derivation. Using Eq. 1, one can write $\mathbb{E}[\mu(t)] = (1-t)\mathbb{E}[p_0] + t\mathbb{E}[p_1]$ and

$$\mathbb{E}[\sigma^2(t)] = (1-t)^2\mathbb{E}[\sigma^2(0)] + t^2\mathbb{E}[\sigma^2(1)] + 2t(1-t)\text{Cov}(p_0, p_1), \tag{2}$$

where $p_0$ and $p_1$ denotes a random pixel from $c_0$ and $c_1$ respectively and $\text{Cov}[\cdot, \cdot]$ denotes the covariance. One of the default ways to do data normalization is to standardize the sets of images $C_0$ and $C_1$ to have zero mean ($\mathbb{E}[p_0] = \mathbb{E}[p_1] = 0$) and unit variance ($\mathbb{E}[\sigma^2(0)] = \mathbb{E}[\sigma^2(1)] = 1$). In this case, one obtains $\mathbb{E}[\mu(t)] = 0$ and $\mathbb{E}[\sigma^2(t)] = t^2 + (1-t)^2 + 2t(1-t)\text{Cov}[p_0, p_1]$. Note that while $\mathbb{E}[\mu(t)]$ is 0 for all $t$, the expected variance is a function of $t$. We support this claim with empirical evidence in the Sup. Fig. S.19. What this means is that during training, for a given mixing ratio $t$, the network sees the superimposed patches drawn from a distribution of images having zero mean and a standard deviation dependent on $t$. To get optimal performance on a superimposed input during inference, we would want the input image to be suitably normalized so as to have similar statistics. This leads to a critical complication when we want to do inference on $c_t$ with an unknown $t \in [0, 1]$ using a trained scSplit network. Without knowing $t$ for an input, we cannot normalize correctly.

The solution we propose is to avoid the problem altogether by introducing *Severity Cognizant Input Normalization Module (SCIN)*, ensuring that for every $t$, $\mathbb{E}[\mu(t)] = 0$ and $\mathbb{E}[\sigma^2(t)] = 1.0$. To enable this, we must first empirically evaluate what $\mathbb{E}[\mu(t)]$ and $\mathbb{E}[\sigma^2(t)]$ are for a partition of the interval $[0, 1]$. We split the interval $[0, 1]$ into $n = 100$ equally sized disjoint partitions. We generate $n$ sets of mixed image patches $C_i = \{c_t : t \in (\frac{i}{n}, \frac{i+1}{n}]\}, i \in [0, \ldots, n-1]$ by extracting image patches of fixed size from image sets $C_0$ and $C_1$ and performing pixelwise weighted average as in Equation 1. For each input patch in each set, we compute the mean and standard deviation. For each set, we aggregate these values to get the expected mean and standard deviation and store them in a list of tuples $D$, such that $D[i] = (\mu_i, \sigma_i)$. After creating a superimposed image patch $c_t$ as described in Eq. 1 during training, we standardize $c_t$ for all $t < 1$ using the mean and variance saved in $D[\lfloor tn \rfloor]$. The normalization module we propose simplifies inference by decoupling input normalization from the mixing ratio. During training, we enforce $\mathbb{E}[\mu(t)] = 0$ and $\mathbb{E}[\sigma^2(t)] = 1$ across all mixing ratios. During inference, test inputs from a single acquisition can therefore be normalized using the mean and variance computed directly from the images in that acquisition.

### 3.3 Network Setup

Similar to InDI [4], we use a Gaussian noise perturbation on the input. Let $c_t^{\text{norm}}$ denote the normalized $c_t$, with normalization done as described above. Input to the network becomes $x_t = c_t^{\text{norm}} + t\epsilon n$, with $n \sim \mathcal{N}(0, I)$ and $\epsilon = 0.01$.

**Generative Network $Gen_i$.** We use two generative networks, $Gen_0$ and $Gen_1$, to give us estimates of $c_0^{\text{norm}}$ and $c_1^{\text{norm}}$ respectively. As shown in Figure 1, they take as input the normalized superimposed image along with an estimate of the mixing ratio, which represents the severity of the unmixing to be done. Unmixed prediction for the channel $i \in \{0, 1\}$ can be expressed as $\hat{c}_i^{\text{norm}} = \text{Gen}_i(x_t, t\delta_i + (1-t)\delta_{1-i})$, where $\delta_k$ denotes Dirac delta. Note that the severity of the unmixing for estimating $c_0$ and $c_1$ is $t$ and $1 - t$, respectively.

**Regressor Network $Reg$ to Estimate the Right Mixing Ratio.** Our regression network $Reg$ predicts an estimate of the mixing ratio $t$ given an input $x_t$, which is then used by $Gen_i$ networks during inference. Crucially, $Reg$ incorporates the same normalization module proposed for the $Gen_i$ networks (Section 3.2) and the reasons are identical. As seen before, the normalization statistics for $x_t$ are inherently dependent on $t$. During inference, inputs must be normalized using statistics consistent with their true $t$ to avoid distributional mismatch. This creates a cyclic dependency: accurate regression of $t$ requires proper normalization, but normalization requires the knowledge of $t$. Resolving this interdependence is central to our framework's design.

Next, we utilize domain knowledge to further improve our estimations of $t$. We know that for all images acquired during a single session at a microscope, the same laser power settings and the same fluorophore types will be used. This means that the mixing ratio of all these images can be assumed to be the same. Hence, we aggregate the $t$ values estimated from the set of images belonging to a single session and use that during inference. The aggregation is implemented as a simple arithmetic mean of the $t$ values obtained for individual images in the session. In the Sup. Sec. H, we experiment with different aggregation methods.

**Distribution for p(t).** During training, we sample the mixing ratio $t$ from a distribution $p(t)$. To model $p(t)$, we modify the distribution denoted as '$linear_a$' in InDI, adapting it to

$$p(t) = \frac{1}{1+a}U[0,1] + \frac{a}{1+a}\delta_{0.5}, \tag{3}$$

with $a = 1$ in all our experiments. Unlike InDI, where more weight was given via the Dirac delta distribution to $t = 1$, we need more weight on $t = 0.5$. This is because the image unmixing task involves inputs containing both structures, making it more appropriate to assign a higher weight to $t = 0.5$ rather than $t = 1$.

## 4 Results

In all qualitative figures and tables, we define the input as $x_w = w * C_{\text{`wanted'}} + (1-w) * C_{\text{`other'}}$. This notation allows us to relate $w$ directly to the strength of the channel we are evaluating. The value of $w$ determines the nature of the prediction task. When $w = 0.1$, the objective is to predict the dim structure within the superimposed input. Conversely, when $w = 0.9$, the task shifts to identifying and removing the dim structure, effectively isolating the dominant structure. This latter scenario is commonly referred to as the bleed-through removal task in the field.

**Datasets and Unmixing Tasks.** We tackle five tasks coming from five real microscopy datasets, namely Hagen et al. [29], BioSR [30], HTT24 [6], HTLIF24 [6], and PaviaATN [3]. From the BioSR dataset, we tackle the ER *vs.* Microtubules task. From the Hagen et al., we tackle the Actin *vs.* Mitochondria task. Following $\mu$Split [3], we clip pixel values at 1993.0 for this dataset to have a fair comparison. From HTT24, we tackle the SOX2 *vs.* Golgi task. From HTLIF24, we choose the Microtubules *vs.* Centromere task and from PaviaATN the Actin *vs.* Tubulin task. The HTT24 and HTLIF24 datasets also include microscope-imaged superimposed inputs, which we additionally evaluate. Using the laser power ratios employed for the two structures in these datasets as a proxy for mixing ratio, we obtain $w = 0.5$ and $w = 0.41$ for HTT24 and HTLIF24, respectively.

| Dataset | | Dominant | | | Balanced | | | Weak | | |
|---|---|---|---|---|---|---|---|---|---|---|
| | | PSNR | SSIM | LPIPS | PSNR | SSIM | LPIPS | PSNR | SSIM | LPIPS |
| Hagen et. al | Inp vs Tar | 34.1 | 0.973 | 0.047 | 25.1 | 0.889 | 0.148 | 21.2 | 0.784 | 0.243 |
| | U-Net | 31.8 | 0.965 | 0.063 | 28.2 | 0.921 | 0.122 | 22.0 | 0.833 | 0.222 |
| | $\mu$Split$_L$ | 33.7 | 0.965 | 0.048 | 31.9 | 0.961 | 0.067 | 23.2 | 0.857 | 0.167 |
| | $\mu$Split$_R$ | 33.9 | 0.962 | 0.046 | 32.4 | 0.960 | 0.062 | 23.6 | 0.858 | 0.165 |
| | $\mu$Split$_D$ | 33.1 | 0.967 | 0.045 | 32.4 | 0.964 | 0.058 | 23.4 | 0.863 | 0.158 |
| | denoiSplit | 32.4 | 0.958 | 0.166 | 31.9 | 0.954 | 0.169 | 23.1 | 0.851 | 0.246 |
| | MicroSplit | 34.7 | 0.957 | 0.167 | 31.7 | 0.956 | 0.168 | 25.0 | 0.862 | 0.240 |
| | InDI | 33.1 | 0.963 | 0.043 | 32.1 | 0.965 | 0.052 | 24.2 | 0.879 | 0.138 |
| | scSplit$_{0.5}$ | 34.1 | 0.979 | 0.032 | 33.7 | 0.975 | 0.045 | 25.0 | 0.881 | 0.141 |
| | scSplit$_{-agg}$ | 40.6 | 0.994 | 0.011 | 33.3 | 0.976 | 0.046 | 28.0 | 0.929 | 0.123 |
| | scSplit | 40.9 | 0.994 | 0.011 | 33.9 | 0.977 | 0.046 | 29.3 | 0.934 | 0.123 |
| HTLIF24 | Inp vs Tar | 42.3 | 0.989 | 0.018 | 33.3 | 0.946 | 0.075 | 29.5 | 0.881 | 0.139 |
| | U-Net | 45.9 | 0.980 | 0.023 | 44.6 | 0.986 | 0.016 | 36.0 | 0.939 | 0.066 |
| | $\mu$Split$_L$ | 46.7 | 0.978 | 0.024 | 45.1 | 0.986 | 0.016 | 36.6 | 0.940 | 0.068 |
| | $\mu$Split$_R$ | 46.4 | 0.978 | 0.024 | 45.1 | 0.986 | 0.016 | 36.5 | 0.940 | 0.068 |
| | $\mu$Split$_D$ | 45.9 | 0.979 | 0.024 | 44.9 | 0.986 | 0.016 | 36.4 | 0.942 | 0.066 |
| | denoiSplit | 44.8 | 0.981 | 0.029 | 42.9 | 0.985 | 0.025 | 35.8 | 0.938 | 0.075 |
| | MicroSplit | 45.0 | 0.982 | 0.029 | 43.7 | 0.986 | 0.025 | 36.5 | 0.939 | 0.073 |
| | InDI | 45.2 | 0.976 | 0.031 | 43.9 | 0.991 | 0.012 | 37.6 | 0.963 | 0.055 |
| | scSplit$_{0.5}$ | 45.9 | 0.987 | 0.015 | 45.1 | 0.991 | 0.013 | 37.4 | 0.951 | 0.065 |
| | scSplit$_{-agg}$ | 50.1 | 0.997 | 0.003 | 44.0 | 0.993 | 0.010 | 38.8 | 0.975 | 0.035 |
| | scSplit | 51.8 | 0.998 | 0.002 | 45.5 | 0.994 | 0.009 | 39.9 | 0.976 | 0.034 |
| BioSR | Inp vs Tar | 33.9 | 0.937 | 0.119 | 24.1 | 0.746 | 0.311 | 21.1 | 0.504 | 0.498 |
| | U-Net | 37.2 | 0.924 | 0.066 | 33.7 | 0.958 | 0.059 | 25.6 | 0.740 | 0.292 |
| | $\mu$Split$_L$ | 37.8 | 0.918 | 0.066 | 33.5 | 0.959 | 0.051 | 25.7 | 0.738 | 0.291 |
| | $\mu$Split$_R$ | 37.8 | 0.921 | 0.060 | 33.0 | 0.960 | 0.049 | 25.7 | 0.748 | 0.276 |
| | $\mu$Split$_D$ | 37.5 | 0.915 | 0.070 | 32.6 | 0.956 | 0.059 | 25.2 | 0.744 | 0.278 |
| | denoiSplit | 36.4 | 0.929 | 0.083 | 33.1 | 0.957 | 0.086 | 25.3 | 0.733 | 0.322 |
| | MicroSplit | 38.5 | 0.932 | 0.068 | 34.3 | 0.966 | 0.065 | 26.6 | 0.759 | 0.274 |
| | InDI | 35.9 | 0.917 | 0.054 | 33.4 | 0.953 | 0.050 | 26.3 | 0.802 | 0.211 |
| | scSplit$_{0.5}$ | 37.3 | 0.957 | 0.033 | 35.0 | 0.967 | 0.037 | 26.4 | 0.770 | 0.236 |
| | scSplit$_{-agg}$ | 39.3 | 0.986 | 0.012 | 33.7 | 0.965 | 0.039 | 27.2 | 0.868 | 0.153 |
| | scSplit | 40.1 | 0.987 | 0.011 | 35.3 | 0.973 | 0.033 | 28.7 | 0.889 | 0.130 |
| HTT24 | Inp vs Tar | 38.7 | 0.978 | 0.015 | 29.6 | 0.900 | 0.075 | 25.8 | 0.783 | 0.149 |
| | U-Net | 37.9 | 0.963 | 0.042 | 37.5 | 0.965 | 0.020 | 30.1 | 0.883 | 0.059 |
| | $\mu$Split$_L$ | 37.3 | 0.953 | 0.046 | 36.6 | 0.959 | 0.021 | 29.7 | 0.880 | 0.059 |
| | $\mu$Split$_R$ | 37.6 | 0.954 | 0.046 | 36.9 | 0.959 | 0.021 | 29.9 | 0.880 | 0.059 |
| | $\mu$Split$_D$ | 37.5 | 0.954 | 0.045 | 36.8 | 0.960 | 0.021 | 29.8 | 0.880 | 0.059 |
| | denoiSplit | 37.3 | 0.954 | 0.055 | 37.5 | 0.964 | 0.028 | 31.0 | 0.896 | 0.062 |
| | MicroSplit | 36.9 | 0.950 | 0.061 | 36.6 | 0.959 | 0.032 | 30.3 | 0.891 | 0.063 |
| | InDI | 37.6 | 0.962 | 0.034 | 36.5 | 0.966 | 0.017 | 30.5 | 0.909 | 0.057 |
| | scSplit$_{0.5}$ | 38.1 | 0.984 | 0.018 | 38.6 | 0.979 | 0.008 | 31.4 | 0.902 | 0.063 |
| | scSplit$_{-agg}$ | 43.4 | 0.993 | 0.002 | 38.2 | 0.979 | 0.007 | 33.7 | 0.939 | 0.030 |
| | scSplit | 44.5 | 0.995 | 0.001 | 39.1 | 0.981 | 0.005 | 34.7 | 0.943 | 0.028 |
| PaviaATN | Inp vs Tar | 31.0 | 0.932 | 0.104 | 22.3 | 0.754 | 0.294 | 18.2 | 0.548 | 0.480 |
| | U-Net | 29.3 | 0.870 | 0.210 | 25.4 | 0.743 | 0.346 | 21.2 | 0.568 | 0.500 |
| | $\mu$Split$_L$ | 27.0 | 0.889 | 0.133 | 24.3 | 0.780 | 0.241 | 21.1 | 0.622 | 0.396 |
| | $\mu$Split$_R$ | 27.4 | 0.905 | 0.120 | 24.7 | 0.800 | 0.228 | 21.1 | 0.639 | 0.387 |
| | $\mu$Split$_D$ | 27.9 | 0.908 | 0.127 | 25.2 | 0.808 | 0.241 | 21.3 | 0.648 | 0.399 |
| | denoiSplit | 27.3 | 0.857 | 0.772 | 26.2 | 0.843 | 0.794 | 21.8 | 0.750 | 0.824 |
| | MicroSplit | 24.0 | 0.771 | 0.896 | 21.8 | 0.688 | 0.952 | 18.8 | 0.540 | 1.000 |
| | InDI | 29.9 | 0.943 | 0.131 | 23.9 | 0.858 | 0.192 | 21.7 | 0.741 | 0.248 |
| | scSplit$_{0.5}$ | 29.0 | 0.948 | 0.082 | 27.1 | 0.904 | 0.135 | 21.2 | 0.774 | 0.226 |
| | scSplit$_{-agg}$ | 33.9 | 0.976 | 0.035 | 27.1 | 0.903 | 0.166 | 23.9 | 0.819 | 0.387 |
| | scSplit | 35.1 | 0.977 | 0.033 | 27.6 | 0.907 | 0.155 | 24.3 | 0.823 | 0.377 |

**Table 1: Quantitative evaluation of unmixing performance across five tasks.** We categorize the input into three regimes based on the dominance of the target channel: *dominant* ($w \in \{0.9, 0.8, 0.7\}$), *balanced* ($w \in \{0.6, 0.5, 0.4\}$), and *weak* ($w \in \{0.3, 0.2, 0.1\}$). The reported metric values are averaged across both channels and all values of $w$ within each regime. To account for the varying difficulty of the regimes, we include a comparison between the input and the target in the first row for each dataset. Metrics include Multiscale SSIM (MS-SSIM) [27] and range-invariant PSNR [28]. The grayed and underlined entries indicate the best and second-best results for each metric, respectively.

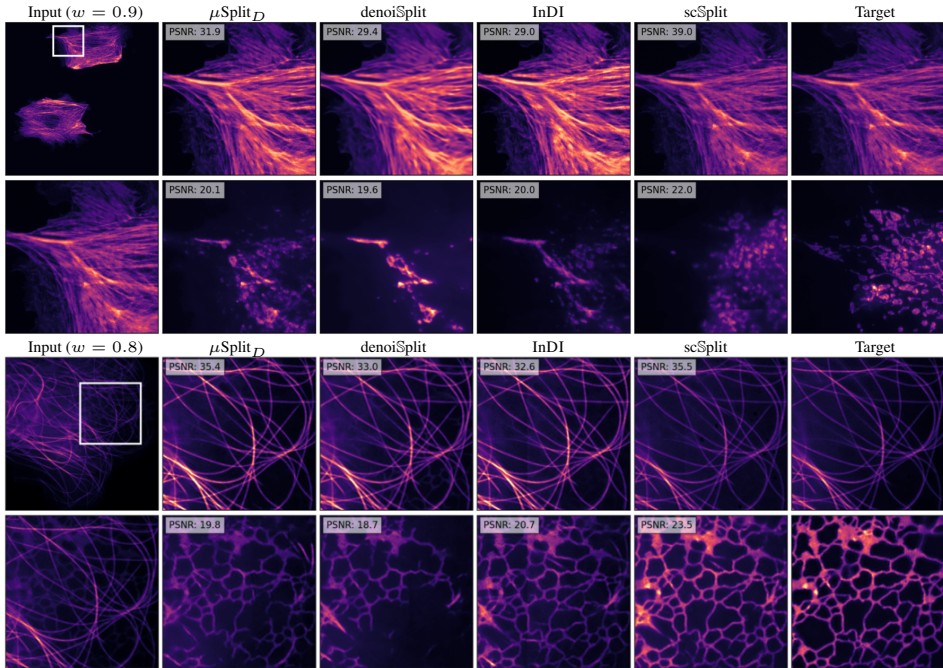

**Figure 2: Qualitative evaluation of unmixing performance.** We show qualitative evaluation on Hagen et al. [29] (top panel) and BioSR [30] (bottom panel). For each dataset, the full input frame (top-left) and a zoomed-in input patch (bottom-left) are displayed. Predictions and corresponding targets (last column) are for the input patch. They are shown for both channels, with each channel displayed in a separate row. PSNR values are also reported for the predicted patch. The mixing ratio $w$ indicated above the input column corresponds to the first channel, with the second channel naturally having a ratio of $(1 - w)$. Additional qualitative evaluations across different $w$ values for all datasets are provided in the Supplementary Figures 4 through 18.

|  | HT-LIF24 | | | HT-T24 | | |
|---|---|---|---|---|---|---|
|  | PSNR | SSIM | LPIPS | PSNR | SSIM | LPIPS |
| Inp vs Tar | 32.9 | .947 | .193 | 29.4 | .894 | .152 |
| U-Net | 40.7 | .990 | .021 | 35.6 | .955 | .015 |
| $\mu$Split$_L$ | 40.6 | .990 | .022 | 35.0 | .949 | .017 |
| $\mu$Split$_R$ | 40.9 | .991 | .021 | 35.1 | .950 | .017 |
| $\mu$Split$_D$ | 40.9 | .991 | .021 | 35.2 | .950 | .017 |
| denoiSplit | 39.8 | .988 | .032 | 36.6 | .961 | .030 |
| MicroSplit | 40.1 | .991 | .034 | 35.8 | .954 | .031 |
| InDI | 41.2 | .992 | .012 | 34.4 | .946 | .037 |
| scSplit$_{0.5}$ | 41.1 | .993 | .016 | 35.9 | .956 | .017 |
| scSplit$_{-agg}$ | 40.5 | .991 | .015 | 35.6 | .955 | .019 |
| scSplit | 40.9 | .992 | .015 | 36.0 | .957 | .017 |

**Table 2: Evaluation on superimposed raw microscopy images.** For the HT-LIF24 and HT-T24 datasets, we evaluate raw superimposed images with roughly balanced channel intensities. The metric 'SSIM' refers to MicroMS3IM [31]. Best and second-best results per metric are indicated by grayed and underlined entries, respectively.

| Model | Mixing-ratio | PSNR | SSIM | LPIPS |
|---|---|---|---|---|
| UNet | (4:1) | 30.1 | 0.906 | 0.074 |
| denoiSplit | (4:1) | 28.8 | 0.854 | 0.096 |
| scSplit$_{0.5}$ | (4:1) | 32.3 | 0.914 | 0.069 |
| scSplit ($t_{agg} = .82$) | (4:1) | 35.8 | 0.956 | 0.020 |
| UNet | (1:4) | 28.5 | 0.878 | 0.109 |
| denoiSplit | (1:4) | 30.3 | 0.920 | 0.115 |
| scSplit$_{0.5}$ | (1:4) | 32.3 | 0.927 | 0.067 |
| scSplit ($t_{agg} = .25$) | (1:4) | 35.6 | 0.955 | 0.020 |

**Table 3: Evaluation on superimposed raw microscopy images exhibiting increased train-test distribution shift**. We use the HT-T24 dataset, with the test set and metrics identical to those in Table 2. However, relative to Table 2, the training data exhibits a substantially different mixing ratio from the test set, resulting in a noticeable drop in the performance of the strongest baseline methods. Despite this, scSplit still demonstrates comparatively robust performance in these conditions.

**Baselines.** As a baseline, we use U-NET [10], with the implementation as used in [3]. Next, we use the three architectures proposed in $\mu$Split [3], namely Lean-LC, Regular-LC, and Deep-LC, as baselines which we refer to as $\mu$Split$_L$, $\mu$Split$_R$, and $\mu$Split$_D$, respectively. For the Hagen et al. [29] and the PaviaATN [3] data, we used publicly available pretrained models for the abovementioned baselines. Next, we use the official implementation of denoiSplit as a baseline where we increased the patch size (from 128 to 512) to ensure a fair comparison with scSplit. We use the official implementation of MicroSplit with its default hyperparameters. Finally, we use InDI as a baseline. Since there is no available official implementation for InDI, we implemented it. To ensure a fair comparison, we utilized the same hyperparameters for both InDI and scSplit implementations. For

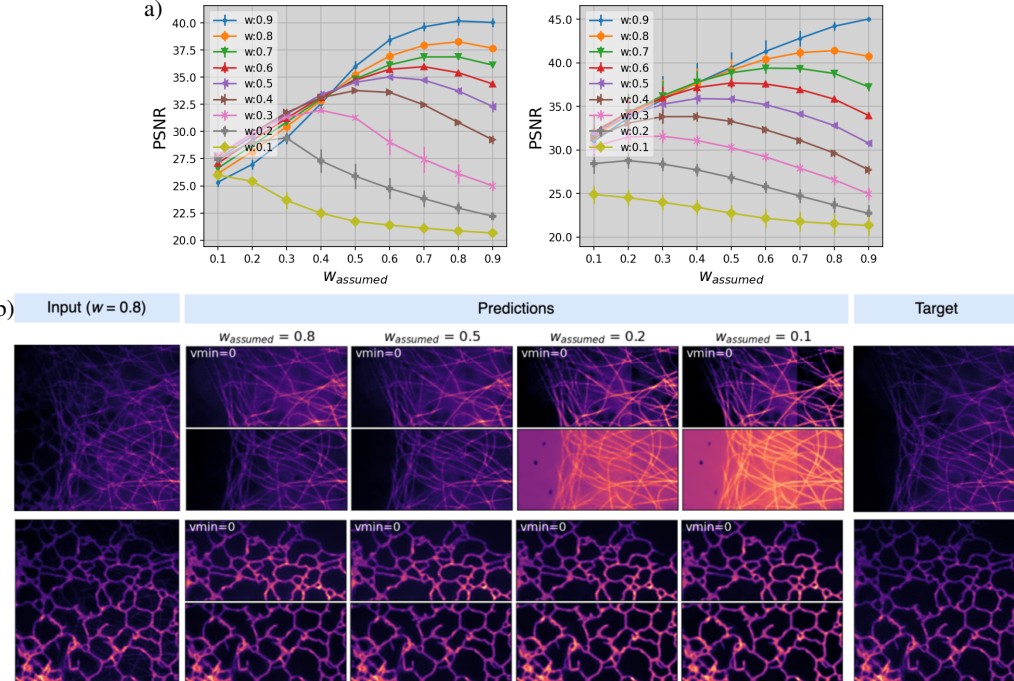

**Figure 3: Performance evaluation of the regressor network *Reg*.** (a) *Quantitative evaluation of performance degradation with incorrect $w$ during inference:* This analysis highlights the sensitivity to inaccurate predictions by *Reg*. The x-axis represents the assumed $w$ during inference, the legend represents the actual $w$, and the y-axis quantifies the performance. (b) *Qualitative evaluation on the BioSR dataset with varying $w$ during inference:* Predictions are shown for each channel (two rows) under different assumed $w$ values. For each $w$, the input is divided into upper and lower halves, displayed in two sub-rows. The first sub-row sets negative pixel values to zero during visualization, while the second sub-row uses default visualization. It is worth noting that significant artefacts (tiling artifacts, disappearance of structures, and increased "crispness" of microtubule curves) get manifested when the assumed $w$ is reasonably far from $w = 0.8$, the $w$ value used to create the input.

our InDI baseline, we use $t = 0.5$ during inference and use the $p(t)$ defined in Eq. 3 during training. Please refer to the supplement for more details on these baselines. To evaluate the models on superimposed images directly captured with a microscope, we trained all models with synthetic inputs and used these 'real' inputs for evaluation. To separately showcase the benefits of the *Reg* network and the aggregation operation, we have two variants of scSplit, namely scSplit$_{0.5}$ and scSplit$_{-agg}$ as additional baselines. scSplit$_{0.5}$ does not use the *Reg* network, and instead always uses $t = 0.5$ during inference. scSplit$_{-agg}$ uses the *Reg* network, but does not perform aggregation of estimated $t$. Finally, for InDI, scSplit, and all variants of scSplit, we employ one-step inference to minimize distortion. For a more detailed discussion on it, please see Sup. Sec. D.

**Quantitative Evaluation.** In Table 1, we present quantitative results. Here, we consider three input regimes, namely '*Dominant*', '*Balanced*', and '*Weak*', each differing from the other on the strength of the structure we are interested in. While in the *Weak* regime, the structure we desire to extract from the input is barely present, the desired structure is dominant in inputs from the '*Dominant*' regime. Within the lexicon of microscopy, inputs derived from the Strong regime are designated as exhibiting 'bleedthrough'. For each regime, we average the performance over the two channels and the 3 different $w$ values as mentioned in Table 1. Please refer to the Sup. Sec. E for more details on the evaluation procedures.

From Table 1, it is clear that scSplit does a good job across all input regimes, and especially for the *Dominant* regime, that is, for higher $w$ values. This result shows that a single trained scSplit network, which is cognizant of the severity of superposition, can solve both the bleedthrough removal task and image unmixing task.

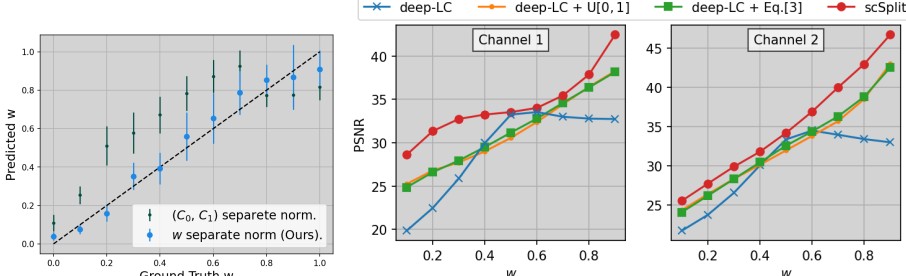

**Figure 4: Analysis of factors contributing to the superior performance of scSplit.** (Left) *Justification for the normalization module on the Hagen et al. dataset:* The conventional normalization of $C_0$ and $C_1$ leads to out-of-distribution issue during test evaluation. In contrast, our $w$-specific normalization scheme demonstrates superior performance. (Right) *Comparison against suitable augmentations:* We investigate one key factor behind scSplit's enhanced performance: its exposure to inputs with varying levels of mixing during training. For this, we introduce two augmentations to the input generation process of $\mu\text{Split}_D$ [3], one of our baselines, allowing it to also observe different mixing levels during training. While these augmented $\mu\text{Split}_D$ variants show improved performance over the vanilla $\mu\text{Split}_D$, scSplit consistently outperforms them across mixing ratios $w$.

As mentioned above, we also have real superimposed images for the balanced regime in the HTT24 and HTLIF24 datasets. We show the quantitative evaluation in Table 2. Although all baseline methods are optimized for the balanced regime and these real input images also belong to that regime, scSplit achieves competitive performance even under these conditions, as evidenced by the results in Table 2. To address potential concerns that scSplit may offer limited advantages for real superimposed images, we conduct an additional experiment on the HTT24 dataset to demonstrate its effectiveness. In this experiment, we demonstrate that scSplit outperforms baseline methods when the relative intensities of structures in real superimposed inputs differ from those in synthetic sums. From the HTT24 dataset, we created two variants by multiplying all pixel values in one structural channel by a factor of 4. In the first variant, the first channel is brighter than the second, while the opposite holds for the second variant. As a result, the mixing ratios between the test set and the synthetically summed inputs (used during training) differ significantly. For each variant, scSplit and the two top baselines (U-Net for LPIPS, denoiSplit for PSNR) were trained and evaluated on the unaltered real superimposed images. The results in Table 3 show that scSplit consistently outperforms both baselines by a substantial margin in PSNR across both variants, demonstrating robustness to variations in relative structure intensity.

**Effectiveness on a Downstream Segmentation Task.** For the bleed-through removal task ($w \in 0.7, 0.8, 0.9$), we used the test set of the BioSR dataset to evaluate segmentation as a downstream task with Featureforest (`max_depth=9`, `numtrees=450`, `encoder=SAM2_large`), a recently developed segmentation method [32]. To avoid model bias, we manually annotated the ground truth and trained Featureforest using these annotations (as target) alongside model predictions (as input). If we had annotated the predictions instead, it would raise concerns about the consistency and quality of annotations when doing it for scSplit's predictions as opposed to when doing the same for the baseline models. We trained $4 \times 2 \times 3 = 24$ Featureforest models—one per model, channel, and mixing ratio. We show the results in Table 4, where the evaluation used the Dice dissimilarity from scipy (lower is better). Among all methods, segmentations from scSplit predictions best matched those of the ground truth channel.

| Model | Channel 1 | | | Channel 2 | | |
|---|---|---|---|---|---|---|
| | $w = 0.7$ | $w = 0.8$ | $w = 0.9$ | $w = 0.7$ | $w = 0.8$ | $w = 0.9$ |
| denoiSplit | 0.077 | 0.073 | 0.065 | 0.064 | 0.061 | 0.056 |
| $\mu$Split | 0.055 | 0.051 | 0.047 | 0.051 | 0.047 | 0.038 |
| InDI | 0.055 | 0.050 | 0.044 | 0.046 | 0.045 | 0.045 |
| scSplit | 0.048 | 0.039 | 0.035 | 0.038 | 0.032 | 0.026 |

**Table 4:** DICE dissimilarity scores for various models and mixing ratios. scSplit achieves the best (lowest) DICE dissimilarity across all settings.

**Utility of SCIN.** Its utility can be inferred by comparing InDI with our scSplit$_{0.5}$ variant in Table 1, with scSplit$_{0.5}$ outperforming by 1.2db PSNR on average across all regimes and all four datasets. Note that all hyperparameters for the InDI and $Gen_i$ networks of scSplit$_{0.5}$ are identical, and the only difference is in the normalization. Please find the ablation for the normalization module in Sup. Sec. C. One can also note its utility by comparing the performance of *Reg* network trained with or without our normalization scheme in Figure 4(left). We also present a comparison between SCIN and Instance Normalization in Sup. Sec. B, as the latter can likewise address the issue we identified.

**Utility of Aggregation and *Reg* network.** Across Tables 1 and 2 scSplit outperforms the ablated network scSplit$_{-agg}$ on all tasks for the PSNR metric, thereby clearly justifying the utility of the aggregation operation. See Sup. Sec. H for experiments with other aggregation methodologies. One can observe the utility of using *Reg* when comparing scSplit$_{0.5}$ with scSplit$_{-agg}$, with the latter variant outperforming the former across several tasks in Table 1. It is worth noting that the improvement is more pronounced with more asymmetric mixing ratios (*Dominant* and *Weak* regimes). We argue that scSplit$_{0.5}$'s assumption of $w = 0.5$ becomes reasonable in the *Balanced* input regime, leading to its competitive performance in this regime.

**Degradation Analysis for *Reg*.** In Figure 3(a), with the BioSR dataset, we analyze the performance degradation when using increasing incorrect estimates for $w$ during inference. We evaluate scSplit using a fixed $w$ (x-axis) in the inference, while the inputs have been created with a different $w$ (legend). We find scSplit to be relatively stable to small differences between the assumed $w$ and actual $w$. We support this claim qualitatively in Figure 3(b).

**Exploring Augmentations.** One of the critical advantages scSplit and InDI have over other baselines is that during training, they observe inputs with different mixing ratios, and so naturally, it makes it easier for them to outperform them. InDI, however, cannot leverage this advantage since it does not have the *Reg* network and is therefore forced to use $t = 0.5$. In this experiment, we attempted to give this advantage to the baselines through augmentations. We experimented with two different augmentations in the training procedure of $\mu$Split$_D$, the most powerful variant of $\mu$Split. During training $\mu$Split$_D$, instead of creating the input $inp$ by simply summing the two channel images, we instead compute $inp = tc_0 + (1 - t)c_1$, where $t$ is sampled from $p(t)$. We work with two variants of $p(t)$: (i) $p(t) = U[0, 1]$ and (ii) $p(t)$ as defined in Eq. 3. Results shown in Figure 4 demonstrate that these augmentations help $\mu$Split$_D$ to improve performance for $w$ further away from $0.5$, along with some performance degradation for $w = 0.5$. However, scSplit still consistently outperforms the best of all three variants by *2.4db* PSNR on average. We note that the suboptimal normalization settings for $\mu$Split$_D$ also contribute to this, which is analyzed in the Sup. Fig. S.3.

## 5 Conclusion, Limitations, and Future Directions

In this work, we introduce scSplit, a network designed to simultaneously address two key challenges in fluorescence microscopy: image unmixing and bleedthrough removal. Our architecture is explicitly designed to account for the severity of the superposition that needs to be unmixed. We also identify limitations in the normalization methodologies of existing image unmixing approaches and propose an alternative normalization strategy that is better suited for inputs with varying levels of superposition. Additionally, we developed an aggregation module that improves the estimation of mixing ratios.

Despite its strengths, scSplit has a couple of limitations that warrant future investigation. First, the current framework is not optimized for noisy data—though a two-step approach (self-supervised denoising followed by unmixing) can be a suitable approach, an end-to-end solution could offer significant advantages. Second, extending scSplit to handle more than two channels would broaden its applicability. Finally, interference-based imaging modalities may challenge the linear superposition assumption of input formation, and therefore the effectiveness of scSplit and other semantic unmixing methods [6, 3, 2] on such modalities remains to be evaluated.

While scSplit is designed for fluorescence microscopy, its core principle of bringing severity cognizance into the inductive bias of the network could benefit general image restoration. In Sup. Sec. J, we do a proof-of-concept experiment to show that our approach can enhance motion deblurring performance. However, adapting scSplit to natural images would require integrating task-specific inductive biases—an exciting direction for future work.

## Acknowledgments

The authors thank Mauricio Delbracio and Peyman Milanfar from Google Research for useful discussions and technical guidance. This work was supported by the European Commission through the Horizon Europe program (IMAGINE project, grant agreement 101094250-IMAGINE) as well as core funding of Fondazione Human Technopole. We want to thank the IT and HPC team at Human Technopole for access to their compute infrastructure and all members of the Jug Group and our image analysis facility NoBIAS for helpful discussions and support.

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

# scSplit: Bringing Severity Cognizance to Image Decomposition in Fluorescence Microscopy

Ashesh Ashesh, Florian Jug
Jug Group, Human Technopole, Milano, Italy

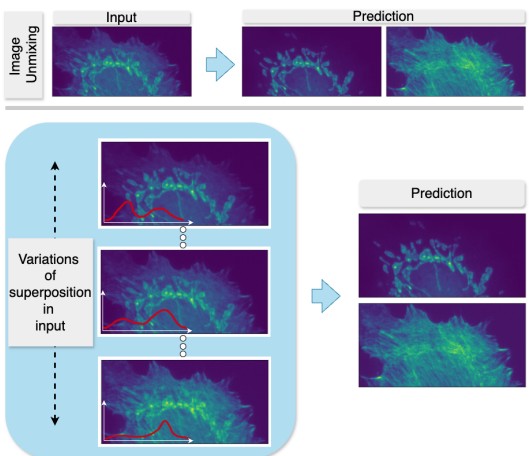

**Figure S.1: Handling varying levels of superposition.** For the objective of image unmixing task, superimposed images acquired with Fluorescence microscopy can have varying levels of mixing of the constituent structures. Additionally, insufficiently precise optical filtering often leads to 'bleedthrough' wherein a structure of interest gets superimposed with a shadowed presence of another structure. scSplit uniquely addresses these varying levels of structural mixing in the superimposed input images. Unlike existing unmixing methods, scSplit's architecture adapts to different degrees of superposition and accounts for the resulting variations in pixel intensity distributions (inset plot in red), enabling effective input image normalization and leading to efficient unmixing across diverse mixing ratios.

## A   Severity Cognizant Input Normalization

In this section, we extend the formulations presented in Section 3.2 to accommodate images of arbitrary dimensions $H \times W$. Let $p_t[i, j]$ be a random variable denoting a pixel intensity value present in $c_t \in C_t$ at the location $(i, j)$. Let us now compute the mean ($\mu(t)$) and variance ($\sigma^2(t)$) of $c_t$.

$$\mu(t) = \frac{1}{P} \sum_{i,j} p_t[i, j], \tag{1}$$

and

$$\sigma^2(t) = \frac{1}{P} \sum_{i,j} p_t[i, j]^2 - \mu(t)^2, \tag{2}$$

where $P$ is the total number of pixels in $c_t$. Note that $\mu(t)$ and $\sigma(t)$ are also random variables. Their expected values $\mathbb{E}[\mu(t)]$ and $\mathbb{E}[\sigma(t)]$ are typically used for normalization.

Next, describing $\mu(t)$ in terms of random variables $p_0$ and $p_1$, we get,

$$
\begin{aligned}
\mu(t) &= \frac{1}{P} \sum_{i,j} ((1 - t)p_0[i, j] + tp_1[i, j]) \\
&= (1 - t)\frac{1}{P} \sum_{i,j} p_0[i, j] + t\frac{1}{P} \sum_{i,j} p_1[i, j] \\
&= (1 - t)\mu(0) + t\mu(1).
\end{aligned}
\tag{3}
$$

Taking the expectation in the above equation, we get

$$\mathbb{E}[\mu(t)] = (1 - t)\mathbb{E}[\mu(0)] + t\mathbb{E}[\mu(1)]. \tag{4}$$

Doing a similar analysis for the variance, we get,

$$\mathbb{E}[\sigma^2(t)] = \frac{1}{P}\sum_{i,j}\mathbb{E}[p_t[i,j]^2] - \mathbb{E}[\mu(t)]^2. \tag{5}$$

Next, we simplify $\mathbb{E}[p_t[i,j]^2]$ and $\mathbb{E}[\mu(t)]^2$ to get

$$\begin{aligned}
\mathbb{E}[p_t^2[i,j]] &= \mathbb{E}[(1-t)^2 p_0^2[i,j] + t^2 p_1^2[i,j] \\
&\quad + 2t(1-t)p_0[i,j]p_1[i,j]] \\
&= (1-t)^2\mathbb{E}[p_0^2[i,j]] + t^2\mathbb{E}[p_1^2[i,j]] \\
&\quad + 2t(1-t)\mathbb{E}[p_0[i,j]p_1[i,j]], \text{ and}
\end{aligned} \tag{6}$$

$$\begin{aligned}
\mathbb{E}[\mu(t)]^2 &= (1-t)^2\mathbb{E}[\mu(0)]^2 + t^2\mathbb{E}[\mu(1)]^2 \\
&\quad + 2t(1-t)\mathbb{E}[\mu(0)]\mathbb{E}[\mu(1)].
\end{aligned} \tag{7}$$

Using expressions derived above, the expression for $\mathbb{E}[\sigma(t)]$ becomes,

$$\begin{aligned}
\mathbb{E}[\sigma^2(t)] = {}&(1-t)^2\left(\frac{1}{P}\sum_{i,j}\mathbb{E}[p_0^2[i,j]] - \mathbb{E}[\mu(0)]^2\right) \\
&+ t^2\left(\frac{1}{P}\sum_{i,j}\mathbb{E}[p_1^2[i,j]] - \mathbb{E}[\mu(1)]^2\right) \\
&+ 2t(1-t)\left(\frac{1}{P}\sum_{i,j}\mathbb{E}[p_0[i,j]p_1[i,j]] - \mathbb{E}[\mu(0)]\mathbb{E}[\mu(1)]\right)
\end{aligned} \tag{8}$$

With biological data, the image acquisition process works in the following way: a random location on the specimen slide is picked and that is then captured to generate an image frame. So, all pixel locations can be assumed to be equally likely to capture any part of any structure. We therefore assume, $\mathbb{E}[p_t[a,b]] = \mathbb{E}[p_t[c,d]] \; \forall a,b,c,d$ and $\mathbb{E}[p_t^2[a,b]] = \mathbb{E}[p_t^2[c,d]] \; \forall a,b,c,d$. So, we remove the pixel location and define $p_t$ to be a random variable denoting a pixel in $c_t$. We define $\mathbb{E}[p_t] := \mathbb{E}[p_t[a,b]] \; \forall a,b$ and $\mathbb{E}[p_t^2] := \mathbb{E}[p_t^2[a,b]] \; \forall a,b$. The expression for $\mathbb{E}[\sigma^2(t)]$ now simplifies to,

$$\begin{aligned}
\mathbb{E}[\sigma^2(t)] = {}&(1-t)^2\left(\frac{1}{P}\sum_{i,j}\mathbb{E}[p_0^2] - \mathbb{E}[\mu(0)]^2\right) \\
&+ t^2\left(\frac{1}{P}\sum_{i,j}\mathbb{E}[p_1^2] - \mathbb{E}[\mu(1)]^2\right) \\
&+ 2t(1-t)\left(\frac{1}{P}\sum_{i,j}\mathbb{E}[p_0 p_1] - \mathbb{E}[\mu(0)]\mathbb{E}[\mu(1)]\right) \\
= {}&(1-t)^2(\mathbb{E}[p_0^2] - \mathbb{E}[\mu(0)]^2) + t^2(\mathbb{E}[p_1^2] - \mathbb{E}[\mu(1)]^2) \\
&+ 2t(1-t)(\mathbb{E}[p_0 p_1] - \mathbb{E}[\mu(0)]\mathbb{E}[\mu(1)]) \\
= {}&(1-t)^2\mathbb{E}[\sigma^2(0)] + t^2\mathbb{E}[\sigma^2(1)] + 2t(1-t)\text{Cov}(p_0, p_1),
\end{aligned} \tag{9}$$

where $\text{Cov}(:,:)$ is the covariance function. Now that we have the expressions for $\mathbb{E}[\mu(t)]$ and $\mathbb{E}[\sigma^2(t)]$, we consider a plausible normalization methodology where $c_0$ and $c_1$ are normalized using the following procedure: for every $c_0$ in $C_0$, we compute its mean and standard deviation. We average the mean and standard deviation values computed over all images from $C_0$ to obtain a global mean and a global standard deviation. We use them to normalize every $c_0$ image. An identical procedure is followed for $c_1$. In this case, by construction $\mathbb{E}[\mu(0)] = \mathbb{E}[\mu(1)] = 0$ and $\mathbb{E}[\sigma^2(0)] = \mathbb{E}[\sigma^2(1)] = 1$. So, from Eq. 4, $\mathbb{E}[\mu(t)] = 0$. However, $\mathbb{E}[\sigma(t)]$ still remains the following function of $t$,

$$\mathbb{E}[\sigma^2(t)] = (1-t)^2 + t^2 + 2t(1-t)\text{Cov}(p_0, p_1) \tag{10}$$

As discussed in the main manuscript, this causes serious issues during normalization.

# B  Instance Normalization: Another Way to Achieve Our Normalization Objectives

If one thinks about the normalization requirements we discovered in this work, one realizes that applying instance normalization (IN) in the first layer can, in principle, also mitigate the normalization issue we discovered for the semantic unmixing task. Instead of making expected mean to zero and expected standard deviation to one as in our approach, it will ensure that for every input patch coming out from IN layer, its mean and standard deviation are zero and one respectively. It is simpler and therefore makes up for an attractive alternative approach. However, we found both empirical and domain-specific reasons led us to have the SCIN normalization approach we proposed in the main text.

For the empirical evaluation, we conducted experiments on the Hagen et al. dataset [29] using instance normalization. We trained two additional baselines: (1) InDI (firstIN), with instance normalization as the first layer, and (2) InDI (allIN), replacing all group normalization layers with instance normalization plus a first-layer instance normalization. Similar variants were also created for the scSplit network. Results presented in the Supplementary Table S.1 show our original scSplit model outperforms most baselines. Among 'firstIN' and 'allIN' variants, the 'firstIN' variant performs better than the 'allIN' variant. Results for the PaviaATN task with 'firstIN' are also reported in which case as well, scSplit has clear out-performance for the bleedthrough regime (Dom.) and is competitive in others. More specifically, scSplit outperforms scSplit (firstIN) across three input regimes and two datasets, with an average PSNR gain of 0.33 dB. Since both models share the same network architecture and incorporate our key innovations—(a) the Reg network, (b) aggregation of t, and (c) correction of the normalization issue—the performance gap could not be large and thus we consider this performance gap as an evidence of superiority of our normalization module over IN.

| Dataset | Model | Dom. | | | Bal. | | | Weak. | | |
|---|---|---|---|---|---|---|---|---|---|---|
| | | PSNR | SSIM | LPIPS | PSNR | SSIM | LPIPS | PSNR | SSIM | LPIPS |
| Hagen et al. | InDI (allIN) | 36.5 | 0.987 | 0.026 | 32.4 | 0.968 | 0.060 | 27.9 | 0.914 | 0.132 |
| Hagen et al. | InDI (firstIN) | 36.7 | 0.987 | 0.032 | 33.5 | 0.974 | 0.052 | 26.6 | 0.881 | 0.154 |
| Hagen et al. | scSplit (allIN) | 36.6 | 0.988 | 0.025 | 32.4 | 0.967 | 0.061 | 27.9 | 0.916 | 0.131 |
| Hagen et al. | scSplit (firstIN) | 40.1 | 0.993 | 0.016 | **34.0** | 0.976 | 0.051 | 29.0 | 0.924 | 0.129 |
| Hagen et al. | scSplit | **40.9** | **0.994** | **0.011** | 33.9 | **0.977** | **0.046** | **29.3** | **0.934** | **0.123** |
| PaviaATN | InDI (firstIN) | 28.8 | 0.950 | 0.102 | 27.1 | 0.903 | **0.150** | 21.3 | 0.768 | **0.248** |
| PaviaATN | scSplit (firstIN) | 33.5 | 0.975 | 0.042 | **27.7** | 0.906 | 0.172 | **24.8** | **0.825** | 0.440 |
| PaviaATN | scSplit | **35.1** | **0.977** | **0.033** | 27.6 | **0.907** | 0.155 | 24.3 | 0.823 | 0.377 |

**Table S.1:** Comparing our normalization scheme (scSplit) with using instance normalization (scSplit (firstIN)).

Next we provide logical arguments which favor SCIN over IN for the task at hand. Instance norm has been reported to degrade discriminative performance in tasks where intensity clues or instance specific contrast is relevant [33, 34]. In microscopy, intensity cues are important. For instance, the background can be easily distinguished from the 'content-less' interior of an organelle like nucleus simply by comparing average intensity value. However, instance norm can make it difficult as it normalizes per patch. This will be more true for smaller patch sizes, typically used for semantic unmixing (patch size of 64 and 128 were used by $\mu$Split [3] and denoiSplit [2] respectively). A similar argument can be made for the Reg network. IN when applied to $c_t$ distorts the structure of the distribution $C_t$, by eliminating statistical differences between individual images, whereas these remain preserved with SCIN. Applying IN to $c_t$ could actually negatively affect the ability of the Reg network to reliably estimate $t$, because by eliminating inter-image statistical differences, IN may discard the very signal that the Reg network requires to estimate t.

Finally, from the computational efficiency standpoint as well, our normalization approach SCIN is better. Our normalization module's extra computation occurs only once before training, when means and standard deviations for each $t$ are calculated—a fast pre-processing step (e.g., about 7 minutes for Hagen et al. in a naive implementation). After this, training (taking 3 days on a Tesla V100 GPU) normalizes input patches using these precomputed values, adding no overhead during training. However, we provide evidence that the IN-based approach could actually be more computationally demanding, particularly when using smaller patch sizes. IN internally needs to compute the standard deviation over the full frame for each input patch , and this computation is performed within the dataset class, and therefore must happen on the CPU. With frame dimensions of several thousand

pixels found typically in microscopy datasets (e.g., $2720 \times 2700$ for PaviaATN, $2048 \times 2048$ for Hagen et al., $4096 \times 4096$ for Chicago-Sch23 [6] ), we conducted a quick estimation: On our high-performance cluster (Intel(R) Xeon(R) Gold 5220 CPU @ 2.20GHz), it takes around 28 milliseconds (measured with %timeit in ipython) to run torch.nn.InstanceNorm2d(1) on a $3000 \times 3000$ image. For a training schedule of $450K$ iterations with a batch size 8 (our configuration), this would amount to about $28 \times 10^{-3} \times 450K \times 8/3600$, which is around 28 hours of added training time solely due to the IN operation. Even with 4 workers (our configuration), this step alone would still require 7 hours. This is about 10 percent increment in training time for our task. Additionally, when using smaller patch sizes, either batch size or iteration count needs to increase if we want the training to see the data 'same number of times', which would only increase the computational cost. For instance, just halving the patch size to 256 would increase the extra computation cost to 28 hours.

## C  Ablation of Normalization Module

| Model | Dom. | | | Bal. | | | Weak. | | |
|---|---|---|---|---|---|---|---|---|---|
| | PSNR | SSIM | LPIPS | PSNR | SSIM | LPIPS | PSNR | SSIM | LPIPS |
| PaviaATN | 35.1 | 0.978 | 0.032 | 25.9 | 0.895 | 0.202 | 23.7 | 0.813 | 0.492 |
| Hagen | 39.4 | 0.990 | 0.017 | 30.8 | 0.962 | 0.067 | 27.8 | 0.923 | 0.120 |
| HTT24 | 42.7 | 0.992 | 0.003 | 37.1 | 0.973 | 0.011 | 33.7 | 0.934 | 0.034 |
| BioSR | 39.3 | 0.979 | 0.017 | 33.6 | 0.936 | 0.059 | 28.1 | 0.874 | 0.146 |

**Table S.2:** Model performance (PSNR, SSIM, LPIPS) without SCIN module.

Here, we conduct an ablation study where the scSplit model was trained without SCIN module. Instead, we used the commonly adopted approach in iterative inference models, where the two images are normalized first using mean-std normalization and then combined via a convex combination to form the intermediate input $x_t$. In the supplementary table S.2, we present the results of scSplit using this 2-image normalization scheme across four datasets. For each task, we trained the Reg and networks under this normalization approach. Across all tasks and all input regimes, it is evident that the PSNR values of the original scSplit model (as reported in Table 1 of the main text) outperform those reported here, with the sole exception of the task on PaviaATN dataset under the dominant input regime, where the PSNR values are equal.

It is important to highlight a limitation in the normalization approach used by current semantic unmixing methods [3, 2, 6]. These methods typically estimate a single set of mean and standard deviation values from all input data and apply this global normalization to input patches from the training, validation, and test sets. In practice, this presents a challenge: test images acquired from microscopes often exhibit distinct intensity distributions compared to the training data. Therefore, it would be more appropriate to normalize test images using statistics computed specifically from each newly acquired dataset. Some existing methods [3, 2] performed well using global normalization because their train, validation, and test images originated from the same acquisition session. In Sup. Tab. S.8, we report the performance of $\mu$Split variants when global normalization statistics—shared across train, validation, and test—are used. Notably, scSplit (shown in Table 1) still achieves superior results. Nevertheless, we emphasize that applying global normalization in this manner is not practically feasible for real-world inference scenarios.

## D  Thoughts on Iterative Inference

Our implementation of InDI has the possibility of doing iterative inference with any number of steps. With minimal change one can also perform iterative inference with out scSplit. However, in this work, we have consciously avoided iterative inference and have always performed 1-step inference. In the next few paragraphs, we explain the rationale behind it.

In Figure 3 of InDI [4], the authors quantitatively show that more iterations lead to worse PSNR (higher distortion) but improved perceptual quality. The same is shown for three datasets in Figure 9 of InDI, that shows a monotonous drop in PSNR when increasing the number of iterations, with total drop being atleast 2db PSNR in all three cases. When working with predictive models on biological data, one can argue that in the perception-distortion tradeoff [35], the motivation is to get the model with a better fidelity to the recorded image (lower distortion). This would not be true if the motivation

was to generate synthetic datasets, in which case better perceptual quality might be the major goal. With a focus on better fidelity, we use one step prediction for both InDI and inDiSplit. Note that training in InDI is not iterative and so this choice does not affect in any way how InDI is trained.

That all being said, iterative inference in indiSplit can done in the following way: Instead of starting from $t = 1$ as done in InDI, one needs to start from the value of $t$ estimated with *Reg* network. $\Delta$, the unit decrement in $t$, can simply be computed as $\Delta = t_{start}/totalSteps$. This is implemented in code (see line 93 in `model/ddpm_modules/indi.py`). The one change that needs to happen is to ensure that $\mathbb{E}[\mu(x_t)] = 0$ and $\mathbb{E}[\sigma(x_t)] = 1$, where $\mu()$ and $\sigma()$ represents the mean and the standard deviation operators. This can easily be achieved by a normalization step in each step of the iterative inference.

## E  Quantitative Evaluation Methodology

In this section, we illustrate the methodology used to compute the results presented in Table 1 of the main manuscript by taking the following example. Let us describe the performance evaluation process for a specific value of $w$, such as $w = 0.7$. To assess a model's performance for $w = 0.7$ on a given dataset, we first generate inputs with $t = 0.3$ using Equation 1 and evaluate the performance metrics for $C_0$ (using *Gen*$_0$). Next, we generate inputs with $t = 0.7$ and evaluate the performance for $C_1$. The average of these two metric values represents the model's overall performance for $w = 0.7$. Each metric is computed on individual image frames, and the average performance reflects the mean metric value across all test set frames. The corresponding standard errors (computed over all test frames) for Table 1, averaged across both channels, are provided in Table S.10. For Table 2, the standard error values, averaged across both channels, are presented in Table S.9.

## F  Normalization Details for Our Baselines

The official implementations of various $\mu$Split variants and denoi$\mathbb{S}$plit are publicly available, and they share an identical input normalization scheme. During training, all pixels from images in both sub-datasets $C_0$ and $C_1$ are aggregated into a one-dimensional array, and the mean and standard deviation of this array are used to normalize all input images in the validation and test sets. It is important to note that the set of normalized patches from training data does not have a zero mean and unit standard deviation. So, similar to InDI, the optimal normalization scheme which should be employed on test images is easy to obtain.

For evaluating these baselines on the tasks presented in Table 1, we adhere to their normalization scheme with one modification: for each mixing ratio $t$, the mean and standard deviation are computed using pixels exclusively from the input images $c_t$. Note that when evaluating test images, we cannot assume access to individual channels and so cannot use their normalization scheme. Note that this scheme is applied solely to these baselines during evaluation.

## G  Hyper-parameters and Training Details

We use a patch size of 512 to train sc$\mathbb{S}$plit, InDI, U-Net and denoi$\mathbb{S}$plit networks. For $\mu$Split variants, we kept the patch size as 64, which is used in their official implementation. The reason is that, in spite of using 64 as the patch size, they effectively see the content of $1024 \times 1024$ sized region surrounding the primary input patch. This is because of the presence of the LC module they have in their network which takes as input additional low-resolution patches centered on the primary patch but spanning larger and larger spatial regions. For training *Gen*$_i$ networks, we use MAE loss and for training *Reg* network, we use MSE loss. We use Adam optimizer with a learning rate of $1e - 3$. To have a fair comparison between InDI and sc$\mathbb{S}$plit, we used the same parameter count, non-linearity, number of layers *etc.* between them. We used MMSE-count of 10 to compute all metrics. In other words, for every input, we predicted 10 times, and used the average prediction for metric computation. All metric computation has been done on predictions of entire frames (and not on patches).

For Hagen et al. dataset, to allow all methods to compare with pretrained models of $\mu$Split variants, we followed $\mu$Split code of applying upper-clip to the data at 0.995 quantile. We upper-clipped the data at intensity value of 1993. This corresponds to 0.995 quantile of the entire training data. Similarly, for PaviaATN, the upper-clipping operation was done at 1308 value.

| GT | Mean | Median | Mode | WgtSum | WgtProd |
|------|------|--------|------|--------|---------|
| 0.00 | 0.13 | 0.12 | 0.11 | 0.14 | 0.15 |
| 0.10 | 0.15 | 0.14 | 0.10 | 0.16 | 0.16 |
| 0.20 | 0.21 | 0.20 | 0.13 | 0.20 | 0.20 |
| 0.30 | 0.31 | 0.28 | 0.27 | 0.29 | 0.27 |
| 0.40 | 0.43 | 0.41 | 0.35 | 0.42 | 0.40 |
| 0.50 | 0.54 | 0.53 | 0.66 | 0.54 | 0.51 |
| 0.60 | 0.64 | 0.64 | 0.61 | 0.63 | 0.61 |
| 0.70 | 0.72 | 0.72 | 0.74 | 0.71 | 0.69 |
| 0.80 | 0.77 | 0.78 | 0.79 | 0.77 | 0.76 |
| 0.90 | 0.80 | 0.79 | 0.78 | 0.79 | 0.80 |
| 1.00 | 0.80 | 0.80 | 0.78 | 0.80 | 0.81 |

**Table S.3:** Quantitative evaluation of different aggregation methodologies on BioSR data. First column is the ground truth $t$ and all other columns are the aggregated predictions using different aggregation methodologies.

| GT | Mean | Median | Mode | WgtSum | WgtProd |
|------|------|--------|------|--------|---------|
| 0.00 | 0.13 | 0.10 | 0.09 | 0.11 | 0.11 |
| 0.10 | 0.14 | 0.12 | 0.10 | 0.12 | 0.12 |
| 0.20 | 0.25 | 0.23 | 0.22 | 0.22 | 0.24 |
| 0.30 | 0.35 | 0.35 | 0.35 | 0.34 | 0.36 |
| 0.40 | 0.45 | 0.45 | 0.46 | 0.45 | 0.46 |
| 0.50 | 0.53 | 0.54 | 0.54 | 0.53 | 0.55 |
| 0.60 | 0.61 | 0.62 | 0.64 | 0.61 | 0.63 |
| 0.70 | 0.69 | 0.70 | 0.71 | 0.69 | 0.71 |
| 0.80 | 0.78 | 0.80 | 0.79 | 0.78 | 0.80 |
| 0.90 | 0.87 | 0.89 | 0.87 | 0.86 | 0.87 |
| 1.00 | 0.92 | 0.96 | 0.96 | 0.93 | 0.92 |

**Table S.4:** Quantitative evaluation of different aggregation methodologies on HTT24 data. First column is the ground truth $t$ and all other columns are the aggregated predictions using different aggregation methodologies.

## H   Different Aggregation Methodologies

In this section, we experiment with different ways to aggregate the estimates of the mixing ratio $t$. We iterate over the test set and for each patch, we get an estimate of $t$. We tried several ways to aggregate the estimates. We aggregated the estimates using mean, median and mode as three different ways. Next, based on the hypothesis that mixing ratio estimates predicted from patches containing both structures could be more accurate than those from patches dominated by a single structure, we implemented two additional aggregation methods. For them, we replicate the scalar mixing ratio predictions to match the shape of the input patches. We then tile these replicated mixing ratio predictions so that they have the same shape as the full input frames. Using $t = 0.5$, we first make a rough estimate of both channels, $\hat{c}_0$ and $\hat{c}_1$. We then take the weighted average of the pixels present in the tiled mixing ratio frame, with weights computed by normalizing (I) $\hat{c}_0 + \hat{c}_1$ and (II) $\hat{c}_0 * \hat{c}_1$. So, pixels in which both structures are present will get a higher weight. We call them *WgtSum* (corresponding to I) and *WgtProd* (corresponding to II). We do not observe any significant advantage for any of the above mentioned aggregation methods in Tables S.3, S.4 and S.5 and so we resort to using mean as our aggregation method on all tasks.

## I   On Design of *Gen_i*

As described in the main text, we worked with a setup which requires one generator network per channel. While one can envisage an alternative implementation using a single generative network with two channels instead of two separate networks $Gen_0$ and $Gen_1$, extending it to multiple channels with each channel contributing differently to the input, which is our future goal, would become complicated and so we decided on this cleaner design.

| GT | Mean | Median | Mode | WgtSum | WgtProd |
|------|------|--------|------|--------|---------|
| 0.00 | 0.13 | 0.11 | 0.09 | 0.13 | 0.13 |
| 0.10 | 0.18 | 0.16 | 0.14 | 0.18 | 0.18 |
| 0.20 | 0.24 | 0.22 | 0.17 | 0.24 | 0.24 |
| 0.30 | 0.34 | 0.33 | 0.26 | 0.32 | 0.32 |
| 0.40 | 0.43 | 0.42 | 0.44 | 0.40 | 0.40 |
| 0.50 | 0.52 | 0.51 | 0.52 | 0.49 | 0.48 |
| 0.60 | 0.60 | 0.58 | 0.57 | 0.56 | 0.56 |
| 0.70 | 0.69 | 0.68 | 0.61 | 0.66 | 0.65 |
| 0.80 | 0.78 | 0.78 | 0.76 | 0.76 | 0.75 |
| 0.90 | 0.87 | 0.88 | 0.88 | 0.88 | 0.87 |
| 1.00 | 0.89 | 0.90 | 0.90 | 0.91 | 0.90 |

**Table S.5:** Quantitative evaluation of different aggregation methodologies on HTLIF24 data. First column is the ground truth $t$ and all other columns are the aggregated predictions using different aggregation methodologies.

| $t_{assumed}$ | PSNR |
|---------------|------|
| 0.3 | 35.0 |
| 0.5 | 37.3 |
| 0.7 | 36.2 |
| 1.0 | 32.4 |

**Table S.6:** Evaluating performance of scSplit on synthetic inputs created from the test sub-dataset of GoPro motion deblurring dataset. Informally speaking, half of the haze was removed from the original test images using Equation 11. scSplit indeed was able to yield superior performance when $t$ was set to $0.5$ during inference.

## J    Application to Natural Images

For image restoration tasks, it is evident that in reality, images with different levels of degradation exist and therefore, a method that is cognizant of the severity of degradation is expected to have advantages.

However, to make our idea applicable to image restoration tasks on natural images, one would need to account for the differences between the image unmixing task performed on microscopy data and those tasks. For example, in fluorescence microscopy, we made a plausible assumption that a single acquisition amounts to a single mixing ratio. However, in a motion deblurring task, the portion of the image containing a moving object, *e.g.* a car, will have more severe blurring when compared to a static object, like a wall. So, one would need to account for this spatial variation of the degradation. For de-hazing and de-raining tasks, a similar challenge holds. Objects more distant from the camera typically have more degradation. So, to handle such spatially varying degradations in a diligent manner, $t$, the input to scSplit also needs to be spatially varying and therefore should not remain a scalar. Secondly, for the image unmixing task described in this work, we have access to the two channels, and so we can correctly define the mixing ratio. However, with these image restoration tasks, one does not have access to the other channel, which would be pure degradation. One instead has access to the clean content and an intermediately degraded image. So, $t = 1$ will have different connotations for the image-unmixing task described in this work and image restoration tasks on natural images, which need to be properly accounted for.

However, as a proof of concept, we trained scSplit with just one generator network $Gen_0$ for the motion deblurring task on GoPro motion deblurring dataset [36]. Since all the variations of severity of degradation present in the training data were explicitly mapped to $t = 1$ during training ( all degraded images belong to $C_1$, which means $t = 1$ for all such images and this is irrespective of their qualitative degradation levels), it is expected that during inference, $t = 1$ will be the optimum choice on the test sub-dataset provided in this dataset.

So, to enhance the diversity of the degradation, we took the test set of the GoPro dataset and created a set of less degraded input images, by simply averaging the inputs with the respective targets *i.e.* ,

$$x^i_{new} = 0.5x^i + 0.5y^i, \tag{11}$$

| Actual Input | Synthesized Input | Pred (t=0.5) | Pred (t=1.0) | Ground Truth |
|---|---|---|---|---|

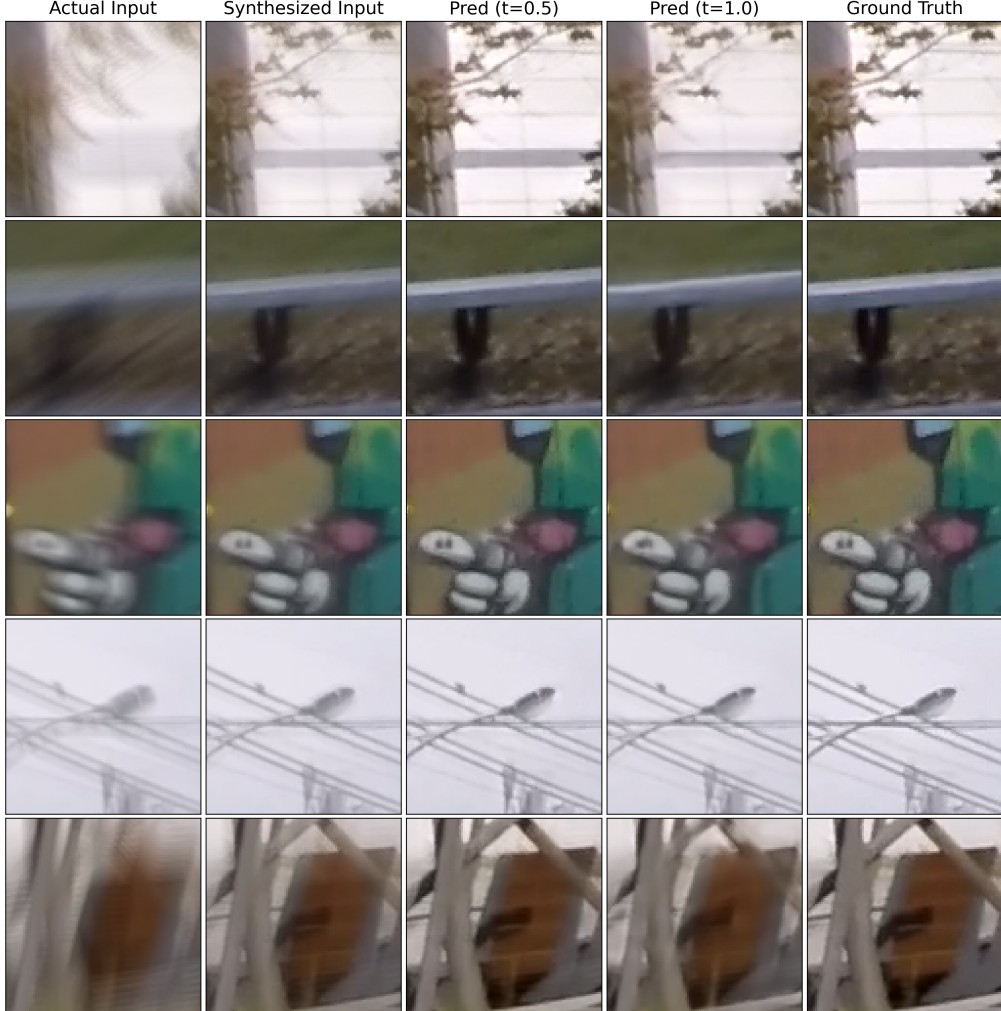

**Figure S.2:** Here, we qualitatively show the potential benefit of regulating the restoration process by making the network cognizant to severity of degradation present in the input image. Predictions are made on 'Synthesized Input', which is created by doing the pixelwise average of Actual Input and the ground truth image. This essentially yields a less blurry image. In the reference frame of the trained scSplit, $t = 0.5$ would be the optimal inference setting for these synthesized inputs. We cherry picked few $100 \times 100$ size crops where the difference between the prediction made by scSplit with $t = 0.5$ and with $t = 1.0$ was clearly visible. We also show quantitative evaluation on all similarly synthesized input frames from the test sub-dataset in Table S.6.

where $x^i$ is the original input image and $y^i$ is the corresponding target image. Next, we evaluate the scSplit network on all $x^i_{new}$ images while using different values of $t$ during inference. We show the results in Table S.6 and the qualitative results in Figure S.2.

In this case, one indeed observes that $t = 0.5$ is the optimal choice. Interestingly, even slightly off estimates of $t$ ($t = 0.3$, and $t = 0.7$) also yield superior performance over $t = 1.0$. This proof-of-concept experiment shows that when handling blurry images with lower levels of degradation, $t = 1$ is not an optimal choice. But due to the non-trivial differences between our current image unmixing task and these restoration tasks, as outlined above, we plan to take up the task of adapting scSplit for natural images in a separate work.

## K    Analyzing the Effect of Precision

In the official configuration of $\mu$Split variants, we found that the training was done with 16 bit floating point precision. The pre-trained models for $\mu$Split are also trained with 16 bit precision. However, other baselines and scSplit are trained with 32 bit precision. So, we trained $\mu$Split$_L$ with 32 bit precision to assess the performance difference. We compare the performance in Table S.7. Across the three input regimes, one observes the average PSNR increment of 0.3db, MS-SSIM increment of 0.002 and LPIPS decrement of 0.004 when one uses 32 bit floating point precision. By observing Table 1, it is evident that this change is smaller than the advantage scSplit has across all three input regimes.

## L    Selecting Number of Intervals $n$

During training, we chose the number of intervals $n$ used to partition the domain of mixing ratio, which is $[0, 1]$, based on the principle that the interval width should not exceed the error in estimating $t$ using Reg. If it did, we would accept reducing the mixing ratio granularity that the Reg network is, in principle, capable of giving us. For example, in the HTLIF24 derived task, the mean absolute error (MAE) between the predicted and actual $t$ values within the valid range $[0.1, 0.9]$ is 0.03 (see Supplementary Table S3, column 2). Therefore, any interval size smaller than 0.03 would be appropriate for this task. Following this reasoning, we selected a smaller interval size of 0.01—resulting in 100 intervals—to ensure the interval width remained sufficiently low across all tasks.

Since interval count is a hyper-parameter rather than a learnable parameter, and we do not perform explicit optimization on it, it would be inaccurate to assert that 100 is the best choice universally. Our selection of 100 was based on an intuitive argument presented in the previous paragraph. However, there is no downside to choosing a sufficiently large number of intervals (apart from the one time compute required to estimate the mean and std for each $t$), so one may opt to do so.

## M    Qualitative Performance Evaluation

For different values of $w$, we show the qualitative results for the Hagen et al. dataset in Figures S.4, S.5 and S.6. For the BioSR dataset, results are shown in Figures S.7, S.8 and S.9. For the HTT24 dataset, results are shown in Figures S.10, S.11 and S.12. For the HTLIF24 dataset, results are shown in Figures S.13, S.14 and S.15. For the PaviaATN dataset, results are shown in Figures S.16, S.17 and S.18.

|  | Dominant | | | Balanced | | | Weak | | |
|---|---|---|---|---|---|---|---|---|---|
|  | PSNR | SSIM | LPIPS | PSNR | SSIM | LPIPS | PSNR | SSIM | LPIPS |
| $\mu$Split$_L$ (16-bit) | 37.8 | 0.918 | 0.066 | 33.5 | 0.959 | 0.051 | 25.7 | 0.738 | 0.291 |
| $\mu$Split$_L$ (32-bit) | 38.1 | 0.918 | 0.065 | 33.9 | 0.962 | 0.045 | 25.9 | 0.741 | 0.284 |

**Table S.7:** Analysing the effect of training with 32 bit *vs.* training with 16 bit floating point precision. In $\mu$Split variants, the official configuration is to train with 16 bit floating point precision.

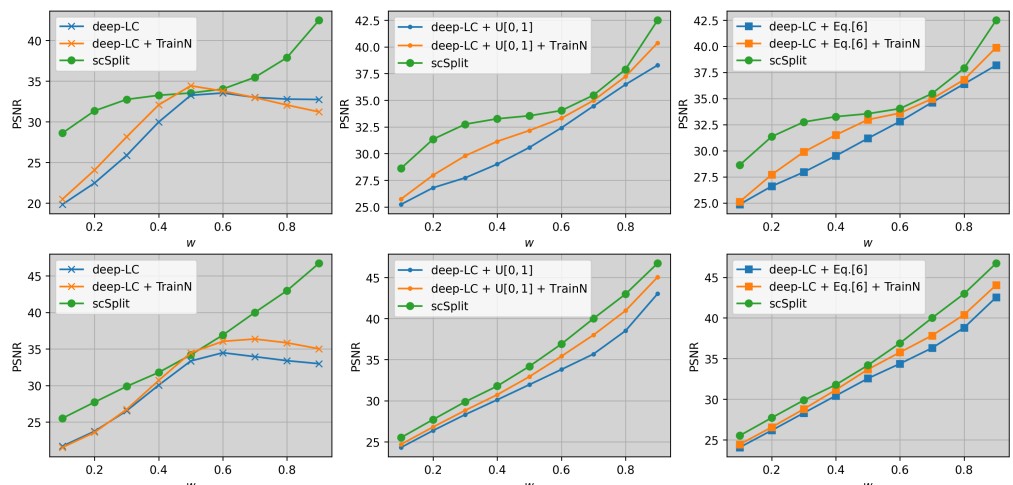

**Figure S.3:** This figure investigates the influence of normalization on the performance of $\mu\text{Split}_D$, emphasizing its inferior outcomes relative to scSplit. Notably, other $\mu$Split variants and denoiSplit share the same normalization setup as $\mu\text{Split}_D$, making this analysis broadly applicable to several existing unmixing methods. We evaluate $\mu\text{Split}_D$ and two of its variants, each utilizing distinct augmentation strategies, as outlined in the main manuscript's Fig 4. Performance is assessed under two normalization schemes: (1) the default approach, where mean and standard deviation are derived from the training data, and (2) a $w$-dependent method, where statistics are calculated separately for inputs with a specific $w$. The results demonstrate that all $\mu\text{Split}_D$ variants underperform compared to scSplit under both normalization strategies. Furthermore, we note that $\mu\text{Split}_D$, trained with uniform normalization statistics, performs more effectively under the default evaluation setup, which as discussed in the main manuscript is not a reasonable choice given the intensity variations across different microscopy acquisitions.

| Dataset | | Dominant | | | Balanced | | | Weak | | |
|---|---|---|---|---|---|---|---|---|---|---|
| | | PSNR | SSIM | LPIPS | PSNR | SSIM | LPIPS | PSNR | SSIM | LPIPS |
| | Inp vs Tar | 34.1 | 0.973 | 0.047 | 25.1 | 0.889 | 0.148 | 21.2 | 0.784 | 0.243 |
| Hagen et. al | U-Net | 33.5 | 0.976 | 0.038 | 33.4 | 0.960 | 0.066 | 23.3 | 0.840 | 0.190 |
| | $\mu\text{Split}_L$ | 34.2 | 0.974 | 0.044 | 32.3 | 0.959 | 0.071 | 23.8 | 0.843 | 0.187 |
| | $\mu\text{Split}_R$ | 34.6 | 0.971 | 0.041 | 32.4 | 0.957 | 0.068 | 24.5 | 0.842 | 0.189 |
| | $\mu\text{Split}_D$ | 33.9 | 0.975 | 0.039 | 33.6 | 0.963 | 0.061 | 24.1 | 0.849 | 0.179 |

**Table S.8:** This table, analogous to Table 1 in the main manuscript, evaluates performance on the Hagen et al. dataset across the same three input categories. Here, we utilize the mean and standard deviation computed from the training data for normalization. Since $\mu$Split variants are trained with these statistics, they achieve superior performance with this normalization setting, since the test images share the same intensity distribution as the training data due to being from the same acquisition. However, as outlined in the main manuscript, this approach is not viable for evaluating images from different acquisitions, where intensity distributions may vary. Notably, scSplit surpasses even these results, demonstrating its robustness and superior performance.

| | HT-LIF24 | | | HT-T24 | | |
|---|---|---|---|---|---|---|
| | PSNR | SSIM | LPIPS | PSNR | SSIM | LPIPS |
| U-Net | .692 | .002 | .002 | .613 | .006 | .002 |
| $\mu\text{Split}_L$ | .635 | .001 | .002 | .614 | .007 | .002 |
| $\mu\text{Split}_R$ | .697 | .002 | .002 | .624 | .007 | .002 |
| $\mu\text{Split}_D$ | .734 | .002 | .002 | .626 | .008 | .002 |
| denoiSplit | .675 | .001 | .002 | .574 | .005 | .003 |
| InDI | .560 | .002 | .002 | .596 | .007 | .004 |
| $\text{scSplit}_{0.5}$ | .704 | .001 | .001 | .627 | .007 | .002 |
| $\text{scSplit}_{-agg}$ | .669 | .002 | .001 | .657 | .008 | .002 |
| scSplit | .736 | .001 | .001 | .627 | .007 | .002 |

**Table S.9: Standard error estimates in Table 2 (channel-averaged).**

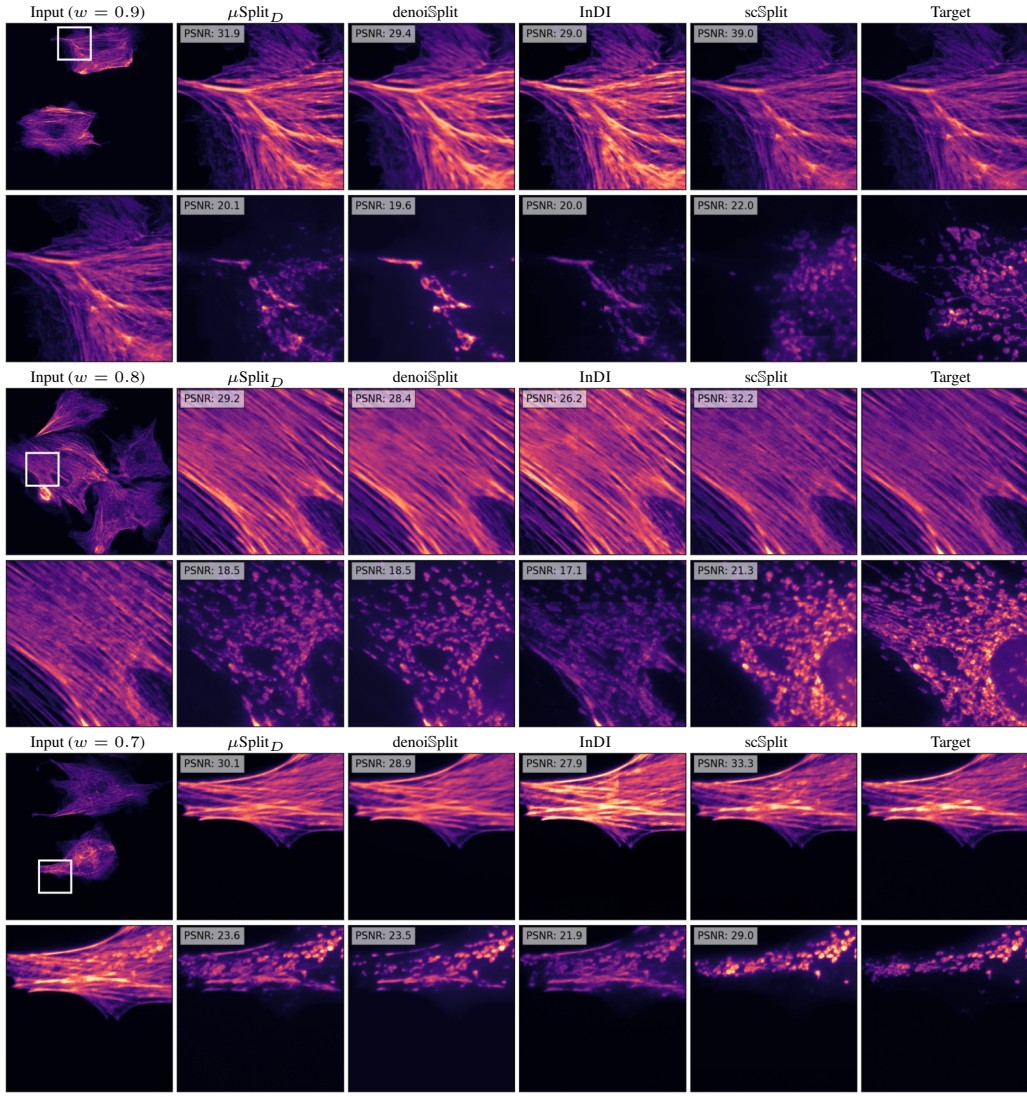

**Figure S.4:** Qualitative evaluation for Hagen et al. In each panel, we show the full input frame (top-left) and the zoomed-in input patch (bottom-left) for which we show the predictions and the targets (last col) for the two channels, one in each row. We also report PSNR values for the patch shown. The $w$ value reported on top of the input column is for the first channel. It naturally becomes $1 - w$ for the second channel.

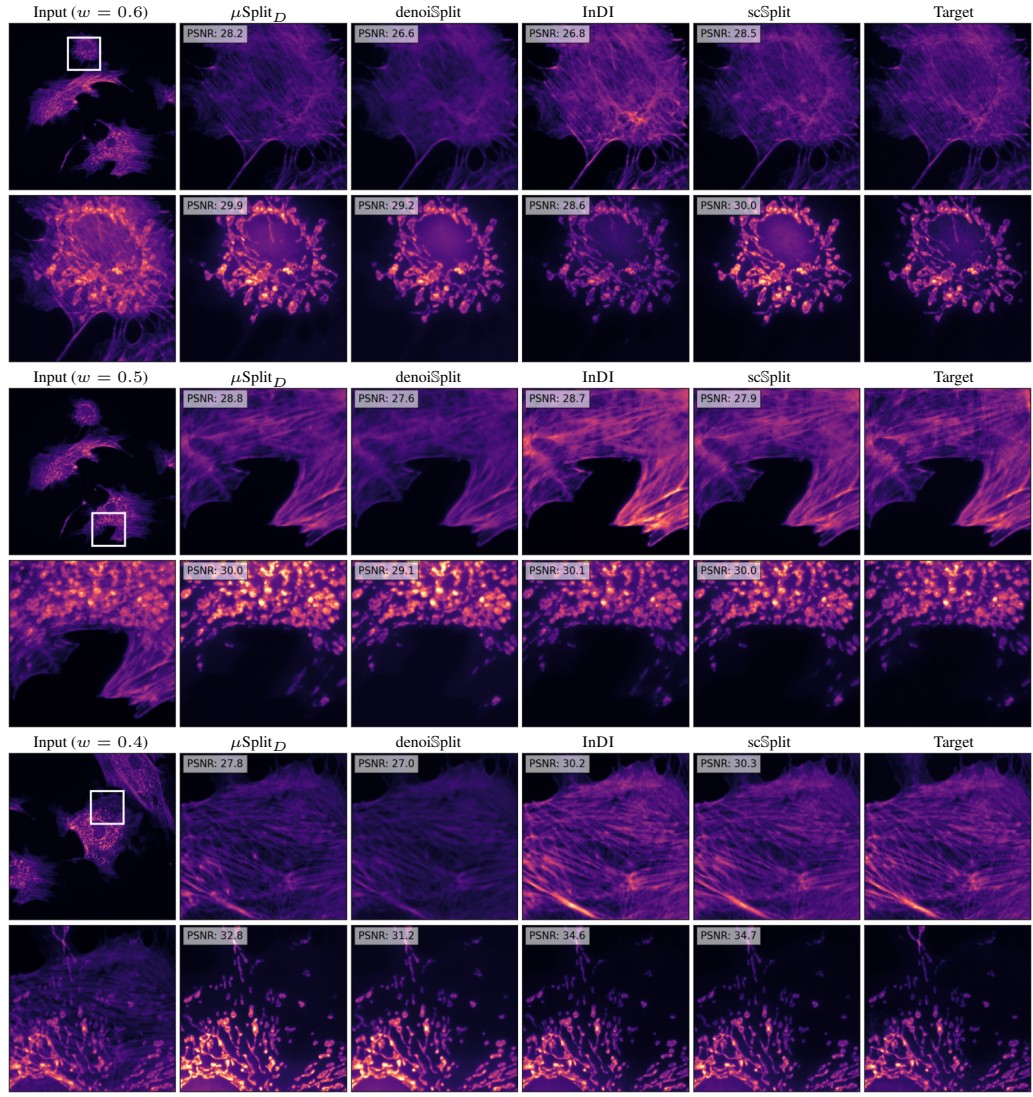

**Figure S.5:** Qualitative evaluation for Hagen et al. In each panel, we show the full input frame (top-left) and the zoomed-in input patch (bottom-left) for which we show the predictions and the targets (last col) for the two channels, one in each row. We also report PSNR values for the patch shown. The $w$ value reported on top of the input column is for the first channel. It naturally becomes $1 - w$ for the second channel.

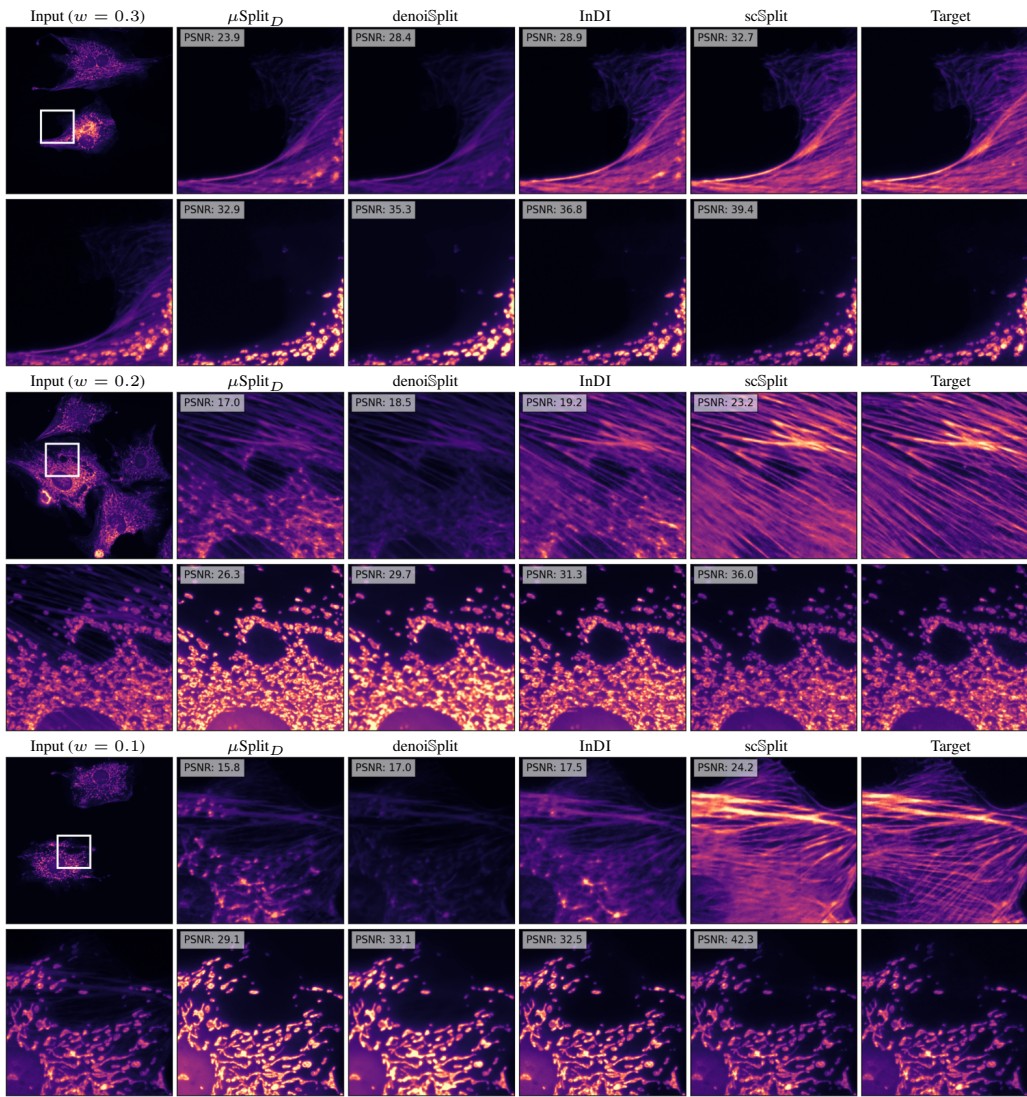

**Figure S.6:** Qualitative evaluation for Hagen et al. In each panel, we show the full input frame (top-left) and the zoomed-in input patch (bottom-left) for which we show the predictions and the targets (last col) for the two channels, one in each row. We also report PSNR values for the patch shown. The $w$ value reported on top of the input column is for the first channel. It naturally becomes $1 - w$ for the second channel.

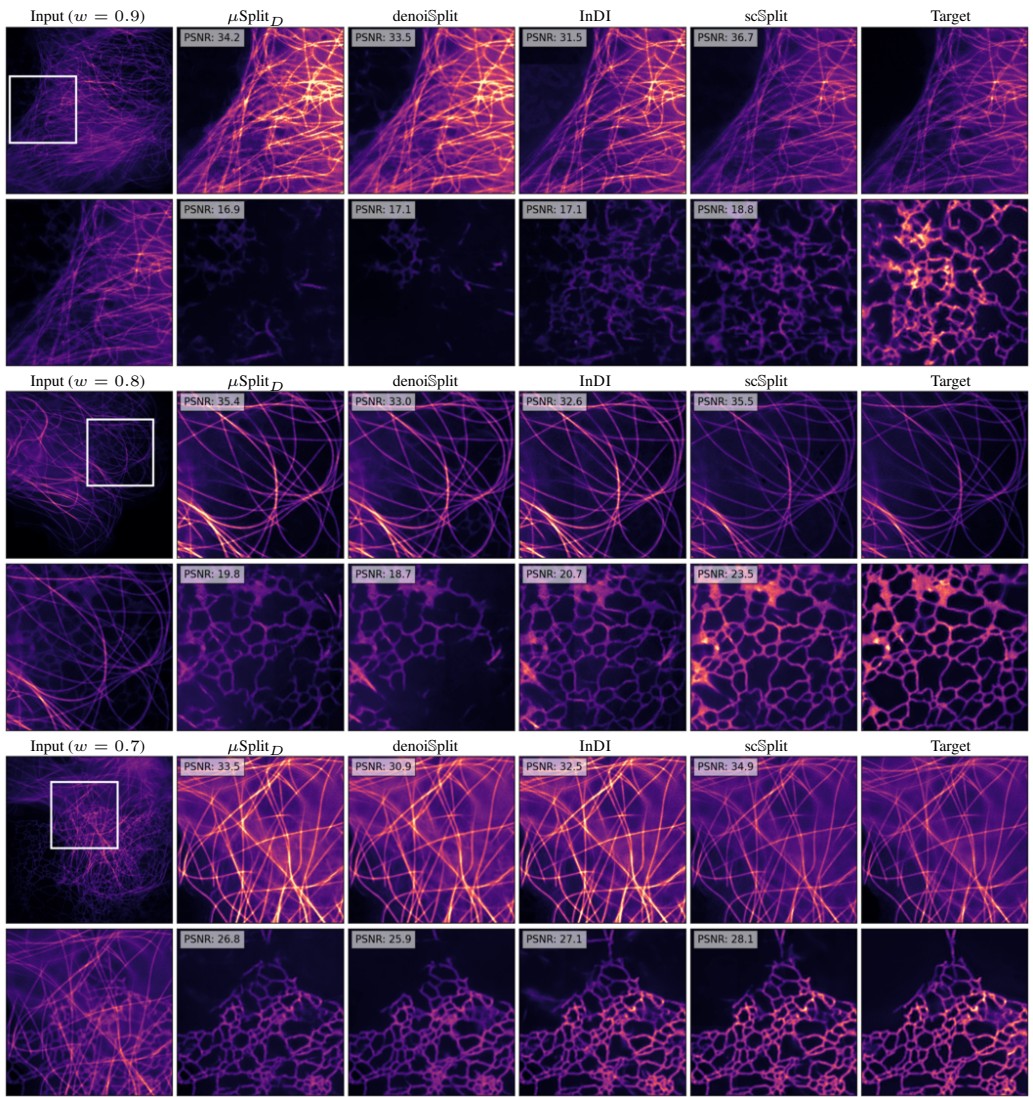

**Figure S.7:** Qualitative evaluation for BioSR dataset. In each panel, we show the full input frame (top-left) and the zoomed-in input patch (bottom-left) for which we show the predictions and the targets (last col) for the two channels, one in each row. We also report PSNR values for the patch shown. The $w$ value reported on top of the input column is for the first channel. It naturally becomes $1 - w$ for the second channel.

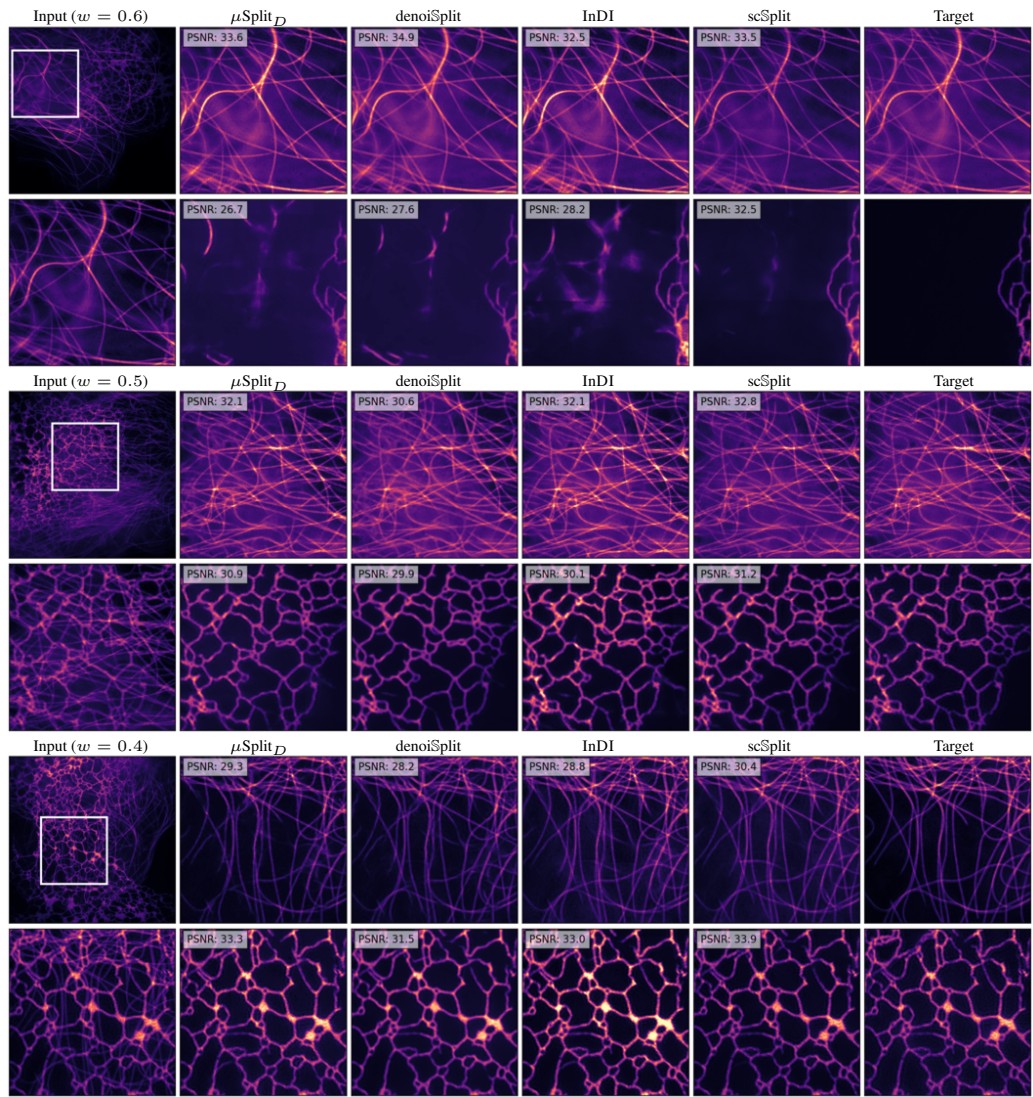

**Figure S.8:** Qualitative evaluation for BioSR dataset. In each panel, we show the full input frame (top-left) and the zoomed-in input patch (bottom-left) for which we show the predictions and the targets (last col) for the two channels, one in each row. We also report PSNR values for the patch shown. The $w$ value reported on top of the input column is for the first channel. It naturally becomes $1 - w$ for the second channel.

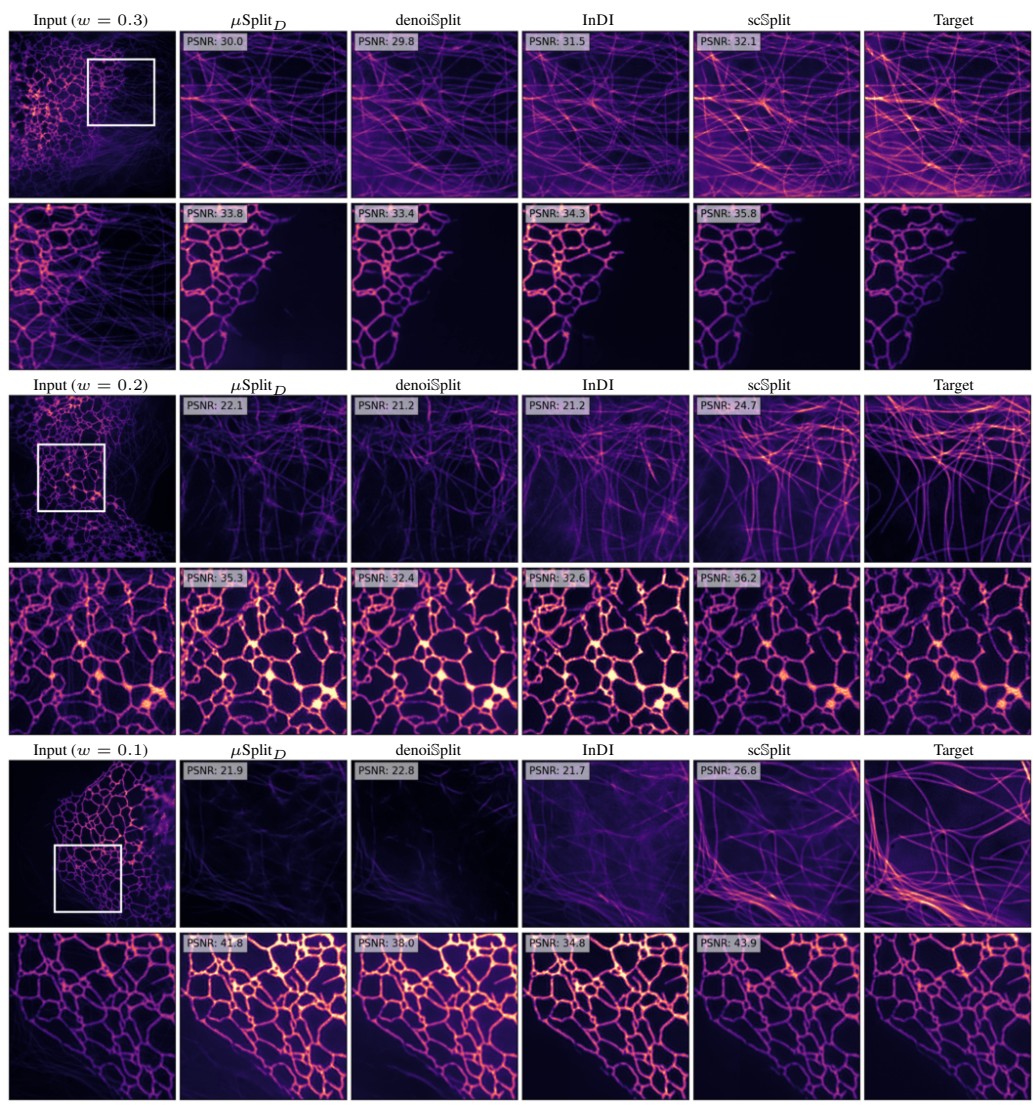

**Figure S.9:** Qualitative evaluation for BioSR dataset. In each panel, we show the full input frame (top-left) and the zoomed-in input patch (bottom-left) for which we show the predictions and the targets (last col) for the two channels, one in each row. We also report PSNR values for the patch shown. The $w$ value reported on top of the input column is for the first channel. It naturally becomes $1 - w$ for the second channel.

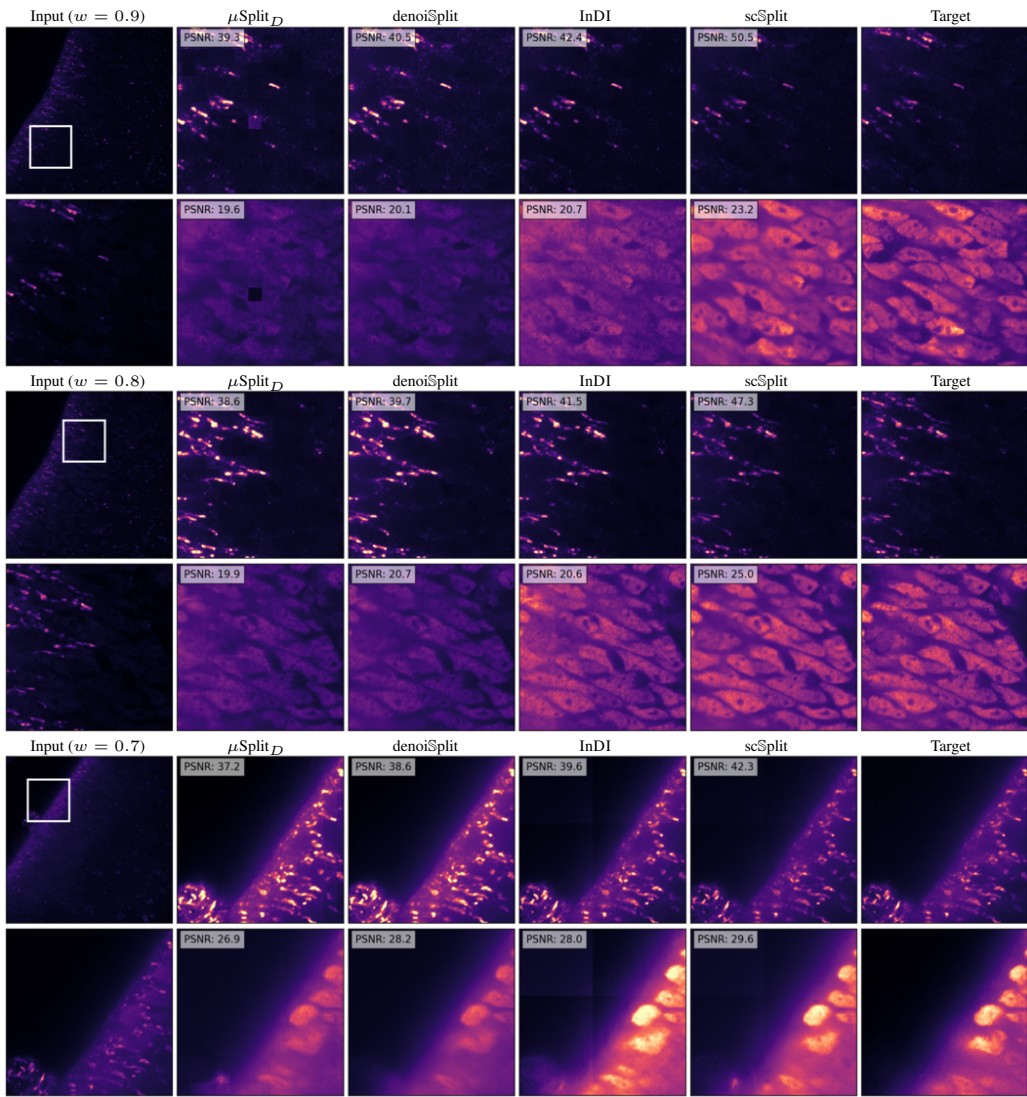

**Figure S.10:** Qualitative evaluation for HTT24 dataset. In each panel, we show the full input frame (top-left) and the zoomed-in input patch (bottom-left) for which we show the predictions and the targets (last col) for the two channels, one in each row. We also report PSNR values for the patch shown. The $w$ value reported on top of the input column is for the first channel. It naturally becomes $1 - w$ for the second channel.

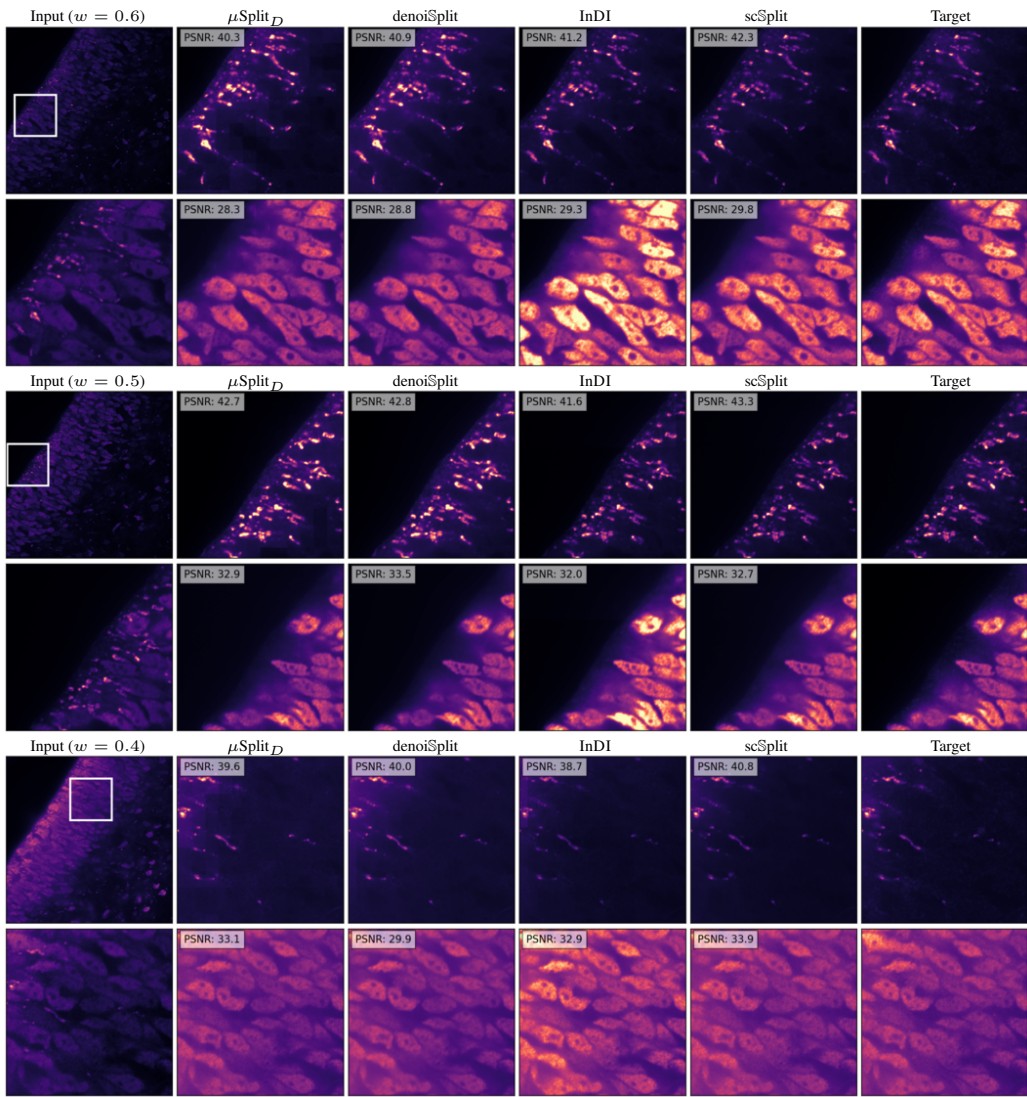

**Figure S.11:** Qualitative evaluation for HTT24 dataset. In each panel, we show the full input frame (top-left) and the zoomed-in input patch (bottom-left) for which we show the predictions and the targets (last col) for the two channels, one in each row. We also report PSNR values for the patch shown. The $w$ value reported on top of the input column is for the first channel. It naturally becomes $1 - w$ for the second channel.

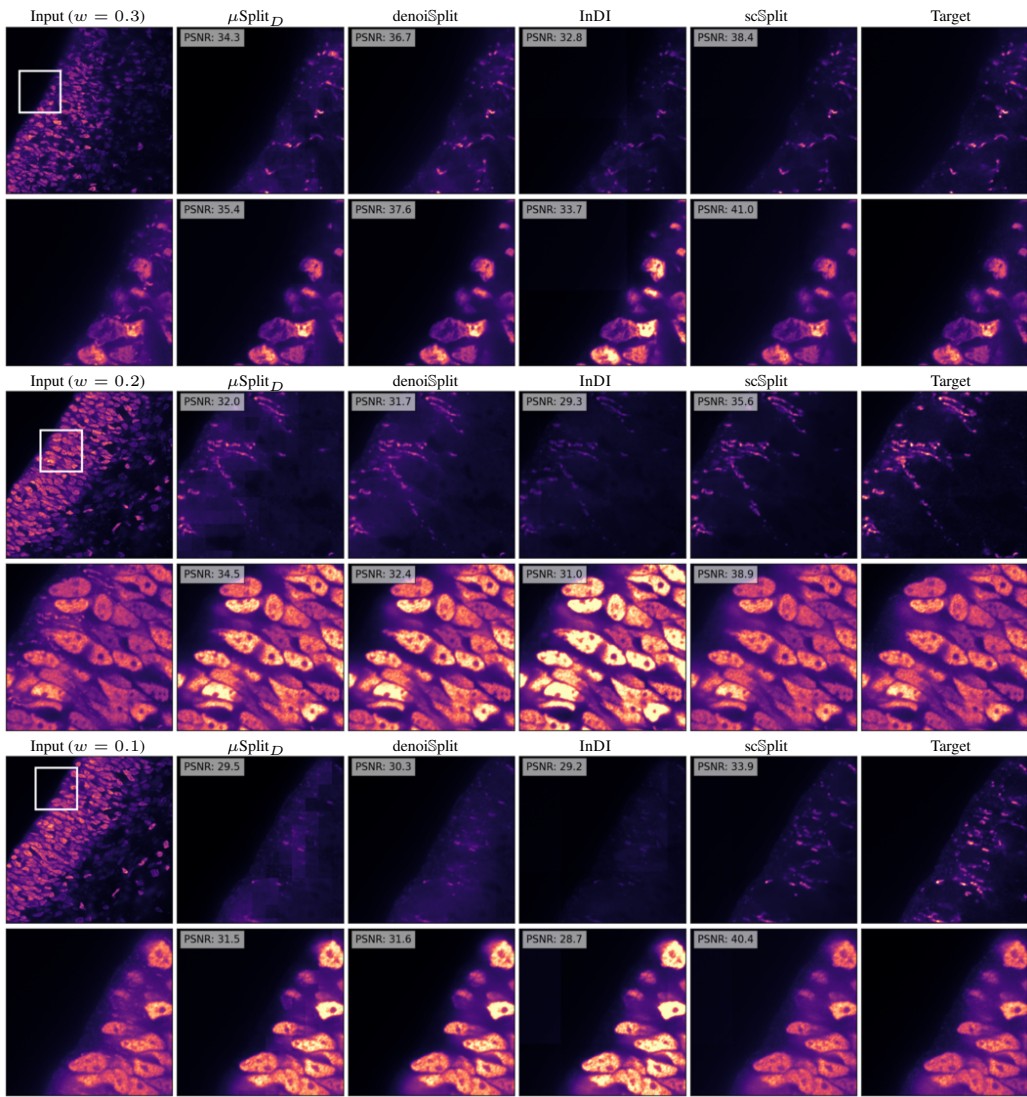

**Figure S.12:** Qualitative evaluation for HTT24 dataset. In each panel, we show the full input frame (top-left) and the zoomed-in input patch (bottom-left) for which we show the predictions and the targets (last col) for the two channels, one in each row. We also report PSNR values for the patch shown. The $w$ value reported on top of the input column is for the first channel. It naturally becomes $1 - w$ for the second channel.

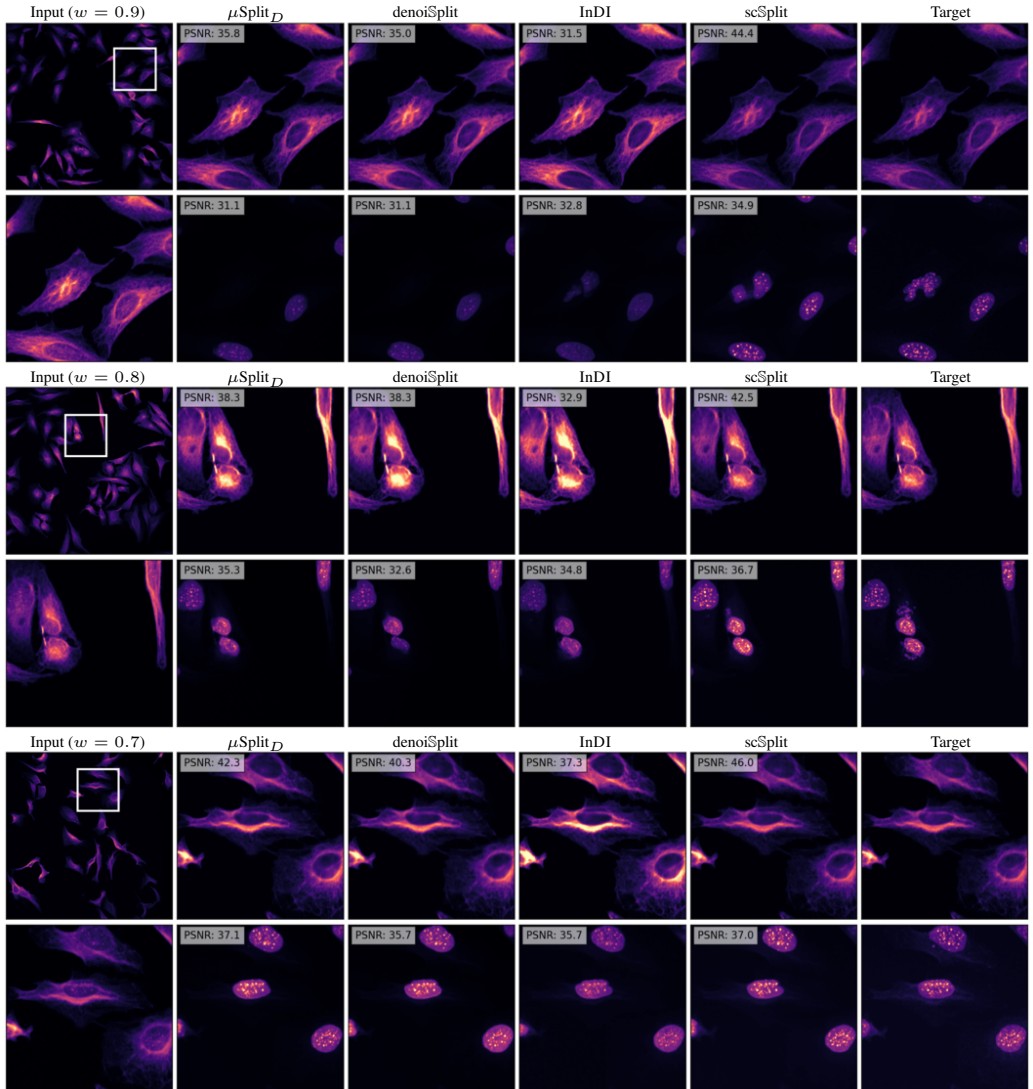

**Figure S.13:** Qualitative evaluation for HTLIF24 dataset. In each panel, we show the full input frame (top-left) and the zoomed-in input patch (bottom-left) for which we show the predictions and the targets (last col) for the two channels, one in each row. We also report PSNR values for the patch shown. The $w$ value reported on top of the input column is for the first channel. It naturally becomes $1 - w$ for the second channel.

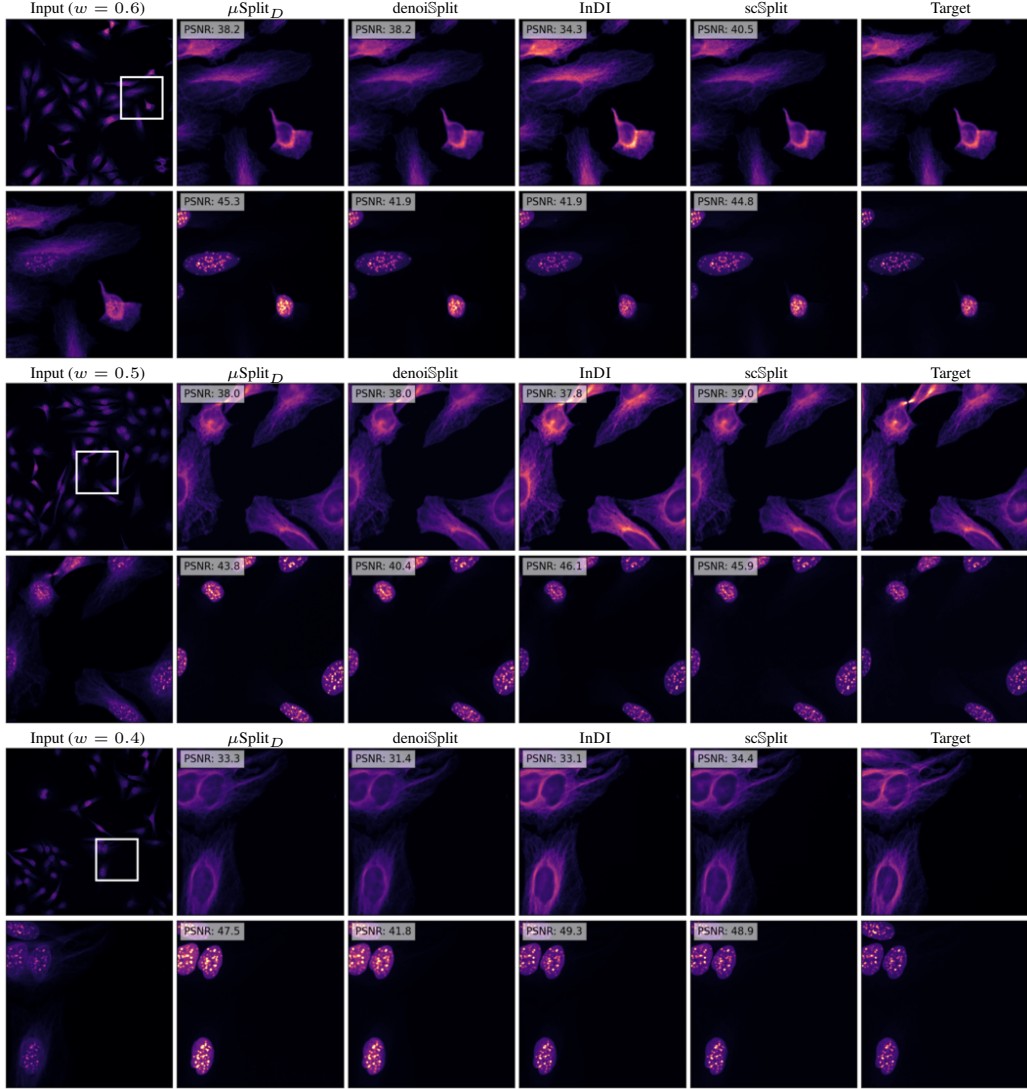

**Figure S.14:** Qualitative evaluation for HTLIF24 dataset. In each panel, we show the full input frame (top-left) and the zoomed-in input patch (bottom-left) for which we show the predictions and the targets (last col) for the two channels, one in each row. We also report PSNR values for the patch shown. The $w$ value reported on top of the input column is for the first channel. It naturally becomes $1 - w$ for the second channel.

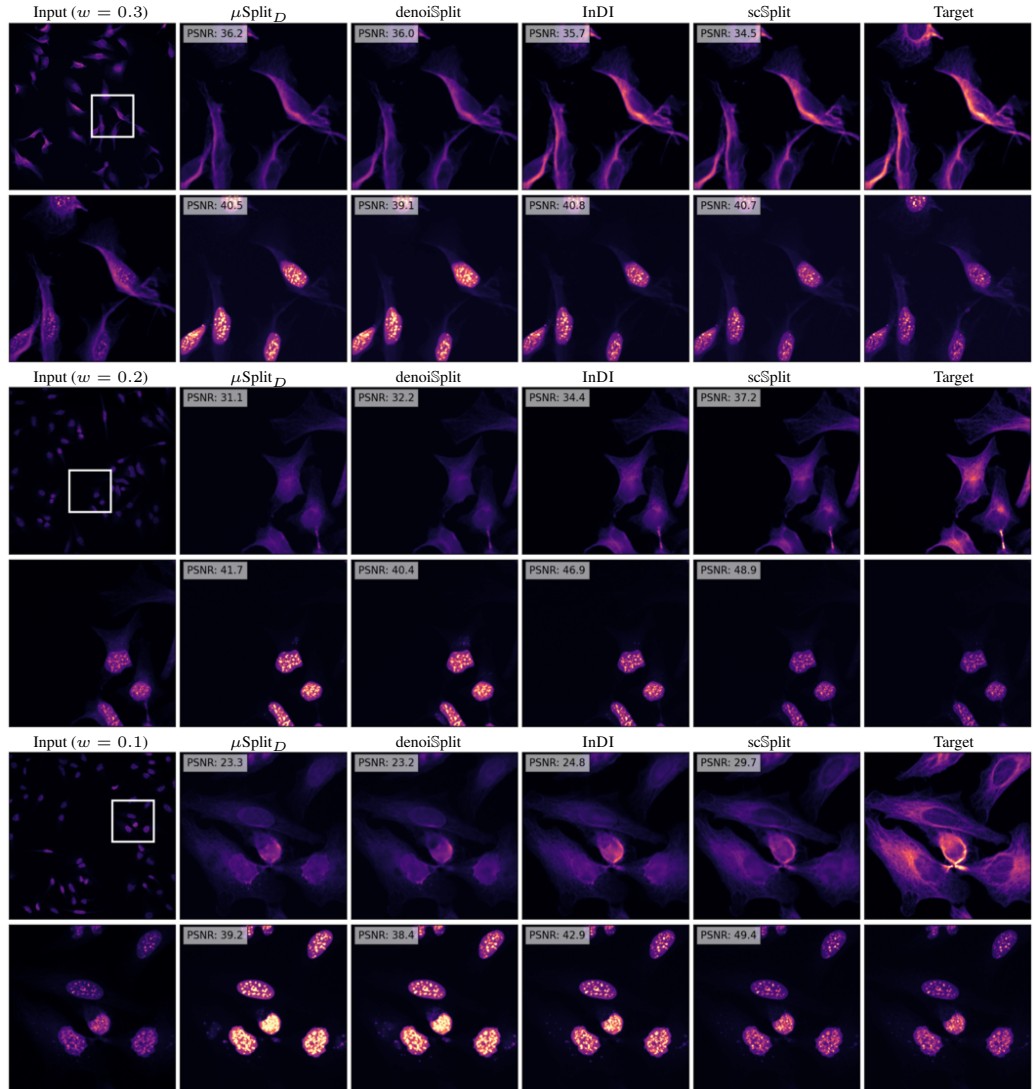

**Figure S.15:** Qualitative evaluation for HTLIF24 dataset. In each panel, we show the full input frame (top-left) and the zoomed-in input patch (bottom-left) for which we show the predictions and the targets (last col) for the two channels, one in each row. We also report PSNR values for the patch shown. The $w$ value reported on top of the input column is for the first channel. It naturally becomes $1 - w$ for the second channel.

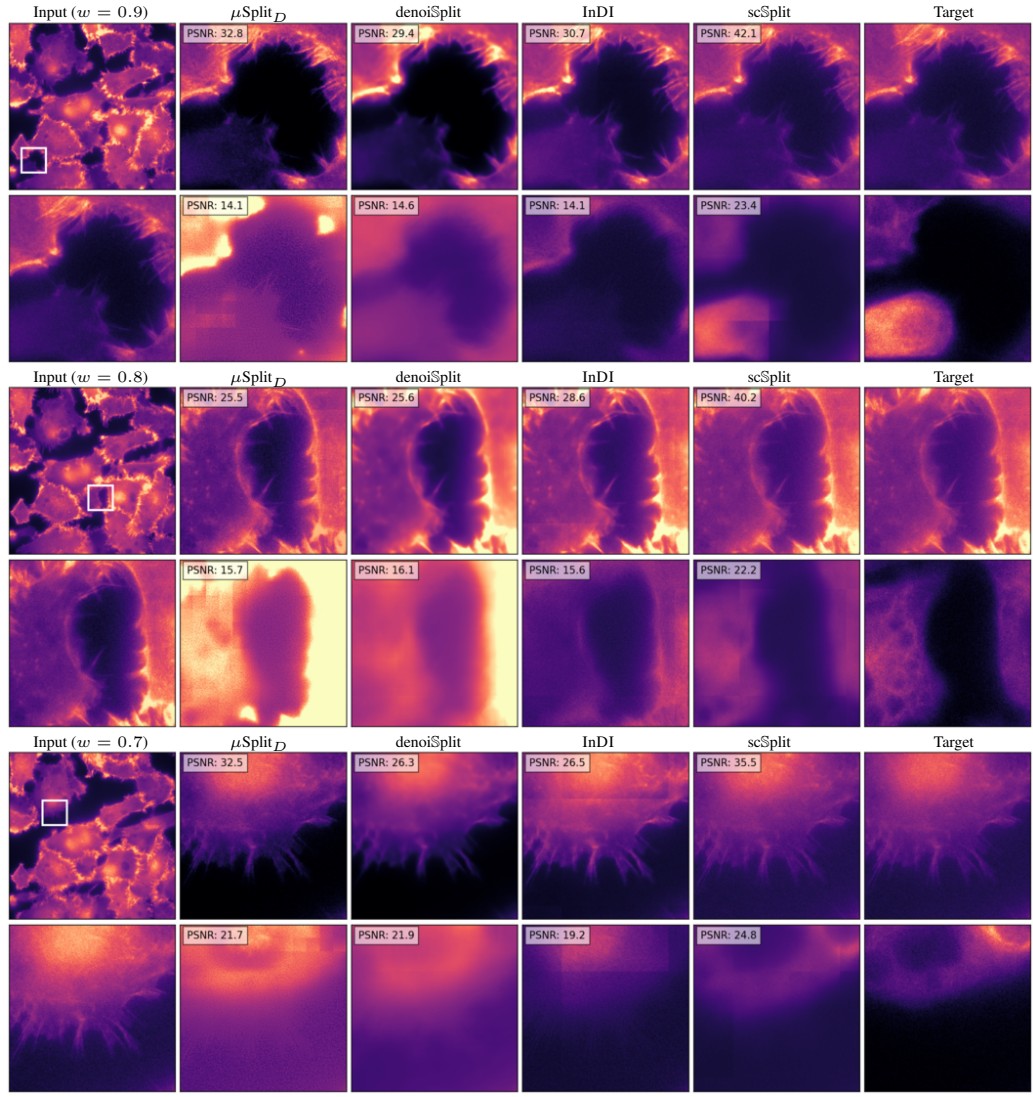

**Figure S.16:** Qualitative evaluation for PaviaATN dataset. In each panel, we show the full input frame (top-left) and the zoomed-in input patch (bottom-left) for which we show the predictions and the targets (last col) for the two channels, one in each row. We also report PSNR values for the patch shown. The $w$ value reported on top of the input column is for the first channel. It naturally becomes $1 - w$ for the second channel.

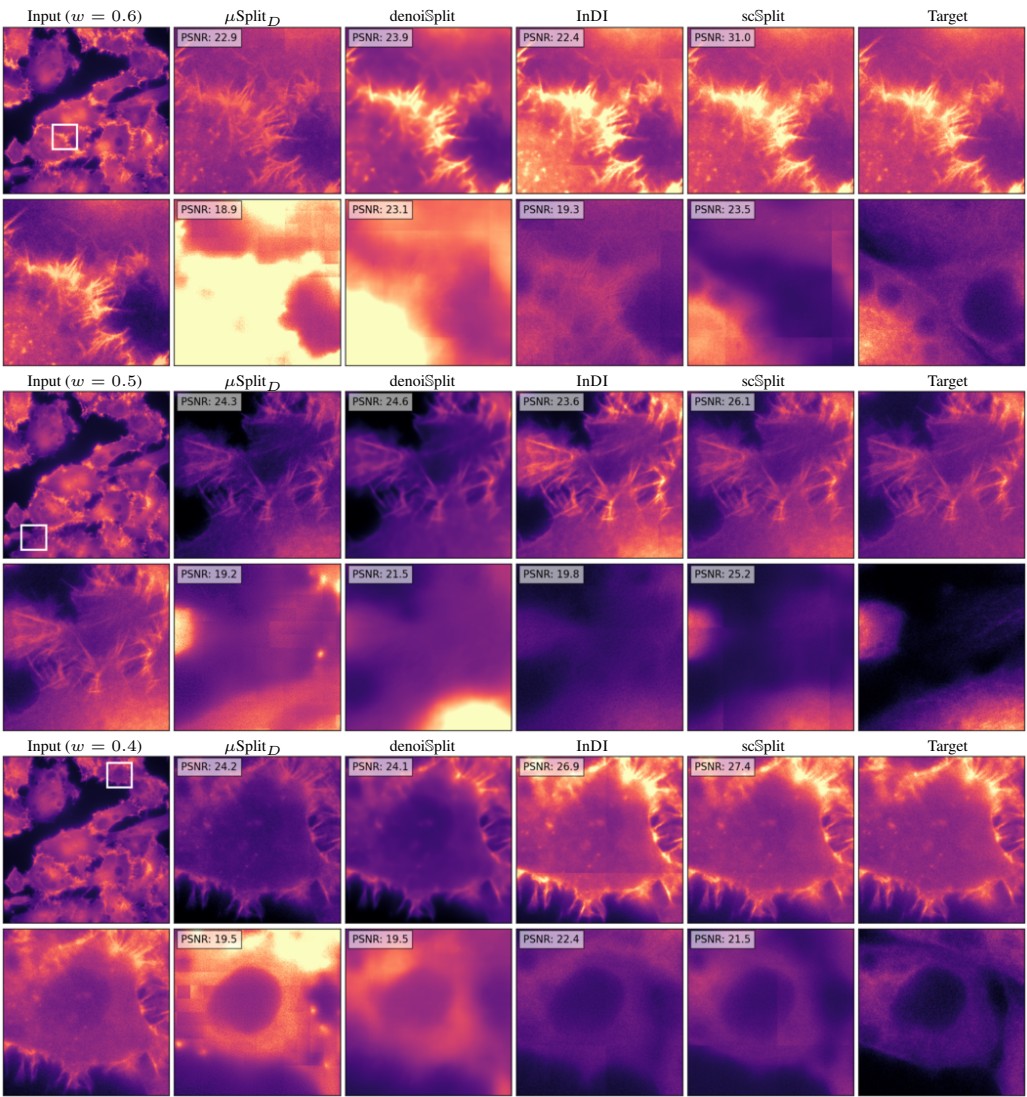

**Figure S.17:** Qualitative evaluation for PaviaATN dataset. In each panel, we show the full input frame (top-left) and the zoomed-in input patch (bottom-left) for which we show the predictions and the targets (last col) for the two channels, one in each row. We also report PSNR values for the patch shown. The $w$ value reported on top of the input column is for the first channel. It naturally becomes $1 - w$ for the second channel.

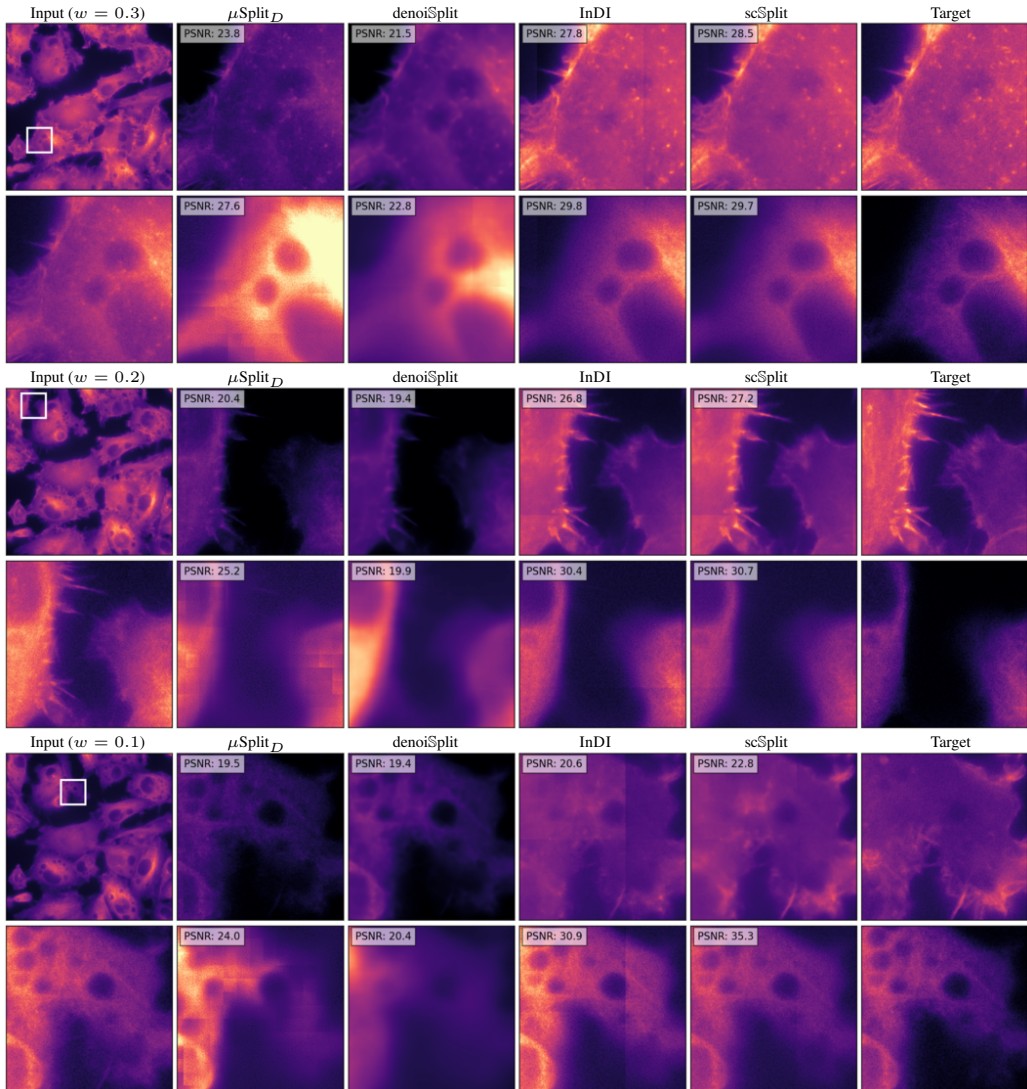

**Figure S.18:** Qualitative evaluation for PaviaATN dataset. In each panel, we show the full input frame (top-left) and the zoomed-in input patch (bottom-left) for which we show the predictions and the targets (last col) for the two channels, one in each row. We also report PSNR values for the patch shown. The $w$ value reported on top of the input column is for the first channel. It naturally becomes $1 - w$ for the second channel.

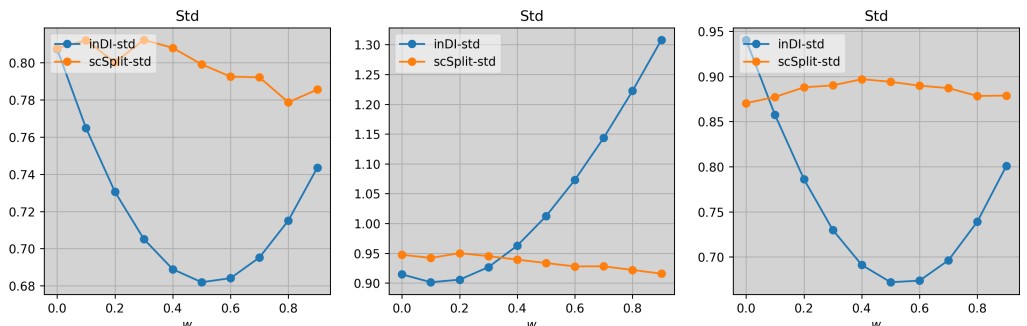

**Figure S.19:** Analysis of input patch variability across mixing factors $w$ using 2000 randomly sampled $512 \times 512$ image pairs $(c_0, c_1)$: (a) Supervised image restoration models like InDI [4] normalizes $c_0$ and $c_1$ separately before interpolation, resulting in input patches with standard deviation strongly correlated with $w$; (b) our proposed method decouples this relationship. Results across Hagen et al. (left), HTLIF24 (center), and BioSR (right) datasets demonstrate reduced dependency on $w$ with our approach.

| Dataset | | Dominant | | | Balanced | | | Weak | | |
|---|---|---|---|---|---|---|---|---|---|---|
| | | PSNR | SSIM | LPIPS | PSNR | SSIM | LPIPS | PSNR | SSIM | LPIPS |
| Hagen et. al | U-Net | 0.5 | 0.003 | 0.003 | 0.5 | 0.007 | 0.007 | 0.6 | 0.015 | 0.014 |
| | $\mu\text{Split}_L$ | 0.8 | 0.004 | 0.004 | 0.6 | 0.005 | 0.006 | 0.5 | 0.015 | 0.014 |
| | $\mu\text{Split}_R$ | 0.8 | 0.004 | 0.004 | 0.6 | 0.005 | 0.006 | 0.6 | 0.015 | 0.015 |
| | $\mu\text{Split}_D$ | 0.7 | 0.003 | 0.004 | 0.6 | 0.004 | 0.005 | 0.6 | 0.015 | 0.013 |
| | denoiSplit | 0.8 | 0.005 | 0.012 | 0.6 | 0.005 | 0.013 | 0.6 | 0.016 | 0.020 |
| | MicroSplit | 0.7 | 0.004 | 0.012 | 0.7 | 0.004 | 0.012 | 0.6 | 0.015 | 0.019 |
| | InDI | 0.8 | 0.003 | 0.004 | 0.7 | 0.004 | 0.005 | 0.5 | 0.013 | 0.012 |
| | $\text{sc}\mathbb{S}\text{plit}_{0.5}$ | 0.8 | 0.002 | 0.003 | 0.6 | 0.003 | 0.004 | 0.6 | 0.013 | 0.012 |
| | $\text{sc}\mathbb{S}\text{plit}_{-agg}$ | 0.5 | 0.001 | 0.001 | 0.5 | 0.003 | 0.004 | 0.5 | 0.007 | 0.010 |
| | $\text{sc}\mathbb{S}\text{plit}$ | 0.5 | 0.001 | 0.001 | 0.5 | 0.002 | 0.004 | 0.6 | 0.007 | 0.010 |
| HTLIF24 | U-Net | 0.7 | 0.001 | 0.002 | 0.7 | 0.001 | 0.001 | 0.8 | 0.005 | 0.004 |
| | $\mu\text{Split}_L$ | 0.7 | 0.002 | 0.002 | 0.7 | 0.001 | 0.001 | 0.8 | 0.005 | 0.004 |
| | $\mu\text{Split}_R$ | 0.7 | 0.002 | 0.002 | 0.8 | 0.001 | 0.001 | 0.8 | 0.005 | 0.004 |
| | $\mu\text{Split}_D$ | 0.7 | 0.002 | 0.002 | 0.7 | 0.001 | 0.001 | 0.8 | 0.005 | 0.004 |
| | denoiSplit | 0.7 | 0.002 | 0.002 | 0.6 | 0.001 | 0.002 | 0.8 | 0.005 | 0.005 |
| | MicroSplit | 0.6 | 0.002 | 0.002 | 0.6 | 0.001 | 0.002 | 0.7 | 0.005 | 0.005 |
| | InDI | 0.7 | 0.002 | 0.002 | 0.7 | 0.001 | 0.001 | 0.7 | 0.003 | 0.004 |
| | $\text{sc}\mathbb{S}\text{plit}_{0.5}$ | 0.7 | 0.001 | 0.001 | 0.8 | 0.001 | 0.001 | 0.8 | 0.005 | 0.004 |
| | $\text{sc}\mathbb{S}\text{plit}_{-agg}$ | 0.8 | 0.000 | 0.000 | 0.7 | 0.001 | 0.001 | 0.8 | 0.003 | 0.003 |
| | $\text{sc}\mathbb{S}\text{plit}$ | 0.8 | 0.000 | 0.000 | 0.7 | 0.001 | 0.001 | 0.7 | 0.003 | 0.003 |
| BioSR | U-Net | 0.6 | 0.003 | 0.005 | 0.4 | 0.004 | 0.006 | 0.7 | 0.014 | 0.015 |
| | $\mu\text{Split}_L$ | 0.4 | 0.004 | 0.005 | 0.2 | 0.004 | 0.006 | 0.6 | 0.011 | 0.014 |
| | $\mu\text{Split}_R$ | 0.4 | 0.004 | 0.004 | 0.2 | 0.004 | 0.006 | 0.7 | 0.013 | 0.016 |
| | $\mu\text{Split}_D$ | 0.3 | 0.004 | 0.007 | 0.3 | 0.005 | 0.007 | 0.5 | 0.011 | 0.014 |
| | denoiSplit | 0.5 | 0.003 | 0.008 | 0.2 | 0.004 | 0.010 | 0.7 | 0.012 | 0.019 |
| | MicroSplit | 0.6 | 0.003 | 0.007 | 0.3 | 0.003 | 0.010 | 0.7 | 0.013 | 0.019 |
| | InDI | 1.0 | 0.003 | 0.005 | 0.4 | 0.004 | 0.006 | 0.9 | 0.014 | 0.015 |
| | $\text{sc}\mathbb{S}\text{plit}_{0.5}$ | 0.9 | 0.003 | 0.003 | 0.4 | 0.003 | 0.005 | 0.7 | 0.015 | 0.014 |
| | $\text{sc}\mathbb{S}\text{plit}_{-agg}$ | 0.4 | 0.002 | 0.002 | 0.7 | 0.003 | 0.006 | 0.8 | 0.011 | 0.016 |
| | $\text{sc}\mathbb{S}\text{plit}$ | 0.4 | 0.002 | 0.002 | 0.4 | 0.003 | 0.005 | 0.6 | 0.011 | 0.014 |
| HTT24 | U-Net | 0.7 | 0.002 | 0.002 | 0.7 | 0.003 | 0.002 | 0.5 | 0.008 | 0.007 |
| | $\mu\text{Split}_L$ | 0.7 | 0.004 | 0.002 | 0.6 | 0.005 | 0.002 | 0.5 | 0.010 | 0.007 |
| | $\mu\text{Split}_R$ | 0.7 | 0.003 | 0.002 | 0.7 | 0.005 | 0.002 | 0.5 | 0.009 | 0.007 |
| | $\mu\text{Split}_D$ | 0.7 | 0.004 | 0.002 | 0.7 | 0.005 | 0.002 | 0.5 | 0.009 | 0.007 |
| | denoiSplit | 0.7 | 0.003 | 0.004 | 0.6 | 0.004 | 0.003 | 0.5 | 0.006 | 0.007 |
| | MicroSplit | 0.7 | 0.003 | 0.004 | 0.7 | 0.004 | 0.003 | 0.5 | 0.007 | 0.007 |
| | InDI | 0.6 | 0.002 | 0.003 | 0.6 | 0.003 | 0.002 | 0.5 | 0.006 | 0.005 |
| | $\text{sc}\mathbb{S}\text{plit}_{0.5}$ | 0.6 | 0.001 | 0.001 | 0.6 | 0.003 | 0.001 | 0.5 | 0.005 | 0.007 |
| | $\text{sc}\mathbb{S}\text{plit}_{-agg}$ | 0.8 | 0.002 | 0.000 | 0.7 | 0.003 | 0.001 | 0.6 | 0.005 | 0.003 |
| | $\text{sc}\mathbb{S}\text{plit}$ | 0.6 | 0.001 | 0.000 | 0.6 | 0.002 | 0.001 | 0.6 | 0.005 | 0.003 |
| PaviaATN | U-Net | 0.3 | 0.002 | 0.001 | 0.3 | 0.004 | 0.001 | 0.3 | 0.006 | 0.002 |
| | $\mu\text{Split}_L$ | 0.2 | 0.002 | 0.001 | 0.3 | 0.004 | 0.001 | 0.2 | 0.006 | 0.002 |
| | $\mu\text{Split}_R$ | 0.3 | 0.002 | 0.001 | 0.3 | 0.004 | 0.001 | 0.2 | 0.007 | 0.002 |
| | $\mu\text{Split}_D$ | 0.3 | 0.003 | 0.002 | 0.3 | 0.005 | 0.002 | 0.3 | 0.007 | 0.002 |
| | denoiSplit | 0.3 | 0.004 | 0.001 | 0.3 | 0.005 | 0.001 | 0.3 | 0.007 | 0.002 |
| | MicroSplit | 0.1 | 0.004 | 0.001 | 0.1 | 0.004 | 0.001 | 0.1 | 0.004 | 0.001 |
| | InDI | 0.2 | 0.001 | 0.001 | 0.1 | 0.003 | 0.001 | 0.1 | 0.005 | 0.002 |
| | $\text{sc}\mathbb{S}\text{plit}_{0.5}$ | 0.3 | 0.002 | 0.001 | 0.3 | 0.003 | 0.001 | 0.2 | 0.004 | 0.002 |
| | $\text{sc}\mathbb{S}\text{plit}_{-agg}$ | 0.2 | 0.001 | 0.001 | 0.3 | 0.003 | 0.002 | 0.2 | 0.004 | 0.003 |
| | $\text{sc}\mathbb{S}\text{plit}$ | 0.2 | 0.001 | 0.000 | 0.3 | 0.003 | 0.002 | 0.2 | 0.004 | 0.003 |

**Table S.10: Standard error values for Table 1 (channel-averaged).**

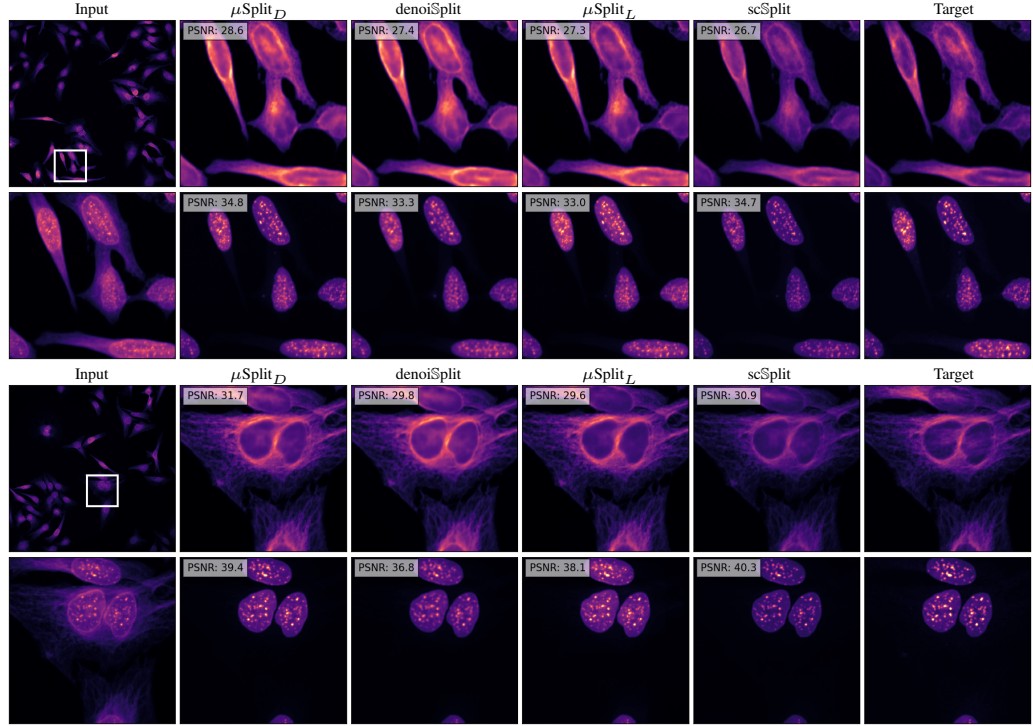

**Figure S.20:** Qualitative evaluation for superimposed raw microscopy images from HTLIF24 dataset. In each panel, we show the full input frame (top-left) and the zoomed-in input patch (bottom-left) for which we show the predictions and the targets (last col) for the two channels, one in each row. We also report PSNR values for the patch shown.

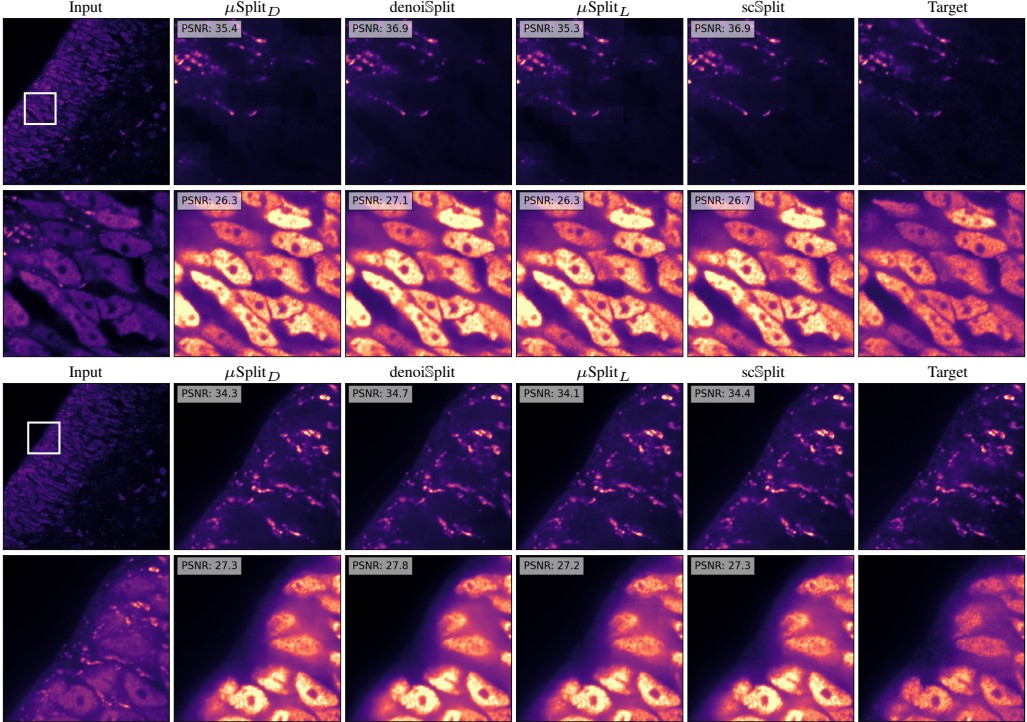

**Figure S.21:** Qualitative evaluation for superimposed raw microscopy images from HTT24 dataset. In each panel, we show the full input frame (top-left) and the zoomed-in input patch (bottom-left) for which we show the predictions and the targets (last col) for the two channels, one in each row. We also report PSNR values for the patch shown.

