# OpenReview forum: "scSplit: Bringing Severity Cognizance to Image Decomposition in Fluorescence Microscopy"
_NeurIPS.cc/2025/Conference — NeurIPS 2025 poster_

### Official Review · Reviewer_Lyze · 2025-06-10

**Clarity:** 2
**Significance:** 3
**Originality:** 3
**Rating:** 4
**Confidence:** 4

**Summary:**

This paper proposes a novel framework, indiSplit, to decompose fluorescence microscopy images containing mixed structures. The authors observe that in real-world scenarios, the mixing ratio of components can vary across images. To address this, they design a two-stage architecture: a Reg model estimates the mixing ratio, and a Gen model then predicts the individual components conditioned on this ratio. Additionally, an auxiliary normalization step is introduced to refine the mixed image, which originally has a variance as a function of the mixing ratio. The indiSplit framework outperforms prior approaches on relevant benchmarks and demonstrates potential generalizability to natural image tasks such as motion deblurring.

**Questions:**

1. Why was 100 chosen as the number of intervals in Section 3.2? Is this value optimal or empirically determined?

2. Please address whether the two-stage pipeline is strictly necessary, or whether an end-to-end model with sufficient training data could perform equally well (related to Weakness 1).

3. Can the model generalize to unseen combinations of component types, or does it require retraining for every new pair (related to Weakness 2)?

4. Can you theoretically or empirically justify the proposed normalization strategy in inference? Given that ground truth is available in your experiments, use it to verify that your normalization leads to the desired variance behavior (related to Weakness 4).

5. What is the precise meaning of t? Is it a mix ratio, a time variable, or both? Please clarify (related to Weakness 5).

**Ethical Concerns:**

["NO or VERY MINOR ethics concerns only"]

**Final Justification:**

The authors have addressed all my concerns in the rebuttal. While the paper has a limitation in that retraining is required if a new combination is provided, the approach of splitting the process into two stages—first estimating the ratio, then separating—offers valuable insight in this field. Therefore, I maintain my original score of Borderline Accept.

**Limitations:**

The paper should clearly state that the current approach requires retraining when the types of mixed components change.

**Quality:**

2

**Strengths And Weaknesses:**

Strengths:
1. The idea of explicitly modeling the varying mixing ratios in composed images is novel and insightful for this task.

2. The proposed normalization mechanism helps align the distribution of mixed images, which empirically improves performance.

3. Decomposing the problem into two sub-tasks (mix ratio estimation and conditional generation) is a meaningful design choice and leads to improved separation quality.

Weaknesses:
Major:
1. It remains unclear why a single unified network could not implicitly learn the mixing ratio and perform separation directly. Typically, multi-stage pipelines are introduced when data scarcity limits end-to-end learning.

2. The method appears to require retraining if the component types change. For example, if a model is trained to separate mixed types $C_0$ and $C_1$, can it generalize to mixed types $C_0$ and $C_2$? This raises concerns about generalizability and scalability.

3. Related to the above, the framework lacks flexibility: different combinations of components require retraining the full model.

4. The inference-time normalization using precomputed statistics indexed by the mean output of the Reg model is underexplained. If the input variance also depends on the mixing ratio, what is the theoretical basis for assuming this strategy leads to unit variance after normalization? A mathematical justification would be helpful.

5. There is ambiguity around the meaning of t. While initially introduced as the mix ratio, Section 3.3 refers to t as time in the “Time Regressor Network Reg,” which is confusing and should be clarified.

Minor:
1. In the abstract, please clarify what is meant by "technical limitations" in the first sentence.

2. Define the term superposition severity, as it is not a standard expression in the literature.

3. Consider writing Bleedthrough in lowercase or using quotation marks to emphasize it.

4. The introduction should explicitly mention that the proposed framework is inspired by InDI.

5. In Section 3.2, specify the data type of the input images (e.g., uint16) to clarify the range and interpretation of pixel intensities.

6. The definition of $C_i$​ appears redundant with the earlier definition of $C_0$​ and $C_1$.

7. There is no definition provided for $\delta$​; this should be clearly introduced.

8. Please standardize the spelling of "artifact" vs. "artefact" throughout the manuscript.

---

> ### Author Rebuttal · Authors · 2025-07-29
>
> ## Weaknesses: Major:
>
> 1. *It remains unclear why a single unified network could not implicitly learn the mixing ratio and perform separation directly. Typically, multi-stage pipelines are introduced when data scarcity limits end-to-end learning.*
>
>     **Answer:** We concur with the reviewer that there is no fundamental conceptual obstacle to jointly training the Reg and the two Gen networks. However, due to practical constraints, e.g. GPU memory and/or network capacity and optimization difficulties for a joint setups, we chose the presented approach. This was further motivated by catering our work to microscopy labs, who are our future users, and typically are even more limited wrt. computational resources.
>
> 2. *The method appears to require retraining if the component types change. For example, if a model is trained to separate mixed types $C_0$ and $C_1$, can it generalize to mixed types $C_0$ and $C_2$? This raises concerns about generalizability and scalability.*
>
>     **Answer:** We agree with the reviewer that one needs to retrain if the mixed structures (component types) change. Today, this is common practice for microscopy denoising and image restoration models, and we would always recommend training from scratch rather then reuse models “out of distribution”.
>
>     Additionally, please note that semantic unmixing is a nascent field. To the best of our knowledge there are only two deep-learning based solutions published in international computer vision/ML conferences (uSplit [3] & denoiSplit [2]). Both of these works do not cater to different mixing ratio. In this work, we show that for both methods, asymetric mixing ratios are leading to problematic results (Table 1, main manuscript). Hence, while we are not suggesting generalizability to unseen structures, our work does help to generalize unmixing to a broader range of mixing ratios than any other method today is capable of.
>
> 3. *Related to the above, the framework lacks flexibility: different combinations of components require retraining the full model.*
>
>     **Answer:** We agree with the reviewer that it will be amazing if such a flexible model is developed and does performs reliably, but the field is yet to develop one. Please see our previous answer for a longer discussion.
>
> 4. *The inference-time normalization using precomputed statistics indexed by the mean output of the Reg model is underexplained. If the input variance also depends on the mixing ratio, what is the theoretical basis for assuming this strategy leads to unit variance after normalization? A mathematical justification would be helpful.*
>
>     **Answer:** Below, we provide both mathematical and empirical justification for our test-time normalization.
>
>     Regrading the mentioned “precomputed statistics”, we want to ensure that there has not been any misunderstandings: The beauty of our training-time normalization method is that inference-time normalization becomes straightforward — one simply calculates the statistics (mean and standard deviation) from the sampled test-image patches, and then use those values for normalization. We will revise the text to make this critical point crystal clear for our future readers.
>
>     **Theoretical Evidence:** In Supplementary Section A, we theoretically show that for images of size HxW, if one was to use iterative inference models such as InDI for semantic unmixing, the standard deviation of intermediate inputs $x_t$ would depend on the mixing ratio $t$. In this work, we have discussed the issues arising from this dependency. To remedy this, during training, we calculate the mean and standard deviation of intermediate input patches $x_t$ separately for each value of t, and use these statistics to normalize the $x_t$. Consequently, by design, the expected mean ($E[\mu(x^{norm}_t)]$) and standard deviation ($E[\sigma(x^{norm}_t)]$) of the normalized $x_t$ are 0 and 1, respectively, for all t. Therefore, at test time, it is sufficient for the input patches to have an expected mean of 0 ($E[\mu(x^{norm})] =0$) and expected standard deviation of 1 ($E[\sigma(x^{norm})] =1$), which is precisely what our inference-time normalization ensures. (Side note: for this argument to be water-tight, we assume that all test-data comes from a single acquisition and therefore has a single value of t.)
>
>     **Empirical Evidence:** We have the empirical evidence in Supplementary Figure S19, where we show the average standard deviation of normalized input patches for $t \in [0.1,0.2,…,0.9]$. We have, for each t, used 10k random input patches from the test dataset to compute the mean and standard deviation. We have then applied those statistics on 2000 other random input patches to get the average standard deviation of normalized input patches, showing that our proposed method leads to standard deviations concentrated around 1 for all mixing ratios t — which is exactly what we have aimed for.
>
> 5. *There is ambiguity around the meaning of t. While initially introduced as the mix ratio, Section 3.3 refers to t as time in the Time Regressor Network Reg, which is confusing and should be clarified.*
>
>
>    **Answer:** We will clarify this in the final manuscript. While t is indeed the mixing ratio, it is also precisely the value the Reg network predicts. We propose to rename the 'Time Regressor Network Reg' to become the 'Mixing-ratio Network Reg'. By avoiding the word 'time', and adding a clear sentence making the connection between t in these two contexts, we believe future readers will be much less likely to be confused.
>
>
>
> ## Raised minor points:
>
> **Answer:** We thank the reviewer for providing these comments. We will naturally incorporate all of them. The phrase 'Technical limiations' refers to excitation and emission spectra overlap, limited photon budgets and other factors which limits the maximum number of structures one can image concurrently in Fluorescence microscopy.
>
>
> ## Questions:
>
> 1. *Why was 100 chosen as the number of intervals in Section 3.2? Is this value optimal or empirically determined?*
>
>     **Answer:** We chose the number of intervals based on the principle that the interval width should not exceed the error in estimating t using Reg. If it did, we would accept to reduce the mixing ratio granularity the Reg network is in principle capable of giving us. For example, in the HTLIF24 derived task, the mean absolute error (MAE) between the predicted and actual t values within the valid range $[0.1, 0.9]$ is 0.03 (see Supplementary Table S3, column 2). Therefore, any interval size smaller than 0.03 would be appropriate for this task. Following this reasoning, we selected a smaller interval size of 0.01—resulting in 100 intervals—to ensure the interval width remained sufficiently low across all tasks.
>
>
> 2. *Please address whether the two-stage pipeline is strictly necessary, or whether an end-to-end model with sufficient training data could perform equally well (related to Weakness 1).*
>
>    **Answer:** See the answer to Major Weakness no. 1.
>
>
> 3. *Can the model generalize to unseen combinations of component types, or does it require retraining for every new pair (related to Weakness 2)?*
>
>    **Answer:** See the answer to Major Weakness no. 2.
>
>
> 4. *Can you theoretically or empirically justify the proposed normalization strategy in inference? Given that ground truth is available in your experiments, use it to verify that your normalization leads to the desired variance behavior (related to Weakness 4).*
>
>      **Answer:** See the answer to Major Weaknesses no. 4.
>
>
> 5. *What is the precise meaning of t? Is it a mix ratio, a time variable, or both? Please clarify (related to Weakness 5).*
>
>      **Answer:** We will clarify in the camera-ready version, as mentioned in our response to Major Weaknesses no. 5.

---

> > ### Author Response · Authors · 2025-08-05
> > **Pending Author-Reviewer discussion**
> >
> > It is a gentle reminder to the reviewer to assess the rebuttal submitted by us. We believe we have answered all of the questions asked by the reviewer, namely justification of proposed normalization strategy, number of intervals, inquiries on generalization and also on two-stage pipeline. We know the reviewer, being a researcher, is super busy by default. But it will greatly benefit us if he/she finds the time for this !

---

### Official Review · Reviewer_ggcu · 2025-06-23

**Clarity:** 2
**Significance:** 3
**Originality:** 2
**Rating:** 4
**Confidence:** 3

**Summary:**

This article addresses the issue of limited observable cell structures in a single fluorescence microscopy image and proposes a method named indiSplit by introducing a regression network to predict mixing proportions and a degradation-specific normalization module. This method solves the tasks of image decomposition and crosstalk elimination, with its effectiveness validated on multiple public datasets.

**Questions:**

1. The input image c_t is generated by linearly superimposing two structural images c_0 and  c_1  with a mixing proportion t in [0,1] , i.e.,  c_t = (1-t)c_0 + tc_1. Why is the superposition implemented as an addition here? Typically, superposition often involves multiplying two textures.
2. Has the author conducted experiments to prove that different mixing proportions significantly affect the results?
3. Is the method applicable to multi-layer superposition?
4. If interference occurs between superimposed textures, does the method in the article remain applicable?
5. It is recommended to split the performance comparison in Table 1 and the model's research on mixing parameters into two separate tables.

**Ethical Concerns:**

["NO or VERY MINOR ethics concerns only"]

**Final Justification:**

My questions and concerns have been addressed, and I will update my final rating to borderline accept.

**Limitations:**

Please refer to my questions part.

**Quality:**

3

**Strengths And Weaknesses:**

### Strengths
The article primarily introduces two modules: mixing ratio perception and degradation-specific normalization. Through a regression network and an aggregation module, the method explicitly models the changes in superimposed intensity, addressing the limitation of traditional methods that assume a fixed mixing ratio. Normalization is employed to ensure consistent input data distribution under different mixing proportions.


### Weakness
Please refer to my detailed comments.

---

> ### Author Rebuttal · Authors · 2025-07-30
>
> ## Questions
>
> 1. *The input image c_t is generated by linearly superimposing two structural images c_0 and c_1 with a mixing proportion t in [0,1] , i.e., c_t = (1-t)c_0 + tc_1. Why is the superposition implemented as an addition here? Typically, superposition often involves multiplying two textures.*
>
>     **Answer:** In fluorescence microscopy, superposition of structures is linear [18-21]. Previous deep-learning based works like uSplit [3] and denoiSplit[2], published in International Computer Vision conferences, also work with the above mentioned weighted average formulation.
>
> 2. *Has the author conducted experiments to prove that different mixing proportions significantly affect the results?*
>
>     **Answer:** There are two potential ways in which different mixing proportions can affect results.
>
>     (a) One way is that the prediction quality for the less represented channel is poorer. This trend appears consistently across all rows of Table 1, where metrics are highest for the Dominant input regime, followed by the Balanced regime, and lowest for the Weak regime. The test set’s underlying structures remain unchanged across these regimes; only the input mixing ratios vary. For instance, given two images $ c_1 $ and $ c_2 $, an input is formed as $x = 0.1 c_1 + 0.9 c_2 $. We trust the reviewer would agree that predicting $ c_2 $ from $ x $ is easier than predicting $ c_1 $, since $ c_2 $ is more prominently represented in the input.
>
>     (b) The second is that if the network was accustomed to train on a specific mixing-ratio, then, at inference time, deviation from that mixing-ratio would lead to inferior results. For this, we conducted an experiment demonstrating that indiSplit outperforms baselines when the relative strengths of structures in real superimposed input differ from those in synthetic sums. Using the HT-T24 dataset, we created two dataset variants by multiplying all pixel values in one structure channel by 4. So, the first dataset variant, named HTT24(4:1), has its first channel multiplied by a factor of 4, and therefore is brighter than the second channel. In the second dataset variant, named HTT24(1:4), the second channel is brighter. Using each variant, indiSplit and top two baselines for this dataset (U-Net for LPIPS, denoiSplit for PSNR) were trained and later evaluated on the (unchanged) real superimposed images. Note that the training did not use real superimposed images. We report the evaluation on real superimposed images below, which shows that IndiSplit consistently outperformed both baselines on both variants by a large margin in PSNR, showing robustness to changes in relative structure strength.
>
>
> |Model|Dataset|PSNR|SSIM|LPIPS|
> |------|--|-|-|-|
> |UNet|HTT24(4:1)|30.1|.906|.074|
> |denoiSplit|HTT24(4:1)|28.8|.854|.096|
> |indiSplit$_{0.5}$|HTT24(4:1)|32.3|.914|.069|
> |indiSplit($t_{agg}=0.82$)|HTT24(4:1)|**35.8**|**.956**|**.020**|
> ||
> |UNet|HTT24(*1:4*)|28.5|.878|.109|
> |denoiSplit|HTT24(*1:4*)|30.3|.920|.115|
> |indiSplit$_{0.5}$|HTT24(*1:4*)|32.3|.927|.067|
> |indiSplit($t_{agg}=0.25$)|HTT24(*1:4*)|**35.6**|**.955**|**.020**|
>
>
> 3. *Is the method applicable to multi-layer superposition?*
>
>     **Answer:** No, it is not. We plan to tackle semantic unmixing of more than two structures as part of our future work.
>
> 4. *If interference occurs between superimposed textures, does the method in the article remain applicable?*
>
>     **Answer:** Interference is not a prominent problem in Fluorescence microscopy. Since fluorescence emission is broadband and incoherent, it reduces the possibility of forming stable interference patterns at the detector [I, II]. Moreover, fluorescence interferometry produces interference fringes only when fluorophores lie very close to a zero-differential path length—within the fluorescence coherence length of a few microns. Fluorophores beyond this range produce no interference pattern; the detector simply measures emission intensity [III]. Thus, typical fluorescence microscopy does not detect interference patterns unless specifically designed for interferometric imaging.
>
>     [I] Bilenca A, Cao J, Colice M, Ozcan A, Bouma B, Raftery L, Tearney G. Fluorescence interferometry: principles and applications in biology. Ann N Y Acad Sci. 2008;1130:68-77. doi: 10.1196/annals.1430.038. PMID: 18596334; PMCID: PMC10902801.
>
>     [II] Sanderson MJ, Smith I, Parker I, Bootman MD. Fluorescence microscopy. Cold Spring Harb Protoc. 2014 Oct 1;2014(10):pdb.top071795. doi: 10.1101/pdb.top071795. PMID: 25275114; PMCID: PMC4711767.
>
>     [III] Bilenca A, Cao J, Colice M, Ozcan A, Bouma B, Raftery L, Tearney G. Fluorescence interferometry: principles and applications in biology. Ann N Y Acad Sci. 2008;1130:68-77. doi: 10.1196/annals.1430.038. PMID: 18596334; PMCID: PMC10902801.
>
> 5. *It is recommended to split the performance comparison in Table 1 and the model's research on mixing parameters into two separate tables.*
>
>     **Answer:** We thank the reviewer for the comment. We will restructure the Table 1 appropriately.

---

> > ### Author Response · Authors · 2025-08-05
> >
> > It is a gentle reminder to the reviewer to assess the rebuttal submitted by us. We believe we have answered most of the questions asked by the reviewer satisfactorily, and we eagerly await the reviewer's opinion on these aspects.
> > I know the reviewer, being a researcher, is super busy by default. But it will greatly benefit us if he/she finds the time for this !

---

> > > ### Comment · Reviewer_ggcu · 2025-08-05
> > > **My concerns have been addressed**
> > >
> > > After reading the authors' rebuttal, most of my concerns have been addressed.
> > > Regarding point 4, concerning interference issues, could the authors explain the differences among interference-based imaging techniques, the moiré effect, the Structured Illumination Microscopy (SIM), and the method presented in this paper？ And it is recommended to add a paragraph in the main text to discuss it.

---

> > > > ### Author Response · Authors · 2025-08-06
> > > >
> > > > We are glad to know that our responses were able to satisfy most of reviewer’s enquiries. Regarding interference-based imaging techniques, we understand the concerns raised. The straightforward answer is that we do not currently know how indiSplit will perform on interference-based imaging modalities, because, as the reviewer notes, the involvement of interference introduces uncertainty around the linearity assumptions central to our approach. Complicating matters further, multiple computational reconstruction methods exist for Structured Illumination Microscopy (SIM), each generating super-resolved images from raw interference patterns. Given this diversity, assessing indiSplit’s applicability will require systematic testing with these various reconstruction pipelines to determine its performance in each context.
> > > >
> > > > However, given the reviewer's familiarity with SIM and the moiré effect, we believe they are most likely aware of the fact that the vast majority of fluorescence microscopy modalities are not interference-based imaging techniques. Most commonly used methods—including widefield (epifluorescence) microscopy, confocal microscopy, light sheet microscopy, and several super-resolution approaches—do **not** depend on optical interference for image formation. Therefore, our method remains significant for fluorescence microscopy.

---

> > > > > ### Author Response · Authors · 2025-08-06
> > > > >
> > > > > We just want to add that in the limitations section of the main text, we will mention this fact regarding applicability of indiSplit to interference-based imaging techniques in sufficient detail.

---

### Official Review · Reviewer_qCeF · 2025-06-25

**Clarity:** 3
**Significance:** 2
**Originality:** 3
**Rating:** 4
**Confidence:** 4

**Summary:**

The authors introduce **indiSplit**, a novel computational method for fluorescence microscopy image decomposition, specifically designed to handle unknown mixing ratios between channels in multiplexed images. The paper proposes a potentially useful method, but I have significant doubts about its evaluation methodology due to reliance on metrics suited for different tasks and unrealistic test data derived from blending itself.

**Questions:**

Are the metrics you used adequate to achieve unmixing? Can you propose another metric that would measure that, for example, a downstream segmentation metric?

**Ethical Concerns:**

["NO or VERY MINOR ethics concerns only"]

**Final Justification:**

I have updated my score following the discussion. However, by authors' own admission, certain limitations such as the necessity to retrain the model for each individual dataset and the inability to tackle artefacts still persist.

**Limitations:**

Please clearly describe the practical problem that the task you are interested in is supposed to solve. Mention limitations of the metrics in assessing how well the task is addressed. Mention clearly how practical the datasets are that you are using to train and test the model. What are the limitations?

**Paper Formatting Concerns:**

Figures in the paper are missing standard elements like channel names and scale bars, which is unusual for scientific image publications.

**Quality:**

3

**Strengths And Weaknesses:**

Strengths:
The paper builds upon an existing technique by adding a regressor network that predicts the severity of the mixing ratio and a normalisation module based on this prediction, allowing degradation-aware splitting and bleed-through removal across various intensity conditions.

Major Weaknesses:

1. The paper primarily uses standard image quality metrics like PSNR (Peak Signal-to-Noise Ratio), SSIM (Structural Similarity Index), and LPIPS (Learned Perceptual Image Patch Similarity). However, these metrics fundamentally assess how closely an output resembles a specific *target* image. As a result, the authors do not measure **channel separation** effectively. The task requires direct bleed-through reduction in the biological signal, which is not represented in any of the datasets used. The task should be measured by how accurately the models are unmixing overlapping signals to reveal distinct cellular structures and quantify markers independently. PSNR/SSIM/LPIPS fail to capture this because these metrics measure quality and therefore bear secondary importance.

2.  The evaluation was performed on synthetic blended image datasets, which do not reflect the reality of unmixing unknown bleed-through from actual experimental acquisitions with variable intensities. This is likely an unrealistic assessment of performance in a real-world microscopy context where mixing ratios are unpredictable.

3.  It's unclear why absolute pixel values (as opposed to relative signal contributions or normalised intensity) are used as the basis for
evaluation, given their potential variability due to acquisition settings. In microscopy, raw intensity levels depend heavily on acquisition settings and can fluctuate naturally.

---

> ### Author Rebuttal · Authors · 2025-07-30
>
> ## Major Weaknesses:
>
> 1. *The paper primarily uses standard image quality metrics like PSNR, SSIM, and LPIPS. However, these metrics fundamentally assess how closely an output resembles a specific *target* image. As a result, the authors do not measure **channel separation** effectively. [...] PSNR/SSIM/LPIPS fail to capture this because these metrics measure quality and therefore bear secondary importance.*
>
>     **Answer:** We thank the reviewer for this question which has three very relevant enquiries for fluorescence microscopy.  We answer them in different paragraphs.
>
>     **Suitability of metrics used for semantic unmixing task:** Previous methods developed for semantic unmixing—such as uSplit [3], denoiSplit [2], and MicroSplit [5]—have all used PSNR and SSIM variants. In line with this, we also report these metrics. While LPIPS, and to some extent SSIM, reflect perceptual quality (which the reviewer refers as 'quality') rather than distortion, PSNR is especially appropriate here, as it quantifies pixelwise error. Moreover we use the range invariant version of PSNR [28] which was developed precisely for microscopy data to compare prediction on a low SNR image (low pixel values) against a high-SNR groundtruth. The SSIM variant we use, was also developed for microscopy [29]. Nonetheless, we also present segmentation results later in our response.
>
>     **Tasks not matching bleedthrough-removal**: Owing to space limitations, we were unable to include qualitative examples of bleed-through removal in the main manuscript. We invite the reviewer to consult the supplementary figures (starting from Figure S4), where qualitative plots for each dataset are presented across all mixing levels. These figures illustrate numerous instances where faint structures visible in the input are successfully removed.
>
>     **Suitability of metrics for bleedthrough-removal task:** We acknowledge that, because bleed-through represents a minor unwanted signal, even input-to-target comparisons will yield relatively high metric values. To address this, we had included an “Input vs. Target” row in Table 1 showing the metric comparison between the target and input. Additionally, the fact that metric values are higher in this setting does not pose an issue, as all baselines are affected similarly; the approach achieving superior performance will continue to deliver better metric values.
>
> 2. *The evaluation was performed on synthetic blended image datasets, which do not reflect the reality of unmixing unknown bleed-through from actual experimental acquisitions with variable intensities....*
>
>     **Answer:** In the following paragraphs, we present our three key responses: (a) the usefulness of synthetically blended semantic unmixing, (b) our work involving real superimposed images, and (c) an additional experiment conducted with real superimposed images, showcasing generalizability of indiSplit to different mixing-ratios.
>
>    **Synthetically blended semantic unmixig is useful:** In Supplementary Section C.3 of Microsplit [5] , the authors evaluated the performance impact of training a semantic unmixing model using synthetically summed inputs, then testing it on real acquired inputs. They found that the performance drop was not detrimental. Furthermore, we have consulted with several microscopists about using real microscopy images as targets and their sums as inputs. Since superposition in fluorescence microscopy is indeed linear [19–21], microscopists—although naturally preferring real superimposed images—generally agree with us in the use of synthetically summed inputs derived from real target channels.
>
>     **Results on real superimposed images**: In Table 2 of the main manuscript, we report results on real superimposed images; however, these correspond to the balanced input regime, where the input contains roughly equal contributions from both structures (Line 206, main manuscript), which is not an ideal dataset for this work. To date, there is no publicly available dataset that fits our use case, which requires both individual target channels and corresponding superimposed input images with varying mixing ratios. Consequently, using synthetic sums remains the only practical approach currently available to study variable mixing ratios. However, we did an experiment to show the generalizability of indiSplit across different mixing-ratios, which we describe next.
>
>    **An additional experiment with real inputs:**  We conducted an experiment demonstrating that indiSplit outperforms baselines when the relative strengths of structures in real superimposed input differ from those in synthetic sums. Using the HT-T24 dataset, we created two dataset variants by multiplying all pixel values in one structure channel by 4. So, the first dataset variant, named HTT24(4:1), has its first channel multiplied by a factor of 4, and therefore is brighter than the second channel. In the second dataset variant, named HTT24(1:4), the second channel is brighter. Using each variant, indiSplit and top two baselines for this dataset (U-Net for LPIPS, denoiSplit for PSNR) were trained and later evaluated on the (unchanged) real superimposed images. Note that the training did not use real superimposed images. We report the evaluation on real superimposed images below, which shows that IndiSplit consistently outperformed both baselines on both variants by a large margin in PSNR, showing robustness to changes in relative structure strength.
>
> |Model|Dataset|PSNR|SSIM|LPIPS|
> |-|-|-|-|-|
> |UNet|HTT24(4:1)|30.1|.906|.074|
> |denoiSplit|HTT24(4:1)|28.8|.854|.096|
> |indiSplit($t_{agg}=0.82$)|HTT24(4:1)|**35.8**|**.956**|**.020**|
> ||
> |UNet|HTT24(*1:4*)|28.5|.878|.109|
> |denoiSplit|HTT24(*1:4*)|30.3|.920|.115|
> |indiSplit($t_{agg}=0.25$)|HTT24(*1:4*)|**35.6**|**.955**|**.020**|
>
> 3. It's unclear why absolute pixel values (as opposed to relative signal contributions or normalised intensity) are used as the basis for evaluation, given their potential variability due to acquisition settings. In microscopy, raw intensity levels depend heavily on acquisition settings and can fluctuate naturally.
>
>     **Answer:** In fluorescence microscopy images, there is considerable variability in pixel intensity levels across different acquisitions due to varying acquisition settings. However, this variability does not impact our work since we do not perform comparisons across different datasets; each model is trained and evaluated on the same dataset split. Moreover, evaluating in a normalized space allows for the possibility of using different normalization methods, which can lead to variations in metric values for the same prediction. This may pose challenges for future studies attempting to directly compare or cite our metric results.
>
> ## Questions
> 1. *Are the metrics you used adequate to achieve unmixing? Can you propose another metric that would measure that, for example, a downstream segmentation metric?*
>
>    **Answer:** Regarding the suitability of metrices used by us, please see our answer to Major weakness no. 1. We agree with the reviewer that improved metrics would be beneficial and not much work has been done to design metrics specifically for microscopy data. However, every metric has its limitations. While segmentation could be used to assess quality, it overlooks the relative intensity variations within a labeled object. Since our predictions involve continuous pixel values rather than just two discrete classes (foreground and background), segmentation alone cannot fully capture the quality of the prediction. For example, a method producing artifacts within a labeled object might still receive a similar ranking to a method without such artifacts, as segmentation may fail to detect these subtle errors.
>
>     That said, for the bleed-through removal task (w=0.7, 0.8, 0.9), we have now used the test set of BioSR dataset to evaluate segmentation as a downstream task with Featureforest (max_depth=9,numtrees= 450, encoder=SAM2_large), a recently developed segmentation method [I]. To avoid model bias, we manually annotated the ground truth and trained Featureforest using these annotations alongside model predictions (If we had annotated the predictions instead, it would raise concerns about the consistency and quality of annotations when doing it for indiSplit and for the baseline models). We trained $4 × 2 × 3 =24$ Featureforest models—one per model, channel, and mixing ratio. Evaluation used the Dice dissimilarity from scipy (lower is better). Among baselines, segmentations from indiSplit predictions best matched those of the ground truth channel, thereby perfectly correlating with PSNR based assessment. **We will add these segmentation-based evaluation results to the manuscript.**
>
> | Model               | .           | DICE | DISSIM. |  SCORE | ↓ | . |
> |-------------------|---------|--------|---------|--------|--------|-------|
> | .                        |             | Ch0    | .          | .         | Ch1    | .        |
> | .                        |w=0.7.  | w=0.8 |w=0.9  |w=0.7 |w=0.8. |w=0.9|
> denoiSplit           | 0.077   | 0.073 | 0.065  | 0.064  | 0.061 | 0.056 |
> $\mu$Split$_D$ | 0.055    | 0.051 | 0.047  | 0.051 | 0.047 | 0.038 |
> InDI                     | 0.055.  | 0.05   | 0.044  | 0.046  | 0.045 | 0.045 |
> indiSplit              | **0.048** | **0.039** | **0.035** | **0.038** | **0.032** | **0.026** |
>
> [I] Seifi et al. FeatureForest: the power of foundation models, the usability of random forests. npj Imaging 3, 32 (2025). https://doi.org/10.1038/s44303-025-00089-9
>
> ## Paper Formatting Concerns:
>
> *Figures in the paper are missing standard elements like channel names and scale bars, which is unusual for scientific image publications.*
>
> **Answer:** We thank the reviewer for bringing to our attention missing scale bars and channel names. We will certainty update them.

---

> > ### Comment · Reviewer_qCeF · 2025-08-04
> >
> > Q1: The authors mention that they present the segmentation results later in the response. These results, however, seem to be missing. Could you please be more specific here?
> >
> > Q2: While authors state that weighted linear mixing is sufficient to emulate the bleedthrough, respectfully, I disagree. Yes, it is true that weighting could eliminate differences in quantum efficiency of fluorescent dyes. However, a fluorescence image is comprised not only of the true signal. In biological specimens, autofluorescence (AF) plays a major role. AF can differ significantly between organelles, cell types, sample preparation techniques and excitation wavelength. This is exactly why an unmixing validation of the experimental images is crucial. Here are some examples of the datasets the authors may consider:
> > https://idr.openmicroscopy.org/webclient/?show=project-1451
> > https://zenodo.org/records/15496000
> > https://www.nature.com/articles/s41597-024-03064-y
> >
> > Particularly in the latter, author could find a signifcant bleedthrough in DAPI, CFP and GFP channels. It would be interesing to see how does the model perform.
> >
> > Q3: Given the authors reply, how do authors guaranty absolute pixel value comparability within the dataset. Are they absolutely certain different images with the dataset are comparable?

---

> > > ### Author Response · Authors · 2025-08-04
> > >
> > > We thank the reviewer for engaging with us after the rebuttal! Please find our response below.
> > >
> > > *Q1: Missing segmentation results.*
> > >
> > > **Answer**:  The segmentation results (a table showing results from 24 segmentation models), are presented at the end of our rebuttal to this reviewer's review, specifically within the ‘Questions’ section. Those results are provided for the question: *‘Are the metrics you used adequate to achieve unmixing? Can you propose another metric that would measure that, for example, a downstream segmentation metric?’*. Due to restrictions by NeurIPS, we could not include images (qualitative segmentation results) and therefore reported the segmentation results solely in quantitative terms in a table using the DICE dissimilarity metric, where lower values indicate better performance. It is visible to us in our rebuttal. In case it is not for the reviewer, we can paste it as our next comment.
> > >
> > > *Q2: On Bleedthrough removal, autofluorescence.*
> > >
> > > **Answer**: We understand that the reviewer would be delighted (and so would we :D ) if our method could remove the artifacts which present themselves onto the useful signal. However, as the reviewer rightly points out, there are several reasons for artefacts, namely auto-fluorescence, non-specific fluorescence, out-of-focus fluorescence, bleed-through etc. However, we limit ourselves to bleed-through removal which happens when the content of one channel bleeds into another. Technically speaking, bleedthrough does not involve auto-fluorescence or other factors which often becomes the source of additional artefacts.
> > >
> > > The three datasets reviewer provided ((a) axon-nucleus-oligodendrocite data, (b) a 5D data and (c) a dataset for artefact removal) are excellent datasets to evaluate the bleed-through removal performance, but do not have what we need to train indiSplit and therefore cannot be used. If C1 and C2 represent artefact-free groundtruth images of two structures, an input with bleedthrough can, for instance, be represented by x1= 0.9C1 + 0.1C2  or x2= 0.1C1 + 0.9C2 and our objective would be to estimate C1 from x1 and C2 from x2. To train our model, we need C1, C2 and optionally the corresponding input. The datasets reviewer pointed generally have x1 and/or x2 (images with bleedthrough), but donot have both C1 and C2 (clean data without bleedthrough). As the reviewer can now understand, indiSplit requires a specific acquisition for the bleedthrough task, which we plan to have through collaborations in near future.
> > >
> > > We would also like to emphasize that this is a deep learning–focused work and focuses on the superposition between the structures exclusively, and therefore caters to both bleed-through removal and the semantic unmixing task.  In doing so, it does make novel technical contributions. Looking ahead, there is potential to submit to journals such as Nature Methods or other leading biologically oriented journals, where the goal would be to integrate technical advances from multiple relevant areas—including denoising of pixel-independent noise, semantic unmixing, and autofluorescence management—and to propose a comprehensive artefact removal pipeline validated on a range of experimental datasets including the ones pointed out by the reviewer.
> > >
> > > *Q3: Pixel values are comparable?*
> > >
> > > **Answer**: We expect that potential users of this work will initially collect a new dataset using their own microscope and then train the indiSplit network on this data. Once trained, they can apply the resulting model to future acquisitions of superimposed inputs captured under similar microscopy settings. Our goal was not to develop a foundational, "download-and-use" model that requires training on an extensively diverse dataset with multiple acquisitions taken in different configurations. In this context, we find comparing pixels within a dataset a reasonable choice.  We will make this point clear in the main manuscript.
> > >
> > > For most datasets we used (including HTT24, HTLIF24 and PaviaATN), all image stacks were acquired with the same configuration (as stated in their publicly available doc.) and so, our normalization assumptions are suitable.

---

> > > > ### Comment · Reviewer_qCeF · 2025-08-05
> > > >
> > > > I thank the authors for their comment. I have now increased my score, taking into account the hard limitations of the work brought forward by the authors.

---

### Official Review · Reviewer_n12V · 2025-07-04

**Clarity:** 3
**Significance:** 1
**Originality:** 3
**Rating:** 4
**Confidence:** 4

**Summary:**

This work introduces a new method, indiSplit, to recover probe-specific signals from fluorescence microscopy images in which two signals have been mixed. Such digital demultiplexing methods could overcome technical limitations in fluorescence microscopy, by allowing for effective “overloading” of individual channels with multiple (instead of just one) structure-specific probes, thereby increasing the number of cellular structures that can be captured and cleanly related through multiplexed image acquisition pipelines. In a well-structured paper, the authors present the roots of their method, introduce technical modifications, and empirical evidence across several relevant datasets, along with numerous technical controls, that generally support their merit. However, certain methodological aspects remain unclear, one key ablation is missing, and the practical significance of indiSplit may be modest.

**Questions:**

1. To address weakness (1), I would ask that the authors address the relationship of the normalization module to instance norm / intrinsic normalizations of $c_t$ should be added.
2. Weakness (3) could be addressed by adding sufficient detail to the text or appendix to clarify the extent to which indiSplit is iterative.
3. The authors should affirm that InDI was faithfully implemented, i.e. as an iterative system. If not, the authors should clearly highlight this (including changing the acronym), and further explicitly delineate how indiSplit is different from their InDI implementation.
4. Depending on the difference between InDI and indiSplit, the authors should consider extending ablation experiments (weakness (2)) around the normalization question to other baselines. For example, for baselines trained by the authors, and in particular for InDI, instance norm of $c_t$ (a very cheap operation) could be explored.
5. If the authors could provide additional empirical data on the extent to which indiSplit generalizes to real images, ideally collected under relevant potential out-of-distribution settings (i.e. different technical circumstances such as a different microscope), that could substantially strengthen the significance of the work.

I expect addressing most of these key weaknesses to be feasible, and I would gladly increase my scores in response. I further extend my sincere apologies to the authors for the late submission of this review.

**Ethical Concerns:**

["NO or VERY MINOR ethics concerns only"]

**Final Justification:**

Over an extensive and indeed rather collaborative rebuttal period, the authors have provided many critical clarifications along with a substantial set of additional experiments.

- In particular, it became clear that the proposed **normalization approach** had not been adequately described in the original manuscript. It also appears that, as per additional results produced by the authors in response to *Weakness 1*, the same effect can be achieved through a well-known approach, i.e. by applying Instance Normalization (IN), even though there may be subtleties with respect to how IN interacts with the proposed Reg module. The authors affirm that they will thoroughly revise Section 3.2. to account for the substantial gaps in formalism and to incorporate their results in IN. While these gaps could well have led to me downgrading my evaluation, I (1) trust that the corrections we have discussed will be implemented, and (2) note that the empirical results that normalization of $C_t$ (by one method or another) helps stand regardless (further supported by additional ablations), based on which I expect such normalization to become default practice in this application domain.

- The authors have **clarified that the method is not iterative in nature**; the method will thus be renamed to scSplit, references to time / removed, and variable name $t$ changed accordingly.

- Finally, we managed to figure out what the images in Table 2 actually correspond to, i.e. that they are indeed "real enough" to count as **a reasonable test of generalization of scSplit to non-synthetic images**. Relevant methodological details will be added to the paper. Generalization to actual application relevant images, and beyond that, to independent test data (OOD), remains unclear. This may be because suitable datasets are not available to the authors, but it's a limitation nonetheless, which compounds with the need to train probe-set specific models for every intended application.

As such, the paper has been clearly improved: **The paper represents a technically meaningful step forward, even as limitations regarding the practical significance of the method remain.**

**Limitations:**

yes

**Quality:**

2

**Strengths And Weaknesses:**

**Strengths:**

1. **A well-presented paper:** The authors present indiSplit in a well-written manuscript, with high-quality figures that clearly communicate motivation, reasoning, experimental design and results. Conclusions are well-calibrated to the level of empirical evidence provided.
2. **Clear application target:** The method has a clear real-world potential to address a concrete limitation in fluorescence microscopy, one of the most foundational and widely used techniques in biomedical research.
3. **High degree of experimental rigor:** Experimental include relevant technical controls which benefits attribution of empirical improvements to their method.

**Weaknesses:**

1. **Normalization module vs instance norm:** While the authors reasoning for the normalization module is clear, and the implementation effective, it appears that, to ensure $σ^2=1$ for input images during training, instead of precomputing a statistical memory bank, the same effect could be trivially achieved by applying instance normalization (i.e. using intrinsic statistics for normalization) to any input $c_t$ on the fly. If there are compelling technical reason against this, they should be discussed and empirically validated. While the use of instance normalization is well-established in computer vision, and in particular for fluorescence microscopy applications in which absolute intensity variations between technical replicates is assumed to be a major source of noise (e.g. cellular profiling [1]), documenting its empirical benefit in this context would still be useful.

2. **Ablation of normalization module is missing:** Experiments to delineate the specific impact of ensuring $σ^2=1$ for any $c_t$ (as opposed to the impact of the time regressor module) are missing.

3. **Unclear to what extent indiSplit is iterative:** The authors explain their study as building upon “InDI: Image Restoration by Direct *Iteration*” [2], but it remains unclear to what extent indiSplit, as well as the authors implementation of InDI for that matter, actually implements an iterative process of image of image restoration. This seems rather significant, because InDI’s primary innovation is exactly the iterative, stepwise recovery of source images in ill-posed settings. Method, experimental, and Appendix provide no information on this regarding indiSplit; Fig. 1 seems to show a one-shot restoration process. Moreover, the use of the Reg network seems more compatible with a single step of inference. In InDI, models learn to restore images over a consistent number of steps N, i.e. t=1 is implicitly set for any input image during inference and the image is restored over N steps of size δ. If indiSplit is indeed iterative in this sense, then it would seem that for any estimated $t_{agg}$<1, either N or δ would need to be adjusted dynamically. Whether or how this is done is unclear.

4. **Generalization seems uncertain:** While indiSplit outperforms a strong set of baselines by substantial margins over datasets in which channels are artificially mixed, no clear performance advantage is apparent for real images. Combined with the fact that indiSplit seems to require two task-specific models to be trained for any particular unmixing setup (i.e. models trained to unmix some channels 1 and 2 are not expected to work for channels 1 and 3), and that these may well fail to generalize to images (with matched probes) collected on difference microscopes, the practical significance of indiSplit will likely be modest.

5. **Minor weaknesses:** There are some minor spelling mistakes. The authors may further want to sharpen their description of the distinction of natural vs. fluorescence microscopy images with respect to their value-distributions / histograms. In particular, the sentence in line 128 “For instance, it is common to have the maximum pixel intensity for a noisy acquisition to be less than where as [spelling mistake] the pixel intensities can easily be larger than 4000 for less noisy acquisitions.” should be refined to take account of varying image bit-depths (often 8-bit for natural images, hence dividing by 255 makes sense, but commonly 12- or 16-bit for microscope cameras).

[1] Pernice, W.M., Doron, M., Quach, A., Pratapa, A., Kenjeyev, S., De Veaux, N., Hirano, M. and Caicedo, J.C., 2023. Out of distribution generalization via interventional style transfer in single-cell microscopy. In Proceedings of the IEEE/CVF Conference on Computer Vision and Pattern Recognition (pp. 4326-4335).
[2] Delbracio, M. and Milanfar, P., 2023. Inversion by direct iteration: An alternative to denoising diffusion for image restoration. arXiv preprint arXiv:2303.11435.

---

> ### Author Rebuttal · Authors · 2025-07-29
>
> **Weaknesses:**
>
> 1. *Normalization module vs instance norm:*
>
>     **Answer:** The reviewer rightly pointed out that applying instance normalization (IN) in the first layer can, in principle, also mitigate the normalization issue we discovered for the semantic unmixing task. However, we have both empirical and domain-specific reasons to prefer our approach.
>
>      To evaluate this, we conducted experiments on the Hagen et al. dataset using instance normalization. We trained two additional baselines: (1) InDI (firstIN), with instance normalization as the first layer, and (2) InDI (allIN), replacing all group normalization layers with instance normalization plus a first-layer instance normalization. Similar variants were also created for the indiSplit network. Results (below) show our original indiSplit model outperforms most baselines. Notably, the 'firstIN' variant performs better than the 'allIN' variant. Results for the PaviaATN task with 'firstIN' are also reported in which case as well, indiSplit has clear out-performance for the bleedthrough regime (Dom.) and is competitive in others.
>
> |Dataset|Model|PSNR|SSIM|LPIPS|PSNR|SSIM|LPIPS|PSNR|SSIM|LPIPS|
> |-----------|-----------------|-|-|-|-|-|-|-|-|-|
> ||||Dom. |||Bal.|||Weak.|
> |Hagen et al.|inDI(allIN)|36.5|0.987|0.026|32.4|0.968|0.060|27.9|0.914|0.132|
> |Hagen et al.|inDI(firstIN)|36.7|0.987|0.032|33.5|0.974|0.052|26.6|0.881|0.154|
> |Hagen et al.|indiSplit(allIN)|36.6|0.988|0.025|32.4|0.967|0.061|27.9|0.916|0.131|
> |Hagen et al.|indiSplit(firstIN)|40.1|0.993|0.016|**34.0**|0.976|0.051|29.0|0.924|0.129|
> |Hagen et al.|indiSplit|**40.9**|**0.994**|**0.011**|33.9|**0.977**|**0.046**|**29.3**|**0.934**|**0.123**|
> ||
> |PaviaATN|inDI(firstIN)|28.8|0.950|0.102|27.1|0.903|**0.150**|21.3|0.768|**0.248**|
> |PaviaATN|indiSplit(firstIN)|33.5|0.975|0.042|**27.7**|0.906|0.172|**24.8**|**0.825**|0.440|
> |PaviaATN|indiSplit|**35.1**|**0.977**|**0.033**|27.6|**0.907**|0.155|24.3|0.823|0.377|
>
> From a technical perspective, when normalization is performed acquisition-wise—as we do during inference—rather than patch-wise (instance normalization), it equally accounts for the variability in pixel values observed in microscopy data caused by different acquisition settings. At the same time, this approach allows us to avoid the drawbacks associated with instance normalization. Instance norm has been reported to degrade discriminative performance in tasks where intensity clues or instance specific contrast is relevant [I,II]. In microscopy, intensity cues are important. For instance, the background can be easily distinguished from the 'content-less' interior of an organelle like nucleus simply by comparing average intensity value. However, instance norm can make it difficult as it normalizes per patch. This will be more true for smaller patch sizes, typically used for semantic unmixing (patch size of 64 and 128 were used by uSplit [3] and denoiSplit [2] respectively).
>
> [I] Hyeonseob Nam, Hyo-Eun Kim. Batch-instance normalization for adaptively style-invariant neural networks. (NIPS'18).
>
> [II] Dmitry Ulyanov, Andrea Vedaldi, Victor Lempitsky. Improved Texture Networks: Maximizing Quality and Diversity in Feed-forward Stylization and Texture Synthesis. CVPR 2017.
>
>
>
> 2. *Ablation of normalization module is missing:*
>
>      **Answer:** We have now conducted the requested ablation study where the indiSplit model was trained without applying our custom normalization method. Instead, we used the commonly adopted approach in iterative inference models like InDI, where the two images are normalized first using mean-std normalization and then combined via a convex combination to form the intermediate input $ x_t $. Below, we present the results of indiSplit using this 2-image normalization scheme across four datasets. For each task, we trained the Reg and $Gen_i$ networks under this normalization approach. Across all tasks and all input regimes, it is evident that the PSNR values of the original indiSplit model (as reported in Table 1 of the main manuscript) outperform those reported here, with the sole exception of the task on PaviaATN dataset under the dominant input regime, where the PSNRs are equal.
> |Model|PSNR|SSIM|LPIPS|PSNR|SSIM|LPIPS|PSNR|SSIM|LPIPS|
> |------|-|-|-|-|-|-|-|-|-|
> |||Dom. |||Bal.|||Weak.|
> |PaviaATN|35.1|0.978|0.032|25.9|0.895|0.202|23.7|0.813|0.492|
> |Hagen|39.4|0.990|0.017|30.8|0.962|0.067|27.8|0.923|0.120|
> |HTT24|42.7|0.992|0.003|37.1|0.973|0.011|33.7|0.934|0.034|
> |BioSR|39.3|0.979|0.017|33.6|0.936|0.059|28.1|0.874|0.146|
>
> 3. *Unclear to what extent indiSplit is iterative:*
>
>     **Answer:** Our implementation of InDI is exactly as mentioned in the InDI paper [22], and therefore has the possibility of doing iterative inference with any number of steps (see  line 240 in model/ddpm_modules/joint_indi.py, given in supplementary.zip).
>
>     However, in the Figure 3 of InDI paper, its authors quantitatively show that more iterations lead to worse PSNR (higher distortion) but improved perceptual quality. When working with predictive models on biological data, one can argue that in the perception-distortion tradeoff [I], the motivation is to get the model with a better fidelity to the recorded image (lower distortion). This would not be true if the motivation was to generate synthetic datasets, in which case better perceptual quality might be the major goal. With a focus on better fidelity,  we use one step prediction for both InDI and inDiSplit.
>
>     Note that training in InDI is not iterative and so this choice does not affect in any way how InDI is trained.
>
>     Finally, we chose InDI as the basis for our work not because of its ability to also be used iteratively, but due to its fitting inductive bias, i.e. the fact that the convex combinations of source and target images coincides perfectly well with the superposition of two differently intense structures being labeled in a single image channel on a fluorescence microscope. (For more details, see L92-L100 in the main manuscript).
>
>     That all being said, iterative inference in indiSplit can done in the following way: Instead of starting from $t=1$ as done in InDI, one needs to start from the value of $t$ estimated with $Reg$ network. $\Delta$, the unit decrement in $t$, can simply be computed as $\Delta =t_{start}/totalSteps$. This is implemented in code (see line 93 in model/ddpm_modules/indi.py, given in supplementary.zip). The only other change that needs to happen is to ensure that $E[\mu(x_t)] =0$ and $E[\sigma(x_t)]=1$, where $\mu()$ and $\sigma()$ represents the mean and the standard deviation operators. This can easily be achieved by a normalization step in each step of the iterative inference. However, we do not focus on iterative generation in this work since it is not helpful in the context of the application use-case, where we clearly want, as described above, minimal distortion (maximum data fidelity). If the reviewer continues to consider indiSplit distinct from inDI, we would be glad to rename it to scSplit (severity cognizant split) or any other name the reviewer suggests.
>
> [I] Y. Blau and T. Michaeli,The Perception-Distortion Tradeoff, CVPR 2018.
>
>
> 4. *Generalization seems uncertain:*
>
>      **Answer:** We believe that the results on real superimposed images were seen in a different light by the reviewer. Here, we present our point of view, and also show an additional experiment with real inputs to further prove our point.
>
>    **Why we are happy with Table 2:** The HTT24 and HTLIF24  datasets we used contains real inputs with balanced mixing ratios ((Line 206, main manuscript), as they were generated in MicroSplit [5] whose focus was not on asymmetric mixing. We see indiSplit’s strong performance within this balanced mixing-ratio range—where baselines were designed to excel—as a positive indicator of its versatility across mixing ratios.
>
>      **An additional experiment:**  We conducted an experiment demonstrating that indiSplit outperforms baselines when the relative strengths of structures in real superimposed input differ from those in synthetic sums. Using the HT-T24 dataset, we created two dataset variants by multiplying all pixel values in one structure channel by 4. So, the first channel is brighter than the second channel in the first dataset variant. The case is opposite for the second dataset variant. Using each variant, indiSplit and top two baselines (U-Net for LPIPS, denoiSplit for PSNR) for this dataset were trained and evaluated on the (unchanged) real superimposed images. Results presented below shows that IndiSplit consistently outperformed both baselines on both variants by a large margin in PSNR, showing robustness to changes in relative structure strength.
>
> |Model|Dataset|PSNR|SSIM|LPIPS|
> |------|--|-|-|-|
> |UNet|HTT24(4:1)|30.1|.906|.074|
> |denoiSplit|HTT24(4:1)|28.8|.854|.096|
> |indiSplit$_{0.5}$|HTT24(4:1)|32.3|.914|.069|
> |indiSplit($t_{agg}=0.82$)|HTT24(4:1)|**35.8**|**.956**|**.020**|
> ||
> |UNet|HTT24(*1:4*)|28.5|.878|.109|
> |denoiSplit|HTT24(*1:4*)|30.3|.920|.115|
> |indiSplit$_{0.5}$|HTT24(*1:4*)|32.3|.927|.067|
> |indiSplit($t_{agg}=0.25$)|HTT24(*1:4*)|**35.6**|**.955**|**.020**|
>
>    **Generalizability across structure types:** We acknowledge the reviewer’s point that models trained to unmix channels 1 and 2 may not generalize to channels 1 and 3. However, given the early stage of research, this is less critical. Semantic unmixing is still nascent, with only two deep-learning methods (uSplit , denoiSplit ) published in top computer vision/ML venues—both using synthetic mixes. MicroSplit, recently uploaded to arXiv, is the only work addressing real superimposed images.
>
>
> 5. *Minor weaknesses:*
>
>     **Answer:** We thank the reviewer for his comment. We will spend time to fix all typos.

---

> > ### Author Response · Authors · 2025-08-05
> > **Pending Author-Reviewer discussion**
> >
> > It is a gentle reminder to the reviewer to assess the rebuttal submitted by us. We believe we have answered all of the questions asked by the reviewer, namely the utility of normalization module, normalization module ablation, using instance norm instead of our normalization module, performance on real images and iterative nature of inDiSplit.  While we are quite satisfied with the results of these additional experiments and analysis, we eagerly await the reviewer's opinion on these aspects.
> > I know the reviewer, being a researcher, is super busy by default. But it will greatly benefit us if he/she finds the time for this !

---

> > > ### Comment · Reviewer_n12V · 2025-08-05
> > >
> > > I appreciate the authors effort to clarify / address key concerns I raised.
> > >
> > > 1. **Normalization**: The authors provide additional experimental results regarding the impact of replacing the proposed normalization module with Instance Normalization (IN) layers. As expected, this approach performs comparably; I disagree with the authors notion that indiSplit "has clear out-performance" in any pertinent comparison here. Instead, indiSplit sometimes performing marginally better, sometimes marginally worse than indiSplit(firstIN). I agree with the authors about the drawbacks about applying IN to patches of ${c_t}$ / in place of group or other, larger-context normalization layers. However, this is again trivially avoided by applying IN to to any input ${c_t}$ itself, instead of image patches, as I proposed, and as the authors do during inference: quoting from the authors response to R4 (Lyze), if "one simply calculates the statistics (mean and standard deviation) from the sampled test-image patches, and then use those values for normalization", normalization indeed "becomes straightforward" and it is exactly this operation that IN implements if applied to ${c_t}$. There appears to be no reason whatsoever to use the more complicated, computationally more expensive, and less elegant proposed "Severity Cognizant Input Normalization module" over this simple operation. The authors should adjust their manuscript accordingly.  which I expect will also address confusion regarding this aspect of the paper raised by other reviewers. Note that the authors finding that ensuring zero mean and unit standard deviation for input images is helpful for this application, stands irrespectively (see below).
> > >
> > > 2. **Normalization ablation**: Additional ablations on the normalization module are appreciated. Since, again, this should be exactly equivalent to IN(${c_t}$), the authors should be able to reuse these results as their change their manuscript with respect to the above.
> > >
> > > 3. **Iterative nature**: The authors have clarified that indiSplit is in fact not iterative. Since the acronym "indi" explicitly includes the term "iteration", I would indeed suggest that the authors avoid reference to it; changing to scSplit seems like a good idea. The authors should adjust their manuscript to make this difference crystal clear, e.g. by revising lines 88-100, and as suggested by the authors in response to R4 ("ambiguity around the meaning of t"), by avoiding the term "time" in the context of this variable (renaming it might help). The authors further claim that (1) an iterative application of their method is readily possible, but (2) not helpful; however, this remains to be empirically shown. As one follow-up question: can the authors confirm that InDI baselines are applied iteratively?
> > >
> > > 4. **Generalization**: There seems to be substantial confusion about what "real images" means (and should mean) with respect to demonstrating that the proposed method works in practice, as a primary concern also echoed by other reviewers (e.g. qCeF). Specifically, the value proposition of methods to computationally de-multiplex fluorescence microscopy images is to increase the number of structures that can be specifically visualized (i.e. to the exclusion of others) in fluorescence microscopy imaging to > 1 per channel. "Real images" therefore must mean images in which at least two specific structures, say the mitochondrial matrix and F-actin, were indeed stained and imaged in the same channel (say both stained with an Alexa-488 probe or equivalent), and are subsequently computationally disentangled to enable the analysis of these cellular substructures as if they had been collected in separate channels. After attempting to learn more about the HTT24 and HTLIF24 datasets it is entirely unclear to me to what extent images pertaining to Table 2 are "real" in this respect, or what exactly "real superimposed images" is supposed to mean. The descriptions provided by the Microsplit paper [5] (specifically, page 13 of Ashesh et al) are also insufficient to discern this. Incidentally, I noticed that while the authors say that they "tackle the SOX2 vs. MAP2 task" in the HTT24 dataset (line 202), this doesn't line up with the descriptions of this dataset by [5]: according to [5] HT-**T**24 contains channels for SOX2 and Grasp65 (for fixed E37 ferret brain sections), whereas it is HT-**H**24 that contains channels for SOX2 and MAP2 for (iPSC-derived DIV25 dorsal forebrain organoids). Apart from this minor confusion which the authors can easily rectify, the authors will need to very clearly explain what task is tested in Table-2. At this point, I have the strong impression that, contrary to my previous understanding, Table-2 does in fact not constitute an evaluation on "real" images and that the extent to which scSplit may be useful in practice this remains entirely unclear. The additional experiments, while appreciated, do not address this question.

---

> > > > ### Author Response · Authors · 2025-08-06
> > > >
> > > > We recognize that the reviewer has dedicated a significant amount of time to this review, and we sincerely appreciate the effort, regardless of the final decision. We also want to clarify that all results presented to all reviewers during the rebuttal phase will be included either in the main text or in the supplementary material. Given the detailed nature of the reviewer’s comments, to ensure clear communication, we will provide a summary response addressing the reviewer’s remarks on Normalization in this comment, followed by a more point-wise reply on Normalization in the next comment. Responses to the remaining queries will be addressed in subsequent comments by the end of today.
> > > >
> > > > ## Normalization:
> > > >
> > > > We are happy that the author acknowledges our contribution regarding normalization at the end of his/her comment by stating ‘…in the authors finding that ensuring zero mean and unit standard deviation for input images is helpful for this application, stands irrespectively’. However, for correctness sake (and to ensure our paper has higher chances of acceptance :D ), it is necessary to highlight issues in  three specific points raised by the reviewer: (a) we believe there is a crucial misunderstanding regarding the word single acquisition: it is not one image. (b) the suggestion that our results (main and/or rebuttal) could be directly reused if we were to agree with IN-based normalization does not align with our understanding; and (c) the view that our normalization strategy is more computationally demanding than simply using IN appears to be mistaken.

---

> > > > > ### Author Response · Authors · 2025-08-06
> > > > >
> > > > > 1. *The reviewer referred to our response to reviewer R4 (Lyze) and said, “it is exactly this operation that IN implements if applied to $c_t$”, meaning using IN on $c_t$ is essentially what we do during test time. This appears to be a misunderstanding arising from arguably an incorrect assumption regarding what single acquisition means.*
> > > > >
> > > > > **Response:** In microscopy, a "single acquisition" usually means all data from one imaging session, including multiple z-stacks (each with dimensions Z × H × W). Thus, it’s a collection of image sequences, not a single image. During testing, we compute scalar mean and standard deviation from randomly sampled patches across the entire acquisition, which differs from applying Instance Normalization (IN) to individual frames.
> > > > >
> > > > > From a microscopy standpoint, different z-stacks can vary significantly in structural density (one z-stack can easily contain much more signal than the other), and the initial and final frames of a z-stack are often empty or contain little information. Due to these variations in frames, we do not recommend applying IN to individual frames.
> > > > >
> > > > > 2. *The reviewer stated “Additional ablations on the normalization module are appreciated. Since, again, this should be exactly equivalent to IN(), the authors should be able to reuse these results as their change their manuscript with respect to the above.”*
> > > > >
> > > > > **Response:** The statement that “this should be exactly equivalent to IN” is incorrect. Neither the rebuttal nor the paper’s results would match if IN were applied to
> > > > > $c_t$. As stated earlier, our test-time normalization differs from the reviewer’s IN approach. However, even if a single acquisition were considered a single image, the reviewer’s argument can still be argued against, as explained next. Our method guarantees that for a random variable $x_t$ representing an input patch at time t, the expected value $E[\sigma(x_t)]=1$ during training. Applying IN to full frames ensures that $\sigma(c_t)=1$ for every $c_t$, and thus $E[\sigma(c_t)]=1$. However, since $x_t$, being a patch, is a sub-region of some $c_t$, it would be incorrect to assume that $E[\sigma(x_t)]=1$. So, if we load the pre-trained weights, and use instead IN based normalization, we will get different results.
> > > > >
> > > > > 3. *'The reviewer stated "There appears to be no reason whatsoever to use the more complicated, computationally more expensive, and less elegant proposed Severity Cognizant Input Normalization module over this simple operation.’*
> > > > >
> > > > > **Response:** indiSplit outperforms indiSplit(firstIN) across three input regimes and two datasets, with an average PSNR gain of 0.33 dB. Since both models share the same network architecture and incorporate our key innovations—(a) the Reg network, (b) t aggregation, and (c) correction of the normalization issue—the performance gap could not be large and thus this performance gap can arguably be treated as an evidence of superiority of our normalization module over IN.
> > > > >
> > > > > The claim that IN is computationally cheaper is incorrect. Our normalization module’s extra computation occurs only once before training, when means and standard deviations for each $t$ are calculated—a fast pre-processing step (e.g., about 7 minutes for Hagen et al. in a naive implementation). After this, training (taking ~3 days on a Tesla V100) normalizes input patches using these precomputed values, adding no overhead during training.
> > > > >
> > > > > Moreover, we provide evidence that the IN-based approach could actually be more computationally demanding, particularly when using smaller patch sizes. In the reviewer's suggested approach, IN internally needs to compute the standard deviation over the full frame $c_t$ for each input patch $x_t$, and this computation is performed within the dataset class, and therefore must happen on the CPU. With frame dimensions of several thousand pixels  found typically in microscopy datasets (e.g., 2720x2700 for PaviaATN, 2048x2048 for Hagen et al., 4096x4096 for Chicago-Sch23 [5] ), we conducted a quick estimation: On our high-performance cluster (Intel(R) Xeon(R) Gold 5220 CPU @ 2.20GHz), it takes around 28 milliseconds (measured with %timeit in ipython) to run torch.nn.InstanceNorm2d(1) on a 3000x3000 image ($c_t$). For a training schedule of 450K iterations with a batch size 8 (our configuration), this would amount to about $28*10^{-3} * 450K * 8/3600 = 28$ hours of added training time solely due to the IN operation. Even with 4 workers (our configuration), this step alone would still require 7 hours. This is about 10% increment in training time for our task. Additionally, when using smaller patch sizes, either batch size or iteration count needs to increase if we want the training to see the data ‘same number of times’, which would only increase the computational cost. For instance, just halving the patch size to 256 would increase the extra computation cost to 28 hours. So, just from the computational aspect itself, IN on full frames should be discouraged.

---

> > > > > > ### Author Response · Authors · 2025-08-06
> > > > > > **Response to more enquiries on iterative nature of inDI.**
> > > > > >
> > > > > > We agree with the reviewer’s suggestions regarding the nomenclature: (a) we will rename our method to scSplit and (b) will update the manuscript to make the difference between inDI and indiSplit crystal clear; among other things, we will specifically replace the word ‘time’ with ‘mixing-ratio’ as suggested.
> > > > > >
> > > > > > In our rebuttal to this reviewer we have already answered the reviewer’s question “can the authors confirm that InDI baselines are applied iteratively?” Nonetheless, we will repeat our argument.
> > > > > >
> > > > > > The authors of inDI paper themselves show quantitatively a non-trivial and monotonous drop in PSNR with increase in number of iterations (For 3 datasets, Fig 9 of InDI shows a monotonous drop in PSNR when increasing the number of iterations, with total drop being atleast 2db PSNR in all three cases). They also mention this observation that using 1 step leads to lowest distortion. So, to keep inDI as competitive as possible on PSNR (in case there is curiosity on why we prefer PSNR over other metrics we used, please refer to our original rebuttal response to this reviewer), we used 1 step inference for inDI. If the reviewer deems it useful, we can discuss this reasoning in the text and refer inDI as inDI$(N=1)$ in the table.

---

> > > > > > > ### Comment · Reviewer_n12V · 2025-08-07
> > > > > > >
> > > > > > > Adjusting nomenclature and text as outlined by the authors above sounds good. If the authors can further incorporate an explicit statement that neither scSplit nor their inDI are used iteratively in their text (possibly including their reasoning regarding PSNR as the metric they deem most valuable), that would address my concerns regarding this topic.
> > > > > > >
> > > > > > > I will note that, while I tend to agree with the notion that among the metrics used by the authors, PSNR is likely the most meaningful one, the actual utility of this and other metrics will depend on the extent to which they predict performance in biologically meaningful downstream experimental use cases. Larger-scale evaluations in this regard (e.g. in cellular profiling applications) are clearly beyond the scope of this paper. Nevertheless, the segmentation results the authors generated in response to reviewer qCeF's concerns regarding more application-relevant downstream could provide some empirical support for this argument, and I encourage the authors refer to consider integrating these results into their argument against using scSplit / inDI iteratively in the context of scientific image analysis applications.

---

> > > > > > > > ### Author Response · Authors · 2025-08-08
> > > > > > > > **On iterative inference**
> > > > > > > >
> > > > > > > > We will clearly state in the main text that neither scSplit nor inDI (N=1) are used iteratively in this work. We will mention how iterative inference influences PSNR in inDI and our rationale for selecting PSNR as the primary evaluation metric. We thank the reviewer for the helpful suggestion to emphasize the alignment between the PSNR and segmentation results we observed, which indicates that iterative inference could potentially have a negative impact on downstream segmentation tasks as PSNR does have a negative impact.
> > > > > > > >
> > > > > > > > Finally, we’re really grateful for the reviewer’s ongoing interest and thoughtful questions about our work. This post-rebuttal phase has been unexpectedly super engaging and has definitely made our work better. Has it improved enough? We hope so and think it has, but we’ll find out soon enough! :D

---

> > > > > > ### Comment · Reviewer_n12V · 2025-08-07
> > > > > >
> > > > > > The authors provide crucial clarification as to the implementational details of their proposed Severity Cognizant Input Normalization method (SCIN for convenience) that are not present in the paper (yet). I appreciate that the authors clarify that what they mean by "single acquisition" is in fact a "dataset" (i.e. a collection of images, possibly z-stacks) that was acquired *over* a single acquisition session. This can be easily addressed by sharpening terminology.
> > > > > >
> > > > > > The rest is actually quite interesting:
> > > > > >
> > > > > > - The authors specify that $\mu_i$ and $\sigma_i$ as constituents of $D$ are actually *not* sampled from images $c_t$ (as section 3.2 suggests and specifically per the equation in line 153) but rather from some patches (random crops?) $x_t$ sampled from $c_t$. The authors will need to clarify this in section 3.2 of their manuscript.
> > > > > >
> > > > > > - In the manuscript, the authors further claim that the point of their method is to ensure that $\mu(t)=0$ and $\sigma^2(t)=1$. Applying IN to every image $c_t$ is a mathematically exact way to ensure this, whereas the SCIN approach (noticeably) approximates this result (D is sampled). Naturally, this also applies to SCIN based on D sampled from image patches $x_t$. With respect to the distributional statistics of $C_t$ ($\mu(t)$ and $\sigma^2(t)$), IN and SCIN achieve the same goal.  The difference then, the authors claim in their response, is that $IN(c_t)$ would not ensure that $E(\sigma^2(x_t)=1$, but I disagree: if $\sigma^2(C_t)=1$ (i.e. the standard deviation of all pixels in $C_t$ is 1) then the standard deviation of all pixels from all patches in $X_t$ pooled together would also be 1. Of course, applying IN would not ensure that $\sigma^2(x_t)=1$ for all individual patches $x_t$, but neither would SCIN.
> > > > > >
> > > > > > - Instead, a *meaningful* difference between IN and SCIN instead is that IN (as correctly stated by the authors) when applied to $c_t$ distorts the structure of the distribution $C_t$, by eliminating statistical differences between *individual* images $c_t$, whereas these remain preserved with SCIN. I am not convinced that variability in signal-intensity (and density) in fluorescence microscopy images (including between z-slices) bears much weight here, as long as either method is applied consistently during training and inference. In fact more aggressive contrast-enhancement applied by IN for low-intensity slices might help highlight weak signal and make it easier for the model to process. The paper + additional results provide no clear empirical evidence one way or the other.
> > > > > >
> > > > > > - **However**, I think that applying IN to all $c_t$ could actually negatively affect the ability of the Reg network to reliably estimate $t$, because by eliminating inter-image statistical differences, IN may discard the very signal that the Reg network requires to estimate t. This could be a nice argument in favor of SCIN vs. IN despite the above which I offer free of charge :)
> > > > > >
> > > > > > - Finally, I am still not convinced that the empirical results on IndiSplit(firstIN) (on patches) are in favor of the authors claims, even when just looking at PSNR: IndiSplit is marginally stronger in 3/6 comparisons. I however gladly yield my claim that IN is more computationally efficient than SCIN.
> > > > > >
> > > > > > In short, I strongly suggest that the authors clarify terminology, revise section 3.2 to explain their sampling method for $D$ (patches), and to clearly explain why SCIN should be preferred over applying IN over all $c_t$ given that the latter is an exact way to achieve $\mu(t)=0$ and $\sigma^2(t)=1$.

---

> ### Author Response · Authors · 2025-08-06
> **Response to more enquiries on Generalization**
>
> We thank the reviewer for figuring out the incorrect channel name for HT-H24 dataset. We will fix it in the text.
>
> *Q: Are superimposed inputs acquired in the same channel for tasks in Table 2?*
>
> **Response** Yes. Table 1 of MicroSplit[5] shows Tasks XIII and XIV as HT-T24 and HT-LIF24, marked ‘TM-III’ (Training Mode III). The paper states (page 4) “Finally, in Training Mode III , we do not create input images by summing images of individual structures but rather acquire them also at the microscope.”
>
> We provide further confirmations from the supplementary text. For HT-T24, in Supplementary section *E.4* of Microsplit[5], the authors state that “the last channel contains the superimposed image containing the abovementioned two structures”. For HT-LIF24, this information is provided in section E.5 (page 29, supplementary material). Using their nomenclature, for our task, we have used GT-B, GT-D and BD channels (GT stands for groundtruth). Here, BD is the channel containing both B and D structures.
>
> We note that these datasets use two stains, which microscopists agree is a practical setup. MicroSplit [5] , a comprehensive study aimed at a biological audience with 10 new datasets, over 30 semantic unmixing tasks, involving several microscopists, also employs this two-stain approach. Below we enumerate the practical benefits:
>
> (a) **Imaging more structures than currently feasible:** Traditionally, when using two fluorophores, they must be chosen to minimize excitation and emission spectral overlap, which limits multi-color imaging to 4 or 5 colors due to difficulty finding compatible fluorophores. Imaging both stains in a single channel eliminates this limitation, enabling the use of more fluorophore pairs with no overlap constraints within each pair and minimal overlap restrictions between different pairs. In this way, more channels can be imaged, simply because a larger set of fluorophores will become compatible with this paired approach.
>
> (b) **Higher acquisition rate:** Additionally, in many parts of the world, multi-color imaging is still done sequentially (channel by channel), and so, a 2-color imaging is 2 times slower than a single color imaging setup. Our inputs, acquired in a single channel, therefore is acquired faster.
>
> (c) **Noise suppression:** Finally, combining signals from different structures reduces pixelwise gaussian noise in the imaged structures, which can allow lower laser power or faster imaging with minimal loss of quality.
>
> *Q: What is generalizability from authors' perspective?*
>
> **Response:** Given two-channel data, several methods exist for semantic unmixing which employ synthetic summation to create the input. Our work focuses on generalizability across mixing ratios—that is, when the relative contributions of the two structures in the superimposed input differ from their synthetic sum, we aim for indiSplit to outperform others. To create a synthetic sum that differs significantly from real balanced superimposed inputs (in HTT24 dataset), we multiplied one channel’s pixel values by 4 in our rebuttal for this reviewer. This made the synthetically summed input dominated by one channel, unlike the balanced real input. When training both indiSplit and baseline models on this asymmetric data, the final table in our rebuttal shows indiSplit clearly outperforms the rest, thanks to its ability to estimate and leverage the mixing ratio ($t_{agg}$).

---

> > ### Comment · Reviewer_n12V · 2025-08-07
> >
> > Finally, I appreciate the authors effort to provide additional information regarding the extent to which their approach generalizes to "real" images, i.e. the *practical* significance of their proposed method.
> >
> > - Unfortunately, the methodological details available in [5] regarding how realistic of a task the images in HT-T24 and HT-LIF24 relevant for Table-2 actually represent are insufficient and substantially below anything that would be acceptable in biomedically oriented peer-reviewed publication venues. But I assume that's not the authors fault. Still, in the future, I would recommend that the authors reach out to relevant dataset generators to ascertain appropriate methodological details if they are otherwise not available. This may no longer be feasible in time for the revision deadline, but (should the paper be accepted) I would request that the authors include this information as feasible in their camera ready version.
> >
> > - Regardless of the extent to which the results in Table-2 pertain to realistic use-cases in line with the targeted application, scSplit does not exhibit a clear performance advantage over other baselines (including in PSNR).
> >
> > With all that being said, I will wait for a final response for the authors before submitting my final evaluation.

---

> ### Author Response · Authors · 2025-08-08
> **Response to the discussion on Normalization (SCIN, IN)**
>
> **Summary**
>
> We thank the reviewer for his continued engagement. We will ensure consistent and correct usage of frame and patch throughout the manuscript, and will generally refine the formalism of Section 3.2. In response to the reviewer’s suggestion, we will mention in the main text that Instance Normalization (IN) can be used for normalization, and will provide the corresponding rebuttal results in the supplementary material. Additionally, we will include a paragraph explaining our preference for SCIN over IN, incorporating arguments regarding its advantages for the Reg network, as very kindly pointed out by the reviewer :) , as well as the computational efficiency of SCIN and the increasing of the efficiency gap between SCIN and IN at smaller patch sizes. We will also address the challenges associated with applying IN to patches, a concern previously acknowledged by the reviewer: “I agree with the authors about the drawbacks about applying IN to patches in place of group or other, larger-context normalization layer.”
>
> ## Specific points
> 1. We apologize for incorrectness at the line 153. Fortunately for us, we can provide the evidence that we work with patches for normalization, since we have provided the code. In split.py:L187, we have the function which returns the normalizer object for SCIN (it returns two objects, one for $Gen_0$ and one for $Gen_1$). As can be seen in the function, it samples from the training dataset object, and therefore works on patches, and not on full frames. We set random_patching=True when we create the training dataset object (split.py:L127).
> 2. With IN applied on $c_t$, $E[\sigma(t)]$ will not be 1: Firstly, we thank the reviewer for providing an argument to use SCIN over IN, ‘free of charge’ :). That is being kind and fair. We also agree that if our notion of $\sigma(t)$ were to pool all pixel values together and take standard deviation, $E[\sigma(t)]$ would be 1, when using IN as well. However, as defined in Supplementary Section A, it is not the definition we use. $E[\sigma(t)]$ is the expected standard deviation of patches of same size as those used during training and inference. (We realize that we have not mentioned that we use patches, and not frames. We will fix the text accordingly). Occasionally, code conveys the meaning more clearly than words, which is why we include a snippet to convey the difference between IN and SCIN.
>
>     ```jsx
>     import numpy as np
>     a = np.arange(1000000).reshape(1000,1000)# 1000x1000
>
>     print('IN ', np.std(a)) # 288675, what IN would use to normalize this image.
>
>     patch_std = []
>     sz = 256
>     for _ in range(10000):
>        i = np.random.randint(1000 - sz)
>        j = np.random.randint(1000 - sz)
>        patch_std.append(a[i:i+sz,j:j+sz].std())
>
>     print('SCIN ', np.mean(patch_std)) # 73900, what SCIN would use as std to normalize patches
>     ```

---

> ### Author Response · Authors · 2025-08-08
> **On discussions about generalization, real inputs**
>
> Please find our responses to different points highlighted by the reviewer.
>
> **scSplit does not exhibit a clear performance advantage on real inputs:** While this may appear to be the case from Table 2, it is no longer accurate. In our rebuttal to this reviewer, under the heading “Generalization seems uncertain”, we presented results on the real inputs of the HTT24 dataset showing that our method achieved approximately a **3 dB PSNR improvement over the next best method**.
>
> For completeness, we explain again why Table 2 did not reflect such a performance gain and what has since changed. Since the MicroSplit authors did not address the mixing-ratio, we hypothesized—with visual assessment— that the relative strengths of the two channels in the real superimposed inputs were similar to those in synthetically generated sums. This meant that the baselines performed competitively simply because the real inputs were effectively in-distribution in terms of mixing-ratio. In this setting, the task was not ideal for evaluating generalization with respect to varying mixing ratios, which is what this work is all about.
>
> To test this hypothesis and create a more suitable scenario, in the rebuttal we modified the HTT24 channel data by multiplying one channel by a factor of four. As this modification can be applied to either channel, we obtained two new datasets in which one channel clearly dominates the other in the synthetic sum. We then trained both our method and the baselines on these modified datasets. As anticipated, when evaluated on the original, unmodified real inputs, our method clearly outperformed—likely because the baselines were unable to handle out-of-distribution mixing ratios.
>
> We will add this table to the main text alongside Table 2.
>
>
> **Relevance of datasets for Table 2:** We respectfully disagree with the reviewer’s assessment regarding the relevance of the datasets sourced from MicroSplit , whose results are presented in Table 2 of the main text. MicroSplit is, to the best of our knowledge, the **only publicly available work** that provides datasets containing **real superimposed inputs** together with their corresponding targets for the semantic unmixing task. This scarcity alone underscores their significance. Moreover, as mentioned in our previous response, the two-stain acquisition method used to acquire the datasets allows to image more structures (due to relaxed emission/excitation spectra constraints), enables faster acquistion with lower noise content.
>
> Furthermore, the biological structures represented in tasks of Table 2—Nucleus, Golgi, microtubules, and centromere—are not rare organelles that microscopists seldom image, but rather commonly studied structures in microscopy. Therefore, the tasks in Table 2 are clearly relevant.
>
> It is also worth highlighting that, for our work, we used all publicly released MicroSplit datasets (two in total) that contained real inputs.
>
> **Insufficient dataset details:** We acknowledge that our original submission provided less than the ideal level of detail regarding the datasets. However, in our rebuttal, we believe we have now included all relevant information. We will also incorporate these details into the supplementary material to ensure that future readers will not need to consult MicroSplit for such specifics.
>
> Additional information—such as microscope type, laser configuration, and stains used—is available in the supplementary material of MicroSplit . Moreover, the raw file metadata contains further microscopy-relevant details. We will extract the pertinent information from these sources and add it to our supplementary text.
>
> With these additions, we are confident that our dataset and task descriptions will be sufficiently comprehensive. Nonetheless, we are willing to contact the authors of MicroSplit for further details and will gladly include any additional information the reviewer considers beneficial.

---

### Note · Authors · 2025-08-14

We want to say that we’re really happy with how much the reviewers engaged in discussions. Every point, from both sides, was addressed and nothing significant was left hanging. So, we don’t have anything more to add. To further assist the area chair, we summarize our discussion outcomes with the reviewers.

Reviewer qCeF appeared satisfied with our response, as indicated by his/her mention of increasing the score. Reviewer ggcu also seemed satisfied, stating, “My concerns have been addressed”. Reviewer Lyze likewise concluded with, “I have no further concern”. With reviewer n12v, we had an engaging discussion which we go into some detail here.
The reviewer requested an ablation of our normalization module, which we provided in the rebuttal. The ablation confirmed its necessity, so discussion shifted to alternate feasible implementations. The reviewer suggested Instance Norm in the first layer as another alternative, but we demonstrated our approach is more computationally efficient, clarified a confusion about 'single acquisition'. n12v agreed and also kindly provided another benefit of our scheme for the Reg network. There was also an inquiry about whether our method or the inDI baseline uses iterative inference. We cited inDI results showing iterative inference increases distortion error, which is more important for our task than perceptual quality. The reviewer agreed, requesting (a) explicit mention that neither method uses iterative inference and (b) renaming our method to scSplit to remove 'iterative' word from the name.
The reviewer expressed concern that our method’s inability to outperform baselines in Table 2 indicated limited generalization. We clarified that Table 2 does not evaluate generalizability with respect to the mixing ratio, which is the central focus of our work. In our rebuttal, we modified the training set so that the training and test sets had different mixing ratios. Under this setup, our method significantly outperformed the baselines, demonstrating strong generalization across mixing ratios. The reviewer questioned the real-world relevance of our tasks. We provided dataset source details, which partially addressed his concerns about generalizing to real images. He still doubted the practical relevance, so we explained that the structures we studied are commonly imaged and that the superimposed acquisition method used in the Table 2 datasets offers advantages—capturing more structures, faster imaging, and reduced noise.

---

### Decision · Program_Chairs · 2025-09-17

**Decision:**

Accept (poster)

**Comment:**

as all the reviewers decide to accept this paper, i recommend to accept this paper. however so many issues are discussed during the rebuttal phase. the authors should revise the paper based on the discussion.